# Low Degree Hardness for Broadcasting on Trees

**Han Huang**
Department of Mathematics
University of Missouri
Columbia, MO 65203
hhuang@missouri.edu

**Elchanan Mossel**
Department of Mathematics
MIT
Cambridge, MA 02139
elmos@mit.edu

## Abstract

We study the low-degree hardness of broadcasting on trees. Broadcasting on trees has been extensively studied in statistical physics, in computational biology in relation to phylogenetic reconstruction and in statistics and computer science in the context of block model inference, and as a simple data model for algorithms that may require depth for inference.

The inference of the root can be carried by celebrated Belief Propagation (BP) algorithm which achieves Bayes-optimal performance. Recent works indicated that this algorithm in fact requires high level of complexity. Moitra, Mossel and Sandon constructed a chain for which estimating the root better than random (for a typical input) is $NC1$ complete. Kohler and Mossel constructed chains such that for trees with $N$ leaves, recovering the root better than random requires a polynomial of degree $N^{\Omega(1)}$. Both works above asked if such complexity bounds hold in general below the celebrated *Kesten-Stigum* bound.

In this work, we prove that this is indeed the case for low degree polynomials. We show that for the broadcast problem using any Markov chain on trees with $N$ leaves, below the Kesten Stigum bound, any $O(\log N)$ degree polynomial has vanishing correlation with the root.

Our result is one of the first low-degree lower bound that is proved in a setting that is not based or easily reduced to a product measure.

## 1 Introduction

Understanding the computational complexity inference problems of random instances has been extensively studies in different research areas including statistics, cryptography, computational complexity, computational learning theory and statistical physics. The emerging field of research is mainly devoted to the study of computational-to-statistical gaps. ([3, 41, 24])

Recently, low-degree polynomials have emerged as a popular tool for predicting computational-to-statistical gaps, especially in the context of the Bayesian framework. Our work follows [23] in studying the polynomial hardness of broadcasting on trees.

A very exciting line of work, including [19, 18, 25, 4, 15, 26, 17, 9, 40] recently showed that the "low-degree heuristic" can be used to predict computational-statistical gaps for a variety of problems such as recovery in general stochastic block models, sparse PCA, tensor PCA, the planted clique problem, certification in the zero-temperature Sherrington-Kirkpatrick model, the planted sparse vector problem, and for finding solutions in random $k$-SAT problems.

Interestingly, it was observed that the predictions from this method often agree with predictions from statistical physics heuristics based on the replica and cavity methods which are closely related to the analysis of BP/AMP fixed points, see e.g. [12, 13, 28]).

38th Conference on Neural Information Processing Systems (NeurIPS 2024).

It is often argued that low-degree polynomials algorithms are relatively easy to use (e.g. compared to proving SOS lower bounds), and that low degree polynomials capture the power of the "local algorithms" framework used in e.g. [16, 10] as well as algorithms which incorporate global information, such as spectral methods or a constant number of iterations of Approximate Message Passing [29].

In this work, we continue to study the power of low-degree polynomials for the (average case) broadcast on trees problem. In broadcasting on trees the goal is to estimate the value of the Markov process at the root given its value at the leaves and the goal is to do so for arbitrarily deep trees. Two key parameters of the model are the arity of the tree $d$ and the magnitude of the second eigenvalue $\lambda$ of the broadcast chain.

A fundamental result in this area [22] is that when $d|\lambda|^2 > 1$ nontrivial reconstruction of the root is possible by counting the number of the leaves of different types, an algorithm that could be described as a linear function of the leave values. In contrast, when $d|\lambda|^2 < 1$, such linear estimators have no mutual information with the root (but more complex statistics of the leaves may) [30, 36].

This threshold $d|\lambda|^2 = 1$ is known as the *Kesten-Stigum threshold*. A series of works showed that the KS threshold is the information theory threshold for non-trivial root inference for some specific channels, including the binary symmetric channel [6, 14, 21, 20] and binary channels that are close to symmetric [8], as well as $3 \times 3$ symmetric channels for large $d$ [39].

While the Kesten-Stigum bound is easy to compute, it turns out that in many cases, it is *not* the information-theoretic threshold for root recovery. This was first established in [30] for symmetric channels with sufficiently many states, specifically when $q \geq C$ for some large constant $C$. Later it was shown for symmetric channels with $q \geq 5$ states in [39]. Recently, the results [37] provide more information about the case of $q = 3$ and $q = 4$. Many of the finer results in this area prove predictions from statistical physics. The connection between the broadcast problems and phase transitions in statistical physics was made in [27], and more recent predictions include [5, 2, 38]. Moreover, the information-theoretic threshold may depend on the specific structure of the channel rather than solely on $d$ and $\lambda$. Notably, [30] also showed that there exists channels where non-trivial predictions of the root are achievable even when $|\lambda| = 0$.

Much of the interest in Kesten-Stigum threshold comes from the fundamental role it plays in problems, such as algorithmic recovery in the stochastic block model [11, 33, 7, 35, 1] and phylogenetic reconstruction [31]. Count statistics can be viewed as degree 1 polynomials of the leaves, which begs the question of what information more general polynomials can extract from the leaves. See [32, 34] for surveys on the topic.

In [23] it was shown that $\lambda = 0$ even polynomials of degree $N^c$, where $N = d^\ell$ is the number of leaves of for a $d$-ary tree of depth $\ell$, for a small $c > 0$ are not able to correlate with the root label (as $\ell$ tends to $\infty$), whereas computationally efficient reconstruction is generally possible as long as $d$ is a sufficiently large constant [30].

The main motivation of [23] was to prove that low degree polynomials fail below the Kesten Stigum bound: "It is natural to wonder if the Kesten-Stigum threshold $d|\lambda|^2 = 1$ is sharp for low-degree polynomial reconstruction, analogous to how it is sharp for robust reconstruction." However the main result of [23] only established this in the very special case of $\lambda = 0$. This problem is also stated in the ICM 2022 paper and talk on the broadcast process [34]: " The authors of [23] ask if a similar phenomenon holds through the non-linear regime. For example, is it true that polynomials of bounded degree have vanishing correlation with $X_0$ in the regime where $d\lambda^2 < 1$? " The main results of this paper prove that this is indeed the case. We proceed with formal definitions and statement of the main result.

## 1.1 Definitions and Main Result

### Rooted Tree

Recall that every rooted tree $T$ inherently defines a partial order relation among its vertices: For a pair of distinct vertices, $u$ is said to be an ancestor of $v$ (and $v$ a descendant of $u$), denoted as $v < u$ in this paper, if $u$ is contained in the unique path from $v$ to the root $\rho$. Specifically, if $v$ and $u$ are directly connected by an edge, we also refer to $v$ as a child vertex of $u$ (and $u$ as the parent vertex of $v$). By $v \leq u$ we mean that $v$ is either a descendant of $u$ or $v = u$. In general, if $v < u$ and the path distance between them is $k$, $v$ is referred to as a $k$th-descendant of $u$ (and $u$ as the $k$th ancestor of $v$).

For a nonnegative integer $k$, the $k$th layer of the tree refers to the set of $k$th descendants of the root $\rho$. (The root here is considered at the 0th layer of the tree.) The depth of the tree is defined as the largest non-negative integer $\ell$ for which the $\ell$th layer is not empty, and we denote the $\ell$th layer by $L$.

The height of a vertex $u$ is defined as

$$\mathrm{h}(u) = \ell - \text{ the layer of } u. \tag{1}$$

In particular, when $L$ is the set of leaves, then $\mathrm{h}(u)$ is simply the distance from $u$ to $L$. In this paper, we may abuse the notation by writing $x \in T$ or $S \subseteq T$ to mean that $x$ is a vertex or $S$ is a subset of vertices in the tree $T$.

The standard rooted $d$-ary tree with depth $\ell$ is a tree where each vertex $u \notin L$ has exactly $d$ children vertices. Let us start by defining the type of trees we will be investigating in this paper, which is a slight generalization of $d$-ary tree.

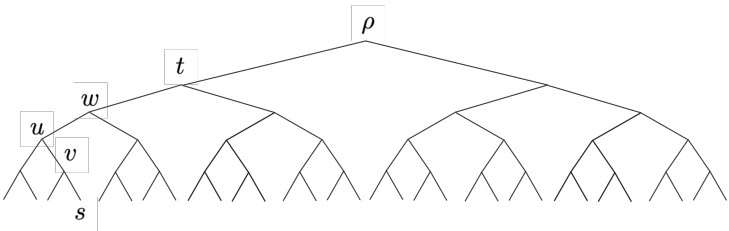

Figure 1: An example of a binary rooted tree of depth 5 is shown. The vertex $u$ is at the 3rd layer and $\mathrm{h}(u) = 2$. Further, the following relationships hold: $v < u$ and $v$ is a child of $u$, $s \in L$ is a 2nd descendant of $u$, $w$ is the parent of $u$, and $t$ is the 2nd ancestor of $u$.

**Definition 1.1.** *A rooted tree $T$ with root $\rho$ has degree dominated by $d \geq 1$ with parameter $R \geq 1$ if for every vertex $u$ and positive integer $k$, the number of $k$th descendants of $u$ is at most $Rd^k$.*

With the above definition, a $d$-ary rooted tree has degree dominated by $d \geq 1$ with parameter $R = 1$. Further, a typical realization of a Galton-Watson tree of Poisson type with average degree $d$ and of depth $\ell$ (a random tree in which each vertex $u \notin L$ has on average $d$ children vertices) has a degree dominated by $d \geq 1$ with parameter $R \simeq \log(\ell)$.

**Broadcasting Process on Rooted Trees**

Next, we will define the broadcasting process on a rooted tree $T$ with root $\rho$. Consider a $q \times q$ ergodic transition matrix $M$, where $q \geq 2$. Recall that every eigenvalue of a transition matrix $M$ has an absolute value at most 1. Let $0 \leq \lambda \leq 1$ represent the second largest absolute value among the eigenvalues of $M$. Additionally, we define the stationary distribution of $M$ as $\pi$.

The broadcasting process $X = (X_v)_{v \in T}$, with state space $[q] := \{1, 2, \ldots, q\}$ and transition matrix $M$, can be formally described as follows: We initialize the value of $X_\rho$ according to some initial distribution $\nu$. As we reveal the values layer by layer, when the value $X_u$ is revealed, the value $X_v$ for any child vertex $v$ of $u$ is independently distributed according to a specific row of $M$ depending on the value of $X_u$:

$$\mathbb{P}\{X_v = t \mid X_u = s\} = M_{st}.$$

In other words, each vertex's value depends only on its parent vertex's value. The definition of the process is given below:

**Definition 1.2** (Broadcasting Process on Tree). *Let $q \geq 2$ be a positive integer. For any rooted tree $T$ with root $\rho$ and a $q \times q$ ergodic transition matrix $M$, the broadcasting process $X = (X_v)_{v \in T}$ with state space $[q]$, according to transition matrix $M$ with an initial distribution $\nu$ for $X_\rho$, is a random process with joint distribution given by:*

$$\forall x = (x_v)_{v \in T} \in [q]^T, \quad \mathbb{P}\{X = x\} = \nu(x_\rho) \prod_{(v,u)} M_{x_u, x_v},$$

*where the product is taken over all pairs $(v, u)$ such that $v$ is a child vertex of $u$.*

*For a subset of vertices $A \subset T$, let us denote*

$$X_A = (X_v)_{v \in A}.$$

If the tree is just a path, then the process reduces to a Markov chain. If we assume $\nu = \pi$, then $X_v \sim \pi$ for every $v \in T$, as $(X_v)_{v \in P}$ for every downward path of $T$ forms a Markov Chain with transition matrix $M$. Further, let us make a remark about the Markov property of the process.

**Remark 1.3** (Markov Property)**.** The broadcasting process establishes a **Markov Random Field** on tree $T$: Given any three disjoint subsets $A$, $B$, and $C$ of $T$, if every path from a vertex in $A$ to a vertex in $C$ passes through a vertex in $B$, then the random variables $X_A$ and $X_C$ are conditionally independent given $X_B$.

### Polynomials of $x_L$ and the Main Result

**Definition 1.4.** *Let $x \in [q]^T$. For $u \in T$, let $x_{\leq u} = (x_v)_{v \leq u}$. For subset $U \subseteq T$, let $x_U = (x_u)_{u \in U}$.*

The next definition is about the notion of degrees for functions with variables $x_L = (x_v)_{v \in L}$. This is the generalization of degree of a polynomial.

**Definition 1.5** (Efron-Stein Degree)**.** *A function $f$ with variables $x_L$ has Efron-Stein degree at most $k$ if it can be expressed as*

$$f = \sum_\alpha f_\alpha,$$

*where the summation is over a finite set of indices $\alpha$, and each $f_\alpha$ is a function of the variable $x_S$ for some $S \subseteq L$ with $|S| \leq k$.*

Now, we could properly formulate the main result of the paper:

**Theorem 1.6.** *Let $T$ be a rooted tree with root $\rho$ of depth $\ell$ and has degree dominated by $d \geq 1$ with parameter $R \geq 1$. Consider the broadcasting process on $T$ with a $q \times q$ transition matrix $M$ and $X_\rho \sim \pi$. If $M$ is ergodic and $d\lambda^2 < 1$, then there exists a constant $c > 0$ which depends on $M$ and $d\lambda^2$ such that the following holds: For any function $f(x_L)$ of Efron-Stein degree $\leq c\frac{\ell}{1+\log(R)}$, we have*

$$\mathrm{Var}(\mathbb{E}\big[f(X_L)\,\big|\,X_\rho\big]) \leq (\max\{d\lambda^2, \lambda\})^{\ell/4}\mathrm{Var}(f(X_L)).$$

**Remark 1.7.** Given that $d\lambda^2 < 1$ implies $\lambda^2 < 1$, and $\lambda < 1$ if and only if $\lambda^2 < 1$, we can infer that $\lambda < 1$ from the given conditions. Consequently, the term $\max\{d\lambda^2, \lambda\} < 1$ follows from the assumptions of the theorem. Therefore, the R.H.S. of the inequality decays exponentially with the depth of the tree.

Follows from the theorem and propteries of conditonal expectation, we have the following corollary.

**Corollary 1.8.** *With the same setting as in Theorem 1.6, for any function $f(x_L)$ of Efron-Stein degree $\leq c\frac{\ell}{1+\log(R)}$, and any function $g(x_\rho)$ of the root value, their correlation satisfies*

$$\mathrm{Corr}(f(X_L), g(X_\rho)) := \frac{\mathbb{E}\big[(f(X_L) - \mathbb{E}f(X_L))(g(X_\rho) - \mathbb{E}g(X_\rho))\big]}{\sqrt{\mathrm{Var}(f(X_L)}\sqrt{\mathrm{Var}(g(X_\rho))}} \leq (\max\{d\lambda^2, \lambda\})^{\ell/8}.$$

The proof of the theorem is based on recursion on a notion of *fractal capacity* of functions. Indeed, the main result is optimal in the fractal sense (Theorem 1.16), as we will later demonstrate that all functions with fractal capacity up to a level proportional to $\ell$ exhibit vanishing correlation with the root, whereas all functions of the leaves have fractal capacity at most $\ell + 1$. Let us introduce the necessary definitions and notations to introduce both the fractal capacity and the proof overview.

### 1.2   Fractal Capacity and Proof Overview

To provide a clearer illustration, we establish a correspondence between the vertices of $T$ and words of varying lengths from $0$ to $\ell$, with vertices at the $k$th layer represented as words of length $k$. We

denote the root $\rho$ as the empty word (). For each vertex $u$, represented by the word $(b_1, b_2, \ldots, b_k)$, we define $d_u$ as the number of children vertices of $u$, and we identify these children vertices as $(b_1, b_2, \ldots, b_k, i)$ with $i \in [d_u] := \{1, 2, \ldots, d_u\}$. Notice that $v$ is a descendant of $u$ is equivalent to $u$ is a prefix of $v$. For brevity, for each $u = (b_1, \ldots, b_k) \in T$ and $i \in [d_u]$, let

$$u_i := (u, i) = (b_1, \ldots, b_k, i).$$

For $I \subseteq [d_u]$, let

$$u_I = \{u_i\}_{i \in I}.$$

Furthermore, we denote the parent vertex of $u$ as $\mathfrak{p}(u) = (b_1, \ldots, b_{k-1})$ and the set of children vertices of $u$ as $\mathfrak{c}(u) = u_{[d_u]}$.

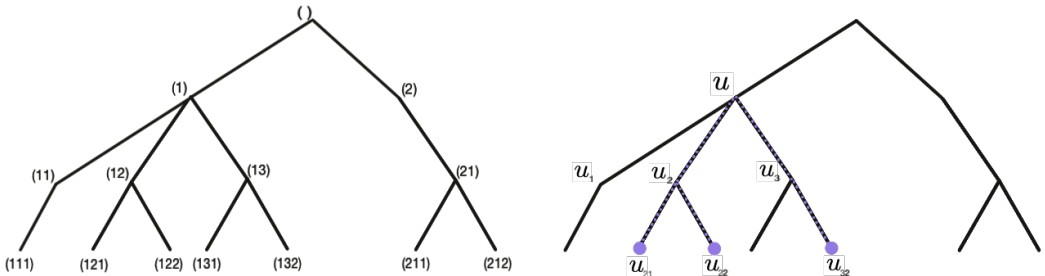

Figure 2: In the these figures, we present the vertices as words and adapt the notations $u_1, u_2$, etc., for the descendants of $u$. For the right figure, if $S = \{u_{21}, u_{22}, u_{32}\}$, then $\rho(S) = u$, $I(S) = \{2, 3\}$, and $S_2 = \{u_{21}, u_{22}\}$, $S_3 = \{u_{32}\}$. Further, $S_1 \in \mathcal{A}_3$, $S_2 \in \mathcal{A}_2$, and $S_3 \in \mathcal{A}_1$.

**Definition 1.9.** *For a non-empty subset $S \subseteq L$, we introduce the notation $\rho(S)$ to represent the nearest common ancestor of the vertices in $S$, meaning that $\rho(S)$ is the vertex with smallest height that is an ancestor of all vertices in $S$.*

*Here we consider the case when $|S| > 1$. Notice that $\rho(S)$ is not at the $\ell$th layer $L$. For each child $\rho(S)_i$ of $\rho(S)$ for $i \in [d_{\rho(S)}]$, we define the set $S_i$ as*

$$S_i = S \cap \{v \in L : v \le \rho(S)_i\},$$

*which is the collection of vertices in $S$ that are descendants of $\rho(S)_i$. Let*

$$I(S) = \{i \in [d_{\rho(S)}] : S_i \ne \emptyset\}.$$

*Then, $S$ can be expressed as the disjoint union of the sets $S_i$ for $i \in I(S)$:*

$$S = \sqcup_{i \in I(S)} S_i.$$

*We call the above disjoint union the **branch decomposition** of $S$, and each $S_i$ for $i \in I(S)$ a **branch part** of $S$.*

The branch decomposition is a key concept in the proof, and we define the *fractal capacity* according to the number of iterations to decompose $S$ into singletons.

**Definition 1.10.** *Let*

$$\mathcal{A}_1 := \Big\{ \{u\} : u \in L \Big\} \subseteq \mathbf{2}^L \backslash \{\emptyset\},$$

*be the collection of singletons of $L$. We say a subcollection $\mathcal{A} \subseteq \mathbf{2}^L \backslash \{\emptyset\}$ is **closed under decomposition** with base $\mathcal{A}_1$ if for every $S \in \mathcal{A} \backslash \mathcal{A}_1$, we have $S_i \in \mathcal{A}$ for $i \in I(S)$.*

**Definition 1.11.** *For any $\mathcal{A} \subseteq \mathbf{2}^L \backslash \{\emptyset\}$, let*

$$\mathcal{B}(\mathcal{A}) \subseteq \mathbf{2}^L \backslash \{\emptyset\}$$

*be a new subcollection defined according to the following rules:*

*For any $S \in \mathbf{2}^L \backslash \{\emptyset\}$, $S \in \mathcal{B}$ if and only if one of the following two conditions holds*

*1.* $S \in \mathcal{A}_1$.

*2.* $S \notin \mathcal{A}_1$ and $S_i \in \mathcal{A}$ for $i \in I(S)$.

For example, $\mathcal{B}(\mathcal{A})$ contains sets of size $\leq 2$.

**Lemma 1.12.** *If $\mathcal{A}_1 \subseteq \mathcal{A} \subseteq \mathbf{2}^L \backslash \{\emptyset\}$ is a subcollection closed under decomposition with base $\mathcal{A}_1$, then the collection $\mathcal{B} = \mathcal{B}(\mathcal{A})$ contains $\mathcal{A}$ and it is also closed under decomposition with base $\mathcal{A}_1$.*

*Proof.* To show $\mathcal{A} \subseteq \mathcal{B}$, it is sufficient to show $\mathcal{A} \backslash \mathcal{A}_1 \subseteq \mathcal{B}$. For any $S \in \mathcal{A} \backslash \mathcal{A}_1$, because $\mathcal{A}$ is closed under decomposition, $S_i \in \mathcal{A}$ for $i \in I(S)$. Hence, $S \in \mathcal{B}$ follows from the definition of $\mathcal{B}$. Now, for $S \in \mathcal{B} \backslash \mathcal{A}_1$, each $S_i$ with $i \in I(S)$ is contained in $\mathcal{A} \subseteq \mathcal{B}$, which in turn implies $\mathcal{B}$ is closed under decomposition. □

Now, we define recursively that

$$\mathcal{A}_k = \mathcal{B}(\mathcal{A}_{k-1}), \tag{2}$$

for positive integer $k \geq 2$. Observe the following two facts: Consider any non-singleton set $S \subseteq L$ and $S_i$ with $i \in I(S)$. First, $S \in \mathcal{A}_k \Rightarrow S_i \in \mathcal{A}_{k-1}$ by the definition of $\mathcal{A}_k$. Second, $\rho(S) > \rho(S_i)$ by the definition of branch decomposition. Given these two facts, we can prove inductively that

$$\mathrm{h}(\rho(S)) = k \Rightarrow S \in \mathcal{A}_{k+1}.$$

Since that there are only $\ell$ layers of the tree, we conclude that every non-emptyset of $S \subseteq L$ is in $\mathcal{A}_{\ell+1}$. Therefore, together with Lemma 1.12, we have the following chain of subcollections:

$$\{\{u\} \, : \, u \in L\} = \mathcal{A}_1 \subseteq \mathcal{A}_2 \subseteq \cdots \subseteq \mathcal{A}_{\ell+1} = \mathbf{2}^L \backslash \{\emptyset\}.$$

**Definition 1.13** (Fractal Capacity). *For any non-empty subset $S \subseteq L$, we define the **fractal capacity** of $S$ as the smallest $k$ such that $S \in \mathcal{A}_k$.*

We introduce the notion of fractal capacity, borrowing terminology from fractal geometry. The recursive nature of our definition on subsets of trees mirrors the self-similar complexity found in fractal structures. This recursive and inherently intricate structure motivates our choice of the term fractal capacity, capturing the fractal-like properties that emerge in the collections $(\mathcal{A}_k)_{k \in [l+1]}$.

**Definition 1.14.** *Given a collection $\mathcal{A} \subseteq \mathbf{2}^L \backslash \{\emptyset\}$. A function $f : [q]^T \to \mathbb{R}$ is called an $\mathcal{A}$-**polynomial** if we can express*

$$f(x) = \sum_{S \in \mathcal{A}} f_S(x_S)$$

*where each $f_S$ is a function of $x_S = (x_v)_{v \in S}$. A function $f : [q]^T \to \mathbb{R}$ has **fractal capacity** $\leq k$ if it is a $\mathcal{A}_k$-polynomial.*

**Remark 1.15.** It is not hard to verify that $\mathcal{A}_k$ contains all non-empty subsets $S \subseteq L$ with $|S| \leq k$. Thus, for any function $f$ with variables $x_L$,

$$\text{Efron-Stein degree of } f \leq k \Rightarrow f \text{ is an } \mathcal{A}_k\text{-polynomial.}$$

On the other hand, it is worth to remark that for the $d$-ary tree of depth $\ell \geq k$, there exists $S \in \mathcal{A}_k$ with $|S| = d^{k-1}$. (Namely, taking $S = \{v \in L \, : \, v < u\}$ for some $u$ with $\mathrm{h}(u) = k - 1$.) Thus, an $\mathcal{A}_k$-polynomial could have an Efron-Stein degree exponential in $k$.

The main result of the paper in terms of the fractal capacity is the following:

**Theorem 1.16.** *With the same setting as in Theorem 1.6, there exists a constant $c > 0$ which depends on $M$ and $d\lambda^2$ such that the following holds: For any function $f(x_L)$ with **fractal capacity** $\leq c \frac{\ell}{1 + \log(R)}$, we have*

$$\mathrm{Var}(\mathbb{E}\big[f(X_L) \,\big|\, X_\rho\big]) \leq (\max\{d\lambda^2, \lambda\})^{\ell/4} \mathrm{Var}(f(X_L)).$$

Indeed, Theorem 1.6 is a direct consequence of Theorem 1.16, as $\mathcal{A}_k$-polynomials contains all polynomials of Efron-Stein degree $\leq k$. Further, in terms of fractal capacity, the theorem is optimal because the correlation decay persists up to an order proportional to $\ell$, while a fractal capacity $\leq \ell$ includes all functions of the leaves.

**Overview of the Proof Idea:** For illustration, let us consider the case where $T$ is a binary tree of depth $\ell$ with $M = \begin{bmatrix} \frac{1+\lambda}{2} & \frac{1-\lambda}{2} \\ \frac{1-\lambda}{2} & \frac{1+\lambda}{2} \end{bmatrix}$, such matrix has eigenvalues $\lambda$ and 1. Here we assume $2\lambda^2 < 1$.

We recall that for this binary symmetric broadcasting process, it is information-theoretically impossible to recover the root label from the leaves below the KS bound. This implies that all polynomials of $X_L$ have vanishing correlation with $X_\rho$. Still, we use this simple process to illustrate the proof idea as our arguments for low-degree polynomials generalize to general broadcasting processes below the Kesten-Stigum threshold, including cases where it is information theoretically possible to estimate the root from the leaves non-trivially (in such cases there exist functions $f$ so that $\liminf_{\ell \to \infty} \mathrm{Corr}(f(X_L), X_\rho) > 0$).

Now, let us consider degree-1 polynomials. Suppose $f$ is a $\mathcal{A}_1$-polynomial (equivalently, of Efron-Stein degree 1), we can express it in the form

$$f(x_L) = \sum_{u \in L} f_u(x_u),$$

where each $f_u$ is a function of $x_u$. Given our focus on the variance, we may assume $\mathbb{E}f_u(X_u) = 0$ for each $u \in L$. Then, our goal is to prove $\mathbb{E}\big[\big(\mathbb{E}[f(X_L) \,|\, X_\rho]\big)^2\big]$ is negligible comparing to $\mathbb{E}\big[(f(X_L))^2\big]$. Following from the Cauchy-Schwarz inequality that

$$\big( \sum_{i \in [m]} a_i \big)^2 = \big( \sum_{i \in [m]} 1 \cdot a_i \big)^2 \leq m \sum_{i \in [k]} a_i^2,$$

we have

$$\mathbb{E}\big[\big(\mathbb{E}[f(X_L) \,|\, X_\rho]\big)^2\big] \leq |L| \sum_{u \in L} \mathbb{E}\big[\big(\mathbb{E}[f_u(X_u) \,|\, X_\rho]\big)^2\big] \tag{3}$$

$$\lesssim 2^\ell \sum_{u \in L} \lambda^{2\ell} \mathbb{E}\big[(f_u(X_u))^2\big] = (2\lambda^2)^\ell \sum_{u \in L} \mathbb{E}\big[(f_u(X_u))^2\big],$$

where the second inequality is derived from the variance decay property of in a Markov Chain. Thus, if we can establish $\sum_{u \in L} \mathbb{E}\big[(f_u(X_u))^2\big]$ is at the same order as $\mathbb{E}\big[(f(X_L))^2\big]$, the proof is complete.

Notice that

$$\mathbb{E}\big[(f(X_L))^2\big] = \sum_{u,v \in L} \mathbb{E}[f_u(X_u) f_v(X_v)] = \sum_{u \in L} \mathbb{E}\big[(f_u(X_u))^2\big] + \sum_{u \neq v \in L} \mathbb{E}[f_u(X_u) f_v(X_v)].$$

Thus, the goal here is to show

$$\Big| \sum_{u \neq v \in L} \mathbb{E}[f_u(X_u) f_v(X_v)] \Big| < c \sum_{u \in L} \mathbb{E}\big[(f_u(X_u))^2\big], \tag{4}$$

for some constant $c \in (0,1)$, which in turn implies the desired result:

$$\mathbb{E}\big[\big(\mathbb{E}[f(X_L) \,|\, X_\rho]\big)^2\big] \lesssim (2\lambda^2)^\ell \sum_{u \in L} \mathbb{E}\big[(f_u(X_u))^2\big] \lesssim (2\lambda^2)^\ell \frac{1}{1-c} \mathbb{E}\big[(f(X_L))^2\big].$$

Roughly speaking, (4) holds if for most pairs $u$ and $v$ within $L$, the correlation between $f_u(X_u)$ and $f_v(X_v)$ is sufficiently small, which is the case for degree-1 polynomials.

Now, let us take a closer look. Fix any two vertices $u$ and $v$ in $L$, with $w$ as their nearest common ancestor. Suppose $u \leq w_1$ and $v \leq w_2$. Let $X_{\not\leq w_1} = (X_{u'})_{u' \not\leq w_1}$. We have

$$\big|\mathbb{E}[f_u(X_u) f_v(X_v)]\big| = \big|\mathbb{E}[\mathbb{E}[f_u(X_u) \,|\, X_{\not\leq w_1}] f_v(X_v)]\big| \tag{5}$$

$$\leq \sqrt{\mathbb{E}[(\mathbb{E}[f_u(X_u) \,|\, X_{\not\leq w_1}])^2]} \cdot \sqrt{\mathbb{E}[(f_v(X_v))^2]}$$

$$\lesssim \lambda^{\mathrm{h}(w)} \sqrt{\mathbb{E}[(f_u(X_u))^2]} \sqrt{\mathbb{E}[(f_v(X_v))^2]},$$

where the last inequality follows from the variance decay of the Markov Chain for length $\mathrm{h}(w)$. The above inequality implies that the correlation between $f_u(X_u)$ and $f_v(X_v)$ is at most of order $\lambda^{\mathrm{h}(w)}$.

The above bound can be improved to $\lambda^{2k}$ by also taking the conditional expectation $\mathbb{E}[f_v(X_v) \,|\, X_{\nleq w_2}]$ into account, which requires the Markov Property that $X_u$ and $X_v$ are independent conditioned on $X_w$. From here, one can properly arrange the terms and apply Cauchy-Schwarz inequality to show (4) holds.

Our proof of the main theorem tries to generalize the argument above to low degree polynomials. Let us summarize it by the following five pieces of descriptions.

**I. Bounding Covariance:** First, we generalized the idea on how (5) works for degree-1 polynomials. Suppose $f_\alpha$ and $f_\beta$ are two functions so that the following holds.

1. $f_\alpha(x_S)$ is a function of $x_S$ for some set $S \subseteq L$ such that
$$\mathbb{E}[(\mathbb{E}[f_\alpha(X_S) \,|\, X_{\nleq w'}])^2] \ll (\mathbb{E}(f_\alpha(X_S)))^2,$$
where we use $a \ll b$ to indicate $a$ is much smaller than $b$. We keep this not precise to avoid technical details, but expect that the ratio is exponentially small in $\mathrm{h}(w')$.

2. $f_\beta(x_{S'})$ is a function of $x_{S'}$ with $S' \subseteq L$ satisfying $S' \cap \{v' : v' \le w'\} = \emptyset$.

Then, following the same derivation as shown in (5) we have
$$\left| \mathbb{E}[f_\alpha(X_S) f_\beta(X_{S'})] \right| = \left| \mathbb{E}\big[\mathbb{E}[f_\alpha(X_S) \,|\, X_{\nleq w'}] f_\beta(X_{S'})\big] \right| \ll \sqrt{\mathbb{E}[(f_\alpha(X_S))^2]} \sqrt{\mathbb{E}[(f_\beta(X_{S'}))^2]}.$$

(See Figure 3 for an illustration.)

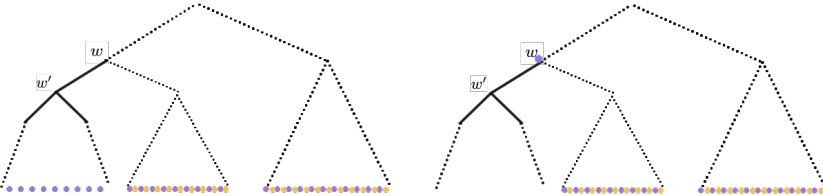

Figure 3: In the left figure, the purple dots represent the corresponding input variables for $f_\alpha$, and the yellow dots represent the corresponding input variables for $f_\beta$. In the right figure, the purple dots represent the corresponding input variables for $\mathbb{E}[f_\alpha(X) \,|\, X_{\nleq w_1}]$ and the variables do not involve $x_v$ for vertex $v$ not illustrated due to the Markov property.

**II. Choosing a good decomposition of the function:** In essence, our proof strategy for any given function $f(x_L)$ with $\mathbb{E}f(X_L) = 0$ revolves around decomposing $f(x_L)$ into a sum of functions $f_\alpha(x_L)$ for $\alpha$ in some index set $\mathcal{I}$, such that

1. $|\mathcal{I}| \lesssim 2^\ell$,
2. For each $\alpha$, $\mathbb{E}f_\alpha(X_L) = 0$ and $\mathbb{E}[(\mathbb{E}[f_\alpha(X_L) \,|\, X_\rho]^2] \ll \mathbb{E}[(f_\alpha(X_L))^2]$,
3. Whenever $\alpha \neq \beta$, we can find $w \in T$ so that $f_\alpha$ and $f_\beta$ satisfy the covariance bound in **I**. (Possibly with a switch of the roles of $\alpha$ and $\beta$.)

Let us remark that while we are writing $f_\alpha(x_L)$, it does not mean that $f_\alpha$ depends on every leave variables.

If this is the case, then we could follow the argument in the degree 1 case to show that desired result holds.

**III. From $\mathcal{A}_k$ polynomials to $\mathcal{A}_{k+1}$ polynomials:** The proof of the main theorem builds on **I** and **II** and advancing through a recursion on the fractal capacity of the function. This recursive approach relies on the following property:

Suppose we have shown the second moment decay $\mathcal{A}_k$-polynomials with mean 0 in the following sense: For every $\mathcal{A}_k$-polynomial $f(x_S)$ with mean 0 and variable $x_S$ where $S \subseteq \{v \in L : v \le \rho'\}$ for some vertex $\rho'$,
$$\mathbb{E}[(\mathbb{E}[f(X_L) \,|\, X_{\rho'}])^2] \le (2\lambda^2)^{\mathrm{h}(\rho') - \mathrm{h}_{\mathcal{A}_k}} \mathbb{E}[(f(X_L))^2], \tag{6}$$

where $\mathrm{h}_{\mathcal{A}_k}$ is some penalty constant depending on $\mathcal{A}_k$. (The bigger the value $\mathrm{h}_{\mathcal{A}_k}$, the weaker the second moment decay.)

Consider a specific type of $\mathcal{A}_{k+1}$-polynomials. Fix a pivot vertex $w$ with $\mathrm{h}(w)$ large enough, let $x_{\leq w_i} = (x_v)_{v \leq w_i}$ for $i \in [2]$. Let $g$ be a function of the form

$$g(x) = g(x_{\leq w_1}, x_{\leq w_2}) = \sum_{\alpha \in \mathcal{J}} g_{\alpha, 1}(x_{\leq w_1}) \cdot g_{\alpha, 2}(x_{\leq w_2}),$$

where $\mathcal{J}$ is a finite index set and every $g_{\alpha, i}$ is an $\mathcal{A}_k$-polynomial whose variables are $x_{\leq w_i}$ (more precisely, its variables are $x_{S'}$ where $S' \subseteq \{v \in L : v \leq w_i\}$) and $\mathbb{E}g_{\alpha, i}(X_{\leq w_i}) = 0$. Then, the function $g(x)$ is a $\mathcal{A}_{k+1}$-polynomial with variable $x_S$ for $S \subseteq \{v \in L : v \leq w\}$.

Observe that, if we fix $x_{\leq w_2}$, then $x_{\leq w_1} \mapsto g(x_{\leq w_1}, x_{\leq w_2})$ is an $\mathcal{A}_k$-polynomial with variable $x_{S'}$ for $S' \subseteq \{v \in L : v \leq w_1\}$ and mean 0. This allows us to apply the assumption for $\mathcal{A}_k$ polynomials to show

$$\mathbb{E}[(\mathbb{E}[g(X_{\leq w_1}, x_{\leq w_2}) \mid X_{\not\leq w_1}])^2] \lesssim (d\lambda^2)^{\mathrm{h}(w) - \mathrm{h}_{\mathcal{A}_k}} \mathbb{E}[(g(X_{\leq w_1}, x_{\leq w_2}))^2].$$

Clearly, the same inequality holds with the roles of $x_{\leq w_1}$ and $x_{\leq w_2}$ switched.

Base on this, one key step in the paper is to show $g$ satisfies

$$\mathbb{E}[(\mathbb{E}[g(X_{\leq w_1}, X_{\leq w_2}) \mid X_{\not\leq w_i}])^2] \lesssim (d\lambda^2)^{\mathrm{h}(w) - \mathrm{h}_{\mathcal{A}_k}} \mathbb{E}[(g(X_{\leq w_1}, X_{\leq w_2}))^2] \text{ for } i \in [2]. \qquad (7)$$

This inequality is immediate if $X_{\leq w_1}$ and $X_{\leq w_2}$ are independent, which is not the case in the broadcasting process. One of the main technical challenges in the proof is to show that the inequality holds when $X_{\leq w_1}$ and $X_{\leq w_2}$ are conditionally independent given $X_w$ by the Markov Property. It turns out to impose a significant technical challenge when some entries of $M$ can be 0.

Observe that (7) not only implies $g$ satisfies the desired second moment decay for **II**(2), but also **I**(1) with $w'$ to be either $w_1$ or $w_2$. Indeed, these two properties will also hold for $\tilde{g}(x) := g(x) - \mathbb{E}g(X)$, the normalized $g$ with mean 0, due to $(\mathbb{E}g(X))^2$ is negligible comparing to $\mathbb{E}g(X)^2$.

**IV. A closer look at the decomposition:**

Consider any $\mathcal{A}_{k+1}$-polynomial $f$ of variable $x_S$ for $S \subseteq \{v \in L : v \leq \rho'\}$ for some vertex $\rho'$. Before we proceed to the discussion of the decomposition, we remark that one cannot simply express $f$ in the form of $\tilde{g}$ described above with any pivot vertex $w$.

Let us give an example to illustrate why: Consider two functions $f_1, f_2$, where $f_i$ is a function of $x_{S'}$ with $S' \subseteq \{v \in L : v \leq \rho'_i\}$ and $S' \in \mathcal{A}_{k+1} \backslash \mathcal{A}_k$. Let $f = f_1 + f_2$. Then one can justify that $f$ cannot be expressed in the form of $\tilde{g}$ with any pivot $w$ by using the property "if we fix $x_{\leq w_2}$, then $x_{x_{\leq w_1}} \mapsto g(x_{\leq w_1}, x_{\leq w_2})$ is a $\mathcal{A}_k$-polynomial" discucssed in **III** to derive a contradiction.

The way we decompose $f$ is to express it as a sum of functions of the form $\tilde{g}(x)$ in **III**. While the actual decomposition requires a bit more adjustment, it follows from the idea to decompose $f$ in the form

$$f = \sum_{w \in \mathcal{I}} f_w,$$

where the index set $\mathcal{I}$ is the set of vertices of $T$ with height slightly greater than $\mathrm{h}_{\mathcal{A}_k}$ (to ensure correlation decay). Each $f_w$ is a function with variable $x_S$ for $S \subseteq \{u \in L : u \leq w\}$ satisfies the description of $\tilde{g}(x)$ in **III**.

Observe that this decomposition satisfies the description in **II**:

- The size of $\mathcal{I}$ is bounded by $2^\ell + 2^{\ell-1} + \cdots 1 \leq 2^{\ell+1}$.

- The second moment decay property of $f_w$ follows from **III**.

- Finally, consider $u, v \in \mathcal{I}$. If $v < u$, say $v \leq u_1$, then $f_u$ and $f_v$ satisies the covariance bound condition stated in **I** with $f_\alpha = f_u$, $f_\beta = f_2$, and $w' = u_2$. The case when $u < v$ is similar. When $u$ and $v$ are not comparable, then following the Markov Property, $\mathbb{E}[f_u(X)f_v(X)] = \mathbb{E}\mathbb{E}[f_u(X) \mid X_{\not\leq w}]\mathbb{E}[f_v(X) \mid X_{\not\leq w}]$ with $w$ being the nearest common ancestor of $u$ and $v$ and $X_{\not\leq w} = (X_{w'})_{w' \not\leq w}$, which makes it easier to show the covariance bound. (See Figure 4 for an illustration.)

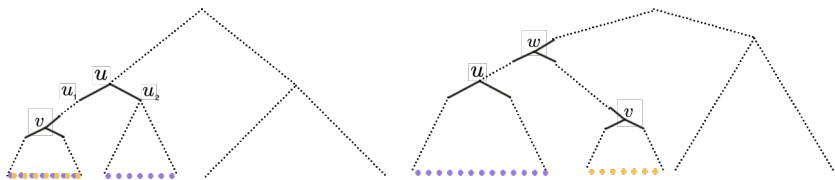

Figure 4: In both figures, the purple dots represent the corresponding input variables for $f_u$, and the yellow dots represent the corresponding input variables for $f_v$. In the left figure, we have $v \leq u_1 < u$. In the right figure, we have $u$ and $v$ are incomparable and $w$ is the nearest common ancestor.

With this desirable decomposition, one could try to apply some argument similar to the degree-1 case to show the second moment decay (6) for mean 0 $\mathcal{A}_{k+1}$-polynomials with a slightly bigger penalty constant $\mathrm{h}_{\mathcal{A}_{k+1}}$ than $\mathrm{h}_{\mathcal{A}_k}$.

**V. Overview on the induction** Given the decomposition of $f$ as described in **IV**, together with the second moment assumption (6) on $\mathcal{A}_k$-polynomials described in **III**, the proof of the main theorem proceeds by induction. The goal is to show that the penalty constant $\mathrm{h}_{\mathcal{A}_k}$ associated with $\mathcal{A}_k$, which appeared (6), satisfies the following recursive inequality:

$$\mathrm{h}_{\mathcal{A}_{k+1}} \leq \mathrm{h}_{\mathcal{A}_k} + C, \text{ and } \mathrm{h}_{\mathcal{A}_1} \leq C$$

for some constant $C$ depending on $M$ and $d\lambda^2$. If true, by taking $k$ to be proportional to $\ell/C$, then the theorem follows. The proof of the theorem requires a careful analysis of the covariance and variance decay to demonstrate that it resembles the behavior observed in the degree-1 case, in order to capture the Kesten-Stigum bound.

**Comparison to other work in low-degree polynomials** While some high-level ideas align with previous work in low degree polynomials mentioned fore, our approach focuses on establishing low-degree lower bounds in a setting where direct comparisons are challenging due to structural differences. Specifically, our work addresses the broadcasting process on trees, where the underlying structures are highly correlated and do not naturally lend themselves to a product measure representation, presenting unique technical challenges not encountered in more independent setups.

## Acknowledgments

Han Huang was supported by Elchanan Mossel's Vannevar Bush Faculty Fellowship ONR-N00014-20-1-2826 and by Elchanan Mossel's Simons Investigator award (622132). Elchanan Mossel was partially supported by Bush Faculty Fellowship ONR-N00014-20-1-2826, Simons Investigator award (622132), ARO MURI W911NF1910217 and NSF award CCF 1918421.

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

## Appendix

**Overview**

- In Section A, we give additional notations and basic tools.
- In Section B, we formulate the main theorem we want to prove as an induction statement.
- In Section C, we discuss the case for degree 1 polynomial, which prove the base case of the induction in the theorem, and the results for degree 1 polynomial will be used in the inductive step.
- In Section D, we give a procedure to decompose $\mathcal{B}$-polynomial $f$ for a given collection $\mathcal{B}$.
- In Section E and F, we derive the proof of Theorem B.6, the inductive step for proving Theorem B.1.
- In Section G and H, we derive the main result in the general case.
- In Section I, we provide a proof of Proposition C.3, which is one technical obstacle for getting our main result from Theorem B.1 to the general setting (Theorem 1.6). It is postponed to this section due to the proof is essentially a result about Markov Chain.
- In Section J, we provided some standard result for decay of Markov Chain.

## A  Additional Notations and Tools

Let us begin with the proof of Remark 1.15 which shows $\mathcal{A}_k$ contains all non-empty subsets of $L$ of size $\leq k$:

*Proof of Remark 1.15.* The proof will be carried out by induction on $k$. The base case with $k = 1$ follows from the definition $\mathcal{A}_1 := \{\{v\} \ : \ v \in L\}$. Suppose the claim holds up to some positive integer $k$. Let $\emptyset \neq S \subseteq L$ of size $|S| \leq k + 1$. If $|S| \leq k$, then $S \in \mathcal{A}_k \subseteq \mathcal{B}(\mathcal{A}_k) = \mathcal{A}_{k+1}$. In the case $|S| = k + 1 \geq 2$, we first observe that $\rho(S)$ is not a leave. Consider the branch decomposition of $S$ (See Definition 1.9):

$$S = \sqcup_{i \in I(S)} S_i.$$

Because $|I(S)| > 1$, for each $i \in I(S)$ we have $|S_i| < |S| = k + 1$. Therefore, $S_i \in \mathcal{A}_k$ for $i \in I(S)$, which in turn implies $S \in \mathcal{B}(\mathcal{A}_k) = \mathcal{A}_{k+1}$. Therefore, the claim follows.

Now, to show the second statement. For every node $w$, let $S_w = \{v \in L : v \leq w\}$. Observe that if $w$ is $k - 1$ layers above $L$, then $S_w$ are the $(k-1)$th descendants of $w$, which has size $|S_w| = d^{k-1}$.

We **claim** that for $w$ which are $k - 1$ layers above $L$, then $S_w \in \mathcal{A}_k$. Let us prove the claim by induction. First, it is clear that for $w \in L$, $S_w = \{w\} \in \mathcal{A}_1$. Suppose the statement holds up to $k$. Take any $w$ which is $k$ layer above $L$. Then, the branch decomposition of $S_w = \sqcup_{i \in [d]} S_{w_i}$. With each $w_i$ is $k - 1$ layer above $L$, we have $S_{w_i} \in \mathcal{A}_k$. Hence, $S_w \in \mathcal{A}_{k+1}$ due to $\mathcal{A}_{k+1} = \mathcal{B}(\mathcal{A}_k)$.

$\square$

### A.1  Additional notations

For any positive integer $n$, let $[n]$ denote the set of positive integers from 1 to $n$, inclusive: $[n] = 1, 2, \ldots, n$. For integers $a$ and $b$ where $a < b$, let $[a, b] = \{a, a + 1, \ldots, b\}$.

**Additional Notation for Trees**

For $u \in T$, we define

$$L_k(u)$$

to be the set of $k$th descendants of $u$. For brevity, let $L_k := L_k(\rho)$. Further, for $h \in [0, \mathrm{h}(u)]$, let

$$D_h(u) = \{v \in T \ : \ v \leq u \text{ and } \mathrm{h}(v) = h\},$$

namely the set of descendants of $u$ which has height $h$. Observe that

$$D_k(u) = L_{\mathrm{h}(u)-k}(u).$$

Last, let

$$T_u$$

be the induced subgraph of $T$ with vertex set $\{v \in T : v \leq u\}$ and $L_u = T_u \cap L$. It is worth noting that $T_u$ can be seen as a rooted tree with root $u$ and $\mathrm{h}(u)$ layers, and the $\mathrm{h}(u)$th layer is $L_u$.

**Additional Notation for collection $\mathcal{A} \subseteq \mathbf{2}^L \setminus \{\emptyset\}$**

**Definition A.1.** *For a given collection of of subsets $\mathcal{A} \subseteq \mathbf{2}^L \setminus \{\emptyset\}$, we define the following subcollections: For each $u \in T$, let $\mathcal{A}_u := \{S \in \mathcal{A} : \rho(S) = u\}$, $\mathcal{A}_{\leq u} := \{S \in \mathcal{A} : \rho(S) \leq u\}$, and $\mathcal{A}_{<u} := \{S \in \mathcal{A} : \rho(S) < u\}$.*

**Notation for Conditional Expectation**

**Definition A.2.** *Recall that an antichain $U \subseteq T$ is a collection of vertices such that no two vertices in $U$ are comparable under the $\leq$ relation. For $x \in [q]^T$, we can decompose $x$ in the form*

$$x = (x_{<U}, x_U, x_{\not\leq U}),$$

*where*

$$x_{<U} = (x_v : \exists u \in U \text{ s.t. } v \leq u), \text{ and } x_{\not\leq U} = (x_v : \forall u \in U, v \not\leq u).$$

*(See Figure 5 for an illustration.)*

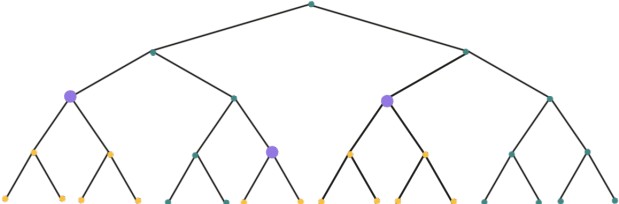

Figure 5: In this figure, the purple dots represent the set $U$. The yellow dots represent the vertices corresponding to the variables $x_{<U}$, and the green dots represent the vertices corresponding to the variables $x_{\not\leq U}$.

**Definition A.3.** *[Conditional Expectation]*

*For each antichain $U \subseteq T$ and $f : [q]^T \to \mathbb{R}$ let*

$$(\mathbb{E}_U f)(x) := \mathbb{E}\left[ f(X) \,\middle|\, X_U = x_U, \, X_{\not\leq U} = x_{\not\leq U} \right].$$

To rephrase it, the function $(\mathbb{E}_U f)(x)$ represents the expected value of $f(X)$ condition on $X_v = x_v$ for all vertices $v$ that are not descendents of any $u \in U$. In the case when $U = \{u\}$, we will abuse the notation and denote $\mathbb{E}_u$ as $\mathbb{E}_{\{u\}}$ for $u \in T$. Futher, for any $\mathrm{h} \in [0, \ell]$, we set

$$(\mathbb{E}_{\mathrm{h}} f)(x) := \mathbb{E}\left[ f(X) \,\middle|\, \forall v \in T \text{ with } \mathrm{h}(v) \geq \mathrm{h}, \, X_v = x_v \right].$$

**Remark A.4** (Conditional Expectation and the Markov Property)**.** Suppose $f$ is a function of $x_L$ and $U$ an antichain of $T$. Let

$$\tilde{L} = \{v \in L : \exists u \in U \text{ s.t. } v \leq u\} \text{ and } L' = L \setminus \tilde{L}.$$

We remark that by the Markov Property,

$$\mathbb{E}_U f(x) = \mathbb{E}_U f(x_U, x_{L'})$$

is a function of $x_U$ and $x_{L'}$. To see that, by the Markov Property, $x_{<U}$ and $x_{\not\leq U}$ are independent conditioned on $x_U$. Thus,

$$
\begin{aligned}
(\mathbb{E}_U f)(x_U, x_{\not\leq U}) =& \mathbb{E}\left[ f(X_{\tilde{L}}, X_{L'}) \,\middle|\, X_U = x_U, X_{\not\leq U} = x_{\not\leq U} \right] \\
=& \mathbb{E}\left[ f(X_{\tilde{L}}, x_{L'}) \,\middle|\, X_U = x_U \right],
\end{aligned}
$$

which implies $\mathbb{E}_U f$ is a function of $x_U$ and $x_{L'}$. (See Figure 6 for an illustration.)

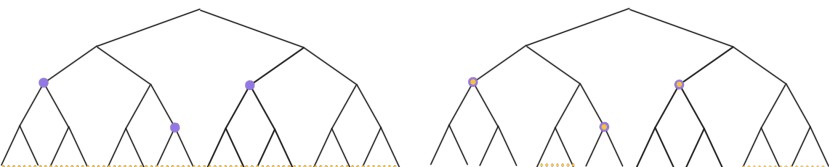

Figure 6: The purple dots represent the set $U$. In the left figure, the yellow dots represent the vertices corresponding to the variables of $f$. In the right figure, the yellow dots represent the vertices corresponding to the variable of $\mathbb{E}_U f$.

## A.2 Basis Functions on $[q]^L$ and some decay properties

The following lemma is a well-known statement from Markov-Chain. Let us formulate it using the Broadcasting Process. (For completeness, we include a proof of this lemma in Section J.)

**Lemma A.5.** *Suppose $M$ is irreducible and aperiodic, then there exists $C = C(M) > 1$ so that the following hold: For any $u \in T$ and $k \in \mathbb{N}$ so that $\mathfrak{p}^k(u) = \underbrace{\mathfrak{p} \circ \mathfrak{p} \circ \ldots \mathfrak{p}}_{k \text{ times}}(u)$, the kth ancester of $u$, exists. For every function $a$ with variable $x_u$,*

$$\mathrm{Var}\Big[(\mathbb{E}_{\mathfrak{p}^k(u)}a)(X_{\mathfrak{p}^k(u)})\Big] \leq Ck^{2q}\lambda^{2k}\mathrm{Var}\big[a(X_u)\big] \tag{8}$$

*and*

$$C^{-1}\left(\max_{\theta \in [q]}\big|a(\theta) - \mathbb{E}a(X_u)\big|\right)^2 \leq \mathrm{Var}_{Y \sim \pi}a(Y) \leq C\left(\max_{\theta \in [q]}\big|a(\theta) - \mathbb{E}a(X_u)\big|\right)^2. \tag{9}$$

*And from the above two inequalities, adjusting the constant $C$ if necessary, we also have*

$$\max_{\theta \in [q]}\big|(\mathbb{E}_{\mathfrak{p}^k(u)}a)(\theta) - \mathbb{E}a(X_u)\big| \leq Ck^q\lambda^k\max_{\theta \in [q]}\big|a(\theta) - \mathbb{E}a(X_u)\big|. \tag{10}$$

In this paper, we will fix a basis for the space of functions from $[q]$ to $\mathbb{R}$ for a Markov chain $M$.

**Definition A.6.** *For a given $q \times q$ ergodic and irreducible transition matrix $M$, we **fix** a basis $\{\phi_i\}_{i \in [0,q-1]}$ for the space of functions from $[q]$ to $\mathbb{R}$ such that $\phi_0$ is the constant function $1$ and $\phi_i$ for $i \in [q-1]$ are functions such that*

$$\mathbb{E}_{Y \sim \pi}\phi_i(Y) = 0 \quad and \quad \mathbb{E}_{Y \sim \pi}\phi_i^2(Y) = 1.$$

**Definition A.7.** *Suppose a given $\ell$ layer rooted tree $T$ and $q \times q$ transition matrix described in Lemma A.5 have been given. For $\sigma \in [0, q-1]^L$, let*

$$S(\sigma) = \{v : \sigma(v) \neq 0\} \subseteq L,$$

*the set of vertices in $L$ in which $\sigma$ is non-zero. Let $|\sigma| = |S(\sigma)|$. When $|\sigma| \geq 1$, we define*

$$\rho(\sigma) = \rho(S(\sigma)).$$

*And when $|\sigma| \geq 2$, we define*

$$I(\sigma) := I(S(\sigma)).$$

*Further, let*

$$\phi_\sigma(x) := \prod_{v \in L}\phi_{\sigma(v)}(x_v) = \prod_{v \in S(\sigma)}\phi_{\sigma(v)}(x_v).$$

*and*

$$\tilde{\phi}_\sigma(x) := \phi_\sigma(x) - \mathbb{E}\phi_\sigma(X).$$

**Remark A.8.** We remark that $\phi_\sigma(x)$ is a function with variables $x_{S(\sigma)}$.

The fact that $\phi_0, \phi_1, \ldots, \phi_{q-1}$ forms a basis implies that:

**Fact A.9.** *Every function $f : [q]^U \to \mathbb{R}$ can be expressed uniquely in the form*

$$f(x) = \sum_{\sigma \,:\, S(\sigma) \subseteq U} c_\sigma \phi_\sigma(x). \tag{11}$$

**Remark A.10.** With the above representation, the **Efron-Stein degree** of function $f$ equals to the largest magnitude of $|\sigma|$ among those $\sigma$ such that $c_\sigma \neq 0$.

**Definition A.11.** *Given a tree $T$ and a $q \times q$ ergodic transition matrix $M$, let $\{\phi_i\}_{i \in [q-1]}$ be the functions described in Lemma A.5. For a collection of subsets $\mathcal{A} \subseteq \mathbf{2}^L \backslash \{\emptyset\}$, let*

$$\mathcal{F}(\mathcal{A}) := \{\sigma \in [0, q-1]^L \,:\, S(\sigma) \in \mathcal{A}\}. \tag{12}$$

*For any $\sigma \in \mathcal{F}(\mathcal{A})$ with $|\sigma| > 1$, let*

$$\psi_\sigma(x) := \prod_{i \in I(\sigma)} \tilde{\phi}_{P_i \sigma}(x), \tag{13}$$

*where, for each $i \in I(\sigma)$, $P_i \sigma$ is the restriction of $\sigma$ to $S(\sigma)_i$:*

$$(P_i \sigma)(v) = \sigma(v) \mathbf{1}(v \in S(\sigma)_i) \text{ for } v \in L.$$

# B   The overall inductive argument

Here we present the version of the theorem with additional assumption on the transition matrix $M$ that

$$c_M := \min_{i,j \in [q]} M_{ij} > 0. \tag{14}$$

**Theorem B.1.** *Given the rooted tree $T$ and the transition matrix $M$ described in Theorem 1.6, and under the additional assumption that $c_M = \min_{i,j \in [q]} M_{ij} > 0$, there exists $c > 0$ dependent on $M$ and $d\lambda^2$ (and implicitly on $c_M$ as well) so that the following holds: For any function $f$ of the leaves with **fractal capacity** $\leq c \frac{\ell}{\log(dR)}$,*

$$\mathrm{Var}(\mathbb{E}\big[f(X_L) \,\big|\, X_\rho\big]) \leq (\max\{d\lambda^2, \lambda\})^{\ell/4} \mathrm{Var}(f(X_L)).$$

We will first derive the version mentioned above, as it substantially reduces the technical complexity without compromising the structural integrity of the proof in the general setting where $c_M$ might be 0.

The proof of Theorem 1.6 will be carried out by induction on $\mathcal{A}_k$-polynomials. Let us introduce the necessary notations to outline this induction process.

**Definition B.2.** *Let $\varepsilon > 0$ be the constant such that*

$$\max\{d\lambda^2, \lambda\} = \exp(-1.1\varepsilon).$$

The constant $\varepsilon$ is introduced to improve the readability of the paper. Intuitively, we aim to define $d\lambda^2 = \exp(-\varepsilon)$, but we relax this definition slightly so that inequalities like the following hold when $\ell$ is sufficiently large:

$$\mathrm{poly}(\ell)(d\lambda^2)^\ell \leq \exp(-\varepsilon\ell).$$

**Assumption B.3.** We say that $\mathcal{A}$ satisfies assumption B.3 with parameters $(\mathrm{h}^*, c^*)$ where $\mathrm{h}^* > 0$ and $0 < c^* < 1$, if

$$\mathcal{A}_1 \subseteq \mathcal{A} \subseteq \mathbf{2}^L \backslash \{\emptyset\}$$

is **closed under decomposition**, and morever,

1. For any $v \in T$ with $\mathrm{h}(v) \geq \mathrm{h}^*$ and a $\mathcal{A}_{\leq v}$-polynomial $f$,

$$\mathrm{Var}\big[(\mathbb{E}_v f)(X)\big] \leq \exp\big(-\varepsilon(\mathrm{h}(v) - \mathrm{h}^*)\big)\mathrm{Var}\big[f(X)\big]. \tag{15}$$

2. For any $v \in T \setminus \{\rho\}$ with $\mathrm{h}(v) \geq \mathrm{h}^*$ and a $\mathcal{A}_{\leq v}$-polynomial $f$ with $\mathbb{E}f(X) = 0$,

$$c^*\mathbb{E}\left[(\mathbb{E}_v f^2)(X_v)\right] \leq \mathbb{E}[(\mathbb{E}_v f^2)(X_v) \mid X_{\mathfrak{p}(v)} = \theta] \leq \frac{1}{c^*}\mathbb{E}\left[(\mathbb{E}_v f^2)(X_v)\right], \quad (16)$$

for all $\theta \in [q]$.

The inequality (15) bears a resemblance to the inequality we aim to prove in Theorem B.1. The second inequality, (16), will later be seen as a crucial step proving the inductive phase of our proof. Indeed, in the case where $c_M > 0$, the condition (16) can be easily satisfied by appropriately choosing $c^*$.

**Lemma B.4.** *For any given $\mathcal{A}_1 \subseteq \mathcal{A} \subseteq 2^L \setminus \{\emptyset\}$ which is closed under decomposition. If it satisfies (15) with a given parameter $\mathrm{h}^*$ and $c_M > 0$, then $\mathcal{A}$ satisfies Assumption B.3 with parameter $\mathrm{h}^*$ and*

$$c^* := \min\left\{c_M, \frac{1}{\min_j \pi(j)}\right\} > 0.$$

*In other words, we can choose $c^*$ with* **no dependence on either** $\mathrm{h}^*$ **or** $\mathcal{A}$*.*

*Proof.* Consider an arbitrary function $f$ with variables $(x_u : u \leq v)$ for some $v \in T \setminus \{\rho\}$.

Let $g(x) := (\mathbb{E}_v f^2)(x)$. By the Markov Property, $(\mathbb{E}_v f^2)(x)$ is a function of $x_v$, which in turn implies $g(x) = g(x_v)$. Now, for any $\theta \in [q]$, fix an index $j_0 \in [q]$ such that $g(j_0) \geq \mathbb{E}g(X_v)$. Relying on $g$ is a non-negative function,

$$\mathbb{E}[g(X_v) \mid X_{\mathfrak{p}(v)} = \theta] = \sum_{j \in [q]} M_{\theta j} g(j) \geq M_{\theta j_0} g(j_0) \geq c_M \mathbb{E}g(X_j).$$

By unraveling the definition of $g$, we can satisfy the first inequality of (16) as long as $c^* < c_M$. The proof for the second inequality follows a similar logic, using the condition $c^* \leq \frac{1}{\min_j \pi(j)}$ and the trival inequality $\max_{i,j} M_{ij} \leq 1$. $\qquad \square$

Given this notation, the proof of Theorem B.1 proceeds by induction, with the base case and inductive articulated in the subsequent two statements.

**Proposition B.5.** *Given the rooted tree $T$ and the transition matrix $M$ described in Theorem 1.6, and under the additional assumption that $c_M = \min_{i,j \in [q]} M_{ij} > 0$. There exists $C = C(M, \varepsilon) \geq 1$ so that the following holds:*

*Fix $\rho' \in T$ and $0 \leq m \leq \mathrm{h}(\rho')$, if $f(x)$ is a degree 1 polynomials of variables $(x_v : v \in D_m(\rho'))$, then*

$$\mathrm{Var}\left[(\mathbb{E}_{\rho'} f)(X)\right] \leq \exp\left(-\varepsilon\big(\mathrm{h}(\rho') - m - C(\log(R) + 1)\big)\right)\mathrm{Var}\left[f(X)\right]. \quad (17)$$

**Theorem B.6.** *Given the rooted tree $T$ and the transition matrix $M$ described in Theorem 1.6, and under the additional assumption that $c_M = \min_{i,j \in [q]} M_{ij} > 0$. Suppose $\mathcal{A}$ is a collection of subsets satisfying Assumption B.3 with parameters $(\mathrm{h}^*, c^*)$. Then, there exists $C = C(M, d, c^*) \geq 1$ such that $\mathcal{B} = \mathcal{B}(\mathcal{A})$ satisfies Assumption B.3 with parameters $\big(\mathrm{h}^* + C(\log(R) + 1), c^*\big)$.*

Let us derive the proof of Theorem B.1 based on the above two statements.

*Proof of Theorem B.1.* We apply Proposition B.5 and Lemma B.4 to get $\mathcal{A}_1$ satisfies Assumption B.3 with parameter $\mathrm{h}^* = C_{B.5}(\log(R) + 1)$, where $C_{B.5} = C(M, d)$ is the constant introduced in the Proposition and

$$c^* = \min\left\{c_M, \frac{1}{\min_j \pi(j)}\right\} > 0.$$

Then, by applying Theorem B.6 inductively on the chain $\mathcal{A}_k$, we can conclude that $\mathcal{A}_k$ satisfies Assumption B.3 with parameter $\mathrm{h}^* = C(\log(R) + 1)k$ and the same $c^*$ described above, provided that $C = C(M, d, c^*)$ is the maximum of the constants $C$ described in Proposition B.5 and Theorem B.6. In other words, for any $\mathcal{A}_k$-polynomial $f$,

$$\mathrm{Var}\left[(\mathbb{E}_\rho f)(X)\right] \leq \exp\left(-\varepsilon(\ell - C(\log(R) + 1)k)\right)\mathrm{Var}\left[f(X)\right].$$

The theorem follows by choosing $k = \frac{1}{2C(\log(R)+1)}\ell$.

$\qquad \square$

# C   Variance Decomposition and Variance Estimate for degree 1 polynomials

To describe the goal of this section, let us begin with the variance decomposition of degree 1 polynomials in a slight generalized form. Essentially, the following statement is a direct consequence of the conditional variance formula. We will state the main results first and provide the proof later in this section.

**Lemma C.1.** *Fix $\rho' \in T$ and $0 \leq k \leq \mathrm{h}(\rho')$, consider a function $g : [q]^T \to \mathbb{R}$ of the form*

$$g(x) = \sum_{v \in D_k(\rho')} g_v(x) \quad where \quad g_v(x) = g_v(x_{\leq v}).$$

*For $w \in T_{\rho'} \backslash \{\rho'\}$ with $\mathrm{h}(w) \geq k+1$, let*

$$g_w(x) := \sum_{v \in D_k(w)} g_v(x).$$

*Then,*

$$\mathrm{Var}[g(X)] = \mathrm{Var}\big[(\mathbb{E}_{\rho'}g)(X_{\rho'})\big] + \sum_{w \in T_{\rho'} \backslash \{\rho'\}: \mathrm{h}(w) \geq k} \mathbb{E}\mathrm{Var}\big[(\mathbb{E}_w g_w)(X_w) \,\big|\, X_{\mathfrak{p}(w)}\big]$$

$$+ \sum_{v \in D_k(\rho')} \mathbb{E}\mathrm{Var}\big[g_v(X) \,\big|\, X_v\big].$$

Our goal is to show that when $d\lambda^2 < 1$, $\mathrm{Var}[g(X)]$ is of the same order as $\sum_{v \in D_k(\rho')} \mathrm{Var}[g_v(X)]$, based on the above lemma.

**Lemma C.2.** *Suppose the transition matrix $M$ satisfies $d\lambda^2 < 1$ and the tree $T$ has growth factor $R$. Then, there exists a constant $C = C(M, \varepsilon) \geq 1$ so that the following holds. Let $\rho' \in T$, $l' := \mathrm{h}(\rho')$, and $k \in [0, l']$. Consider a function of the form $g(x) = \sum_{v \in D_k(\rho')} g_v(x_v)$. Then,*

$$\mathrm{Var}[g(X)] \leq CR \sum_{v \in D_k(\rho')} \mathrm{Var}[g_v(X_v)]. \tag{18}$$

The opposite bound does not depend on $d\lambda^2 \leq 1$. However, the proof in the general case where $c_M = 0$ is not straight-forward. We state it in full generality but will defer the general proof and prove it here in the simpler case where $c_M > 0$.

**Proposition C.3.** *There exists a constant $C = C(M, d) \geq 1$ so that the following holds. Let $\rho' \in T$, and $k \in [0, \mathrm{h}(\rho')]$. For any degree-1 function $g$ with variables $(x_v : v \in D_k(\rho'))$. There exists functions $g_v(x) = g_v(x_v)$ for $v \in D_k(\rho')$ so that the following holds:*

1. *$g(X) = \sum_{v \in D_k(\rho')} g_v(X_u)$ almost surely. (They may not agree as functions from $[q]^T$ to $\mathbb{R}$.)*

2. *For any $u \in T_{\rho'}$ with $\mathrm{h}(u) \geq k$,*

$$\sum_{v \in D_k(u)} \mathrm{Var}[g_v(X_v)] \leq CR^3 \mathrm{Var}\Big[\sum_{v \in D_k(u)} g_v(X_v)\Big]. \tag{19}$$

   *In particular, taking $u = \rho'$ we have*

$$\sum_{v \in D_k(\rho')} \mathrm{Var}[g_v(X_v)] \leq CR^3 \mathrm{Var}[g(X)]. \tag{20}$$

We postpone the proof of Proposition in full generality in Appendix I, due to the technical complexity of the proof and the fact that the proof is about properties of a Markov Chain. Instead, a statement of the proposition and its proof in the case where $c_M > 0$ is provided in this section.

Now, the purpose of this section is twofold.

- First, it is the derivation of the variance related estimates: Lemma C.1, Lemma C.2, and Proposition C.3 with the additional assumption that $c_M > 0$. Additionally, we summarise the estimates into a single statements, as stated in Lemma C.7.
- Second, it is the derivation of the base case of the induction, Proposition B.5.

## C.1 Variance Decomposition and Estimates

Before we proceed to the proof of Lemma C.1, let us remark on the following consequence of the lemma.

**Remark C.4.** For any $g$ described in Lemma C.1, if we define $h(x) := (\mathbb{E}_k g)(x) = \sum_{v \in D_k(\rho')} (\mathbb{E}_v g_v)(x_v)$ and $h_v(x) := (\mathbb{E}_v g_v)(x_v)$, then by applying the lemma to both $g$ and $h$, we conclude that

$$\mathrm{Var}[g(X)] = \mathrm{Var}[(\mathbb{E}_k g)(X)] + \sum_{v \in D_k(\rho')} \mathbb{E}\mathrm{Var}\big[g_v(X) \,\big|\, X_v\big].$$

*Proof of Lemma C.1.* First, for $v \in T_{\rho'} \backslash \{\rho'\}$ with $\mathrm{h}(v) \geq k$,

$$\tilde{g}_v(x) := g_v(x) - \mathbb{E}g_v(X).$$

Notice the following holds:

$$\tilde{g}(x) := g(x) - \mathbb{E}g(X) = \sum_{v \in D_k(\rho')} \tilde{g}_v(x).$$

Let us start decomposing the variance of $g$.

$$\mathrm{Var}[g(X)] = \mathrm{Var}[\tilde{g}(X)] = \sum_{v,v' \in D_k(\rho')} \mathbb{E}\big[\tilde{g}_v(X)\tilde{g}_{v'}(X)\big]$$

$$= \sum_{w \in T_{\rho'}:\mathrm{h}(w)>k} \sum_{v,v' \in D_k(\rho'):\rho(v,v')=w} \mathbb{E}\big[\tilde{g}_v(X)\tilde{g}_{v'}(X)\big] + \sum_{v \in D_k(\rho')} \mathbb{E}\big[\tilde{g}_v^2(X)\big]$$

By the Markov Property and rearrangement of the terms, for each $w \in T_{\rho'}$ with $\mathrm{h}(w) > k$,

$$\sum_{v,v' \in D_k(\rho'):\rho(v,v')=w} \mathbb{E}\big[\tilde{g}_v(X)\tilde{g}_{v'}(X)\big]$$

$$= \sum_{v,v' \in D_k(\rho'):\rho(v,v')=w} \mathbb{E}\big[(\mathbb{E}_w\tilde{g}_v)(X)(\mathbb{E}_w\tilde{g}_{v'})(X)\big]$$

$$= \sum_{v,v' \in D_k(w)} \mathbb{E}\big[(\mathbb{E}_w\tilde{g}_v)(X)(\mathbb{E}_w\tilde{g}_{v'})(X)\big] - \sum_{v,v' \in D_k(w):\rho(v,v')<w} \mathbb{E}\big[(\mathbb{E}_w\tilde{g}_v)(X)(\mathbb{E}_w\tilde{g}_{v'})(X)\big]$$

$$= \mathbb{E}\big[(\mathbb{E}_w\tilde{g}_w)^2(X)\big] - \sum_{w' \in \mathfrak{c}(w)} \mathbb{E}\big[(\mathbb{E}_w\tilde{g}_{w'})^2(X)\big]$$

Hence,

$$\mathrm{Var}[g(X)]$$

$$= \sum_{w \in T_{\rho'}:\mathrm{h}(w)>k} \left( \mathbb{E}\big[(\mathbb{E}_w\tilde{g}_w)^2(X)\big] - \sum_{w' \in \mathfrak{c}(w)} \mathbb{E}\big[(\mathbb{E}_w\tilde{g}_{w'})^2(X)\big] \right) + \sum_{v \in D_k(\rho')} \mathbb{E}\big[\tilde{g}_v^2(X)\big]$$

$$= \sum_{w \in T_{\rho'}:\mathrm{h}(w)>k} \mathbb{E}\big[(\mathbb{E}_w\tilde{g}_w)^2(X)\big] - \sum_{w' \in T_{\rho'}\backslash\{\rho'\}:\mathrm{h}(w')\geq k} \mathbb{E}\big[(\mathbb{E}_{\mathfrak{p}(w')}\tilde{g}_{w'})^2(X)\big] + \sum_{v \in D_k(\rho')} \mathbb{E}\big[\tilde{g}_v^2(X)\big]$$

$$= \mathbb{E}\big[(\mathbb{E}_{\rho'}\tilde{g}_{\rho'})^2(X)\big] + \sum_{w \in T_{\rho'}\backslash\{\rho'\}:\mathrm{h}(w)>k} \mathbb{E}\big[(\mathbb{E}_w\tilde{g}_w)^2(X) - (\mathbb{E}_{\mathfrak{p}(w)}\tilde{g}_w)^2(X)\big]$$

$$+ \sum_{v \in D_k(\rho')} \Big( \mathbb{E}\big[\tilde{g}_v^2(X)\big] \underbrace{- \mathbb{E}\big[(\mathbb{E}_v\tilde{g}_v)^2(X)\big] + \mathbb{E}\big[(\mathbb{E}_v\tilde{g}_v)^2(X)\big]}_{=0} - \mathbb{E}(\mathbb{E}_{\mathfrak{p}(v)}\tilde{g}_v)^2(X) \Big)$$

$$= \mathbb{E}\big[(\mathbb{E}_{\rho'}\tilde{g}_{\rho'})^2(X)\big] + \sum_{w \in T_{\rho'}\backslash\{\rho'\}:\mathrm{h}(w)\geq k} \mathbb{E}\mathrm{Var}\big[(\mathbb{E}_w g_w)(X_w) \,\big|\, X_{\mathfrak{p}(w)}\big] + \sum_{v \in D_k(\rho')} \mathbb{E}\mathrm{Var}\big[g_v(X) \,\big|\, X_v\big]$$

$\square$

Next, let us show the proof of Lemma C.2.

*Proof of Lemma C.2.* Let $C_0 = C_0(M, d)$ denote the constant introduced in the statement of the Lemma. Its precise value will be determined along the proof. Without lose of generality, we may assume both $\mathbb{E}g(X) = 0$ and $\mathbb{E}g_v(X_v) = 0$ for $v \in D_k(\rho')$, and the variance of each function is simply its the second moment. For brevity, let $\tau_u := (\mathbb{E}(g_u(X))^2)^{1/2}$ for $u \in D_k(\rho')$ and $\ell' := \mathrm{h}(\rho')$.

By (8) from Lemma A.5, for $s \in [l' - k]$,

$$\mathbb{E}\big[(\mathbb{E}_{\mathfrak{p}^s(u)}g_u)^2(X_{\mathfrak{p}^s(u)})\big] \leq C_{A.5}s^{2q}\lambda^{2s}\tau_u^2, \tag{21}$$

where $C_{A.5} \geq 1$ is the constant introduced in the Lemma. In particular, if $\rho(u, u') = \mathfrak{p}^s(u)$ for $u, u' \in D_k(\rho')$, then

$$\begin{aligned}|\mathbb{E}g_u(X)g_{u'}(X)| =& \big|\mathbb{E}\big[(\mathbb{E}_{\mathfrak{p}^s(u)}g_u)(X_{\mathfrak{p}^s(u)})(\mathbb{E}_{\mathfrak{p}^s(u)}g_{u'})(X_{\mathfrak{p}^s(u)})\big]\big| \\ \leq& \big(\mathbb{E}\big[(\mathbb{E}_{\mathfrak{p}^s(u)}g_u)^2(X_{\mathfrak{p}^s(u)})\big]\big)^{1/2} \cdot \big(\mathbb{E}\big[(\mathbb{E}_{\mathfrak{p}^s(u)}g_{u'})^2(X_{\mathfrak{p}^s(u)})\big]\big)^{1/2} \\ \leq& C_{A.5}s^{2q}\lambda^{2s}\tau_u\tau_{u'}.\end{aligned}$$

Then,

$$\begin{aligned}\mathbb{E}(g(X))^2 \leq& \sum_{u,u' \in D_k(\rho')} |\mathbb{E}g_u(X)g_{u'}(X)| \\ =& \sum_{s \in [l'-k]} \sum_{v \in D_{k+s}(\rho')} \sum_{u,u'} C_{A.5}s^{2q}\lambda^{2s}\tau_u\tau_{u'},\end{aligned}$$

where the sum $\sum_{u,u'}$ is taken over all ordered pairs $(u, u')$ with $u, u' \in D_k(v)$ with $\rho(u, u') = v$. By relaxing the constraint of the summation we have

$$\begin{aligned}\mathbb{E}(g(X))^2 \leq& \sum_{s \in [l'-k]} \sum_{v \in D_{k+s}(\rho')} \sum_{u,u' \in D_k(v)} C_{A.5}s^{2q}\lambda^{2s}\tau_u\tau_{u'} \\ =& \sum_{s \in [l'-k]} \sum_{v \in D_{k+s}(\rho')} C_{A.5}s^{2q}\lambda^{2s}\Big(\sum_{u \in D_k(v)} \tau_u\Big)^2 \\ \leq& \sum_{s \in [l'-k]} \sum_{v \in D_{k+s}(\rho')} C_{A.5}s^{2q}\lambda^{2s}Rd^s \sum_{u \in D_k(v)} \tau_u^2 \\ =& \Big(\sum_{s \in [l'-k]} C_{A.5}s^{2q}\lambda^{2s}Rd^s\Big) \sum_{u \in D_k(\rho')} \tau_u^2.\end{aligned}$$

Next,

$$\sum_{s=1}^{\infty} C_{A.5}s^{2q}\lambda^{2s}Rd^s \leq \sum_{s=1}^{\infty} RC_{A.5}s^{2q}\exp(-1.1\varepsilon s) := C_0 R.$$

Hence, $C_0$ depends on $\varepsilon$, $q$, and $C_{A.5}$. It is a constant which is determined by $M$ and $\varepsilon$.

$\square$

Let us formulate Proposition C.3 under the additional assumption that $c_M = \min_{i,j \in [q]} M_{ij} > 0$. Indeed, in this case, the bound does not depend on $R$.

**Proposition C.5.** *Suppose the transition matrix $M$ satisfies $c_M > 0$. There exists a constant $C = C(M, \varepsilon) \geq 1$ so that the following holds: Let $\rho' \in T$, $l' := \mathrm{h}(\rho')$, and $k \in [0, l']$. For any function $g = [q]^T \to \mathbb{R}$ of the form*

$$g(x) = \sum_{v \in D_k(\rho')} g_v(x_v).$$

*The following holds: For any $u \in T_{\rho'}$ with $h(u) \geq k$,*

$$\sum_{v \in D_k(u)} \mathrm{Var}[g_v(X_v)] \leq C \mathrm{Var}\Big[\sum_{v \in D_k(u)} g_v(X_v)\Big]. \tag{22}$$

*In particular, taking $u = \rho'$ we have*

$$\sum_{v \in D_k(\rho')} \mathrm{Var}[g_v(X_v)] \leq C \mathrm{Var}[g(X)]. \tag{23}$$

The proof of the Proposition relies on the following immediate consequence of $c_M > 0$:

**Lemma C.6.** *Suppose $M$ is a $q \times q$ ergodic transition matrix with $c_M = \min_{i,j \in [q]} M_{ij} > 0$. There exists $C = C(M) \geq 1$ so that the following holds. For any $u \in T \backslash \{\rho\}$ and a function $h(x) = h(x_u)$,*

$$\mathbb{E}\mathrm{Var}[h(X_u) \mid X_{\mathfrak{p}(u)}] \geq \frac{1}{C(M)} \mathrm{Var}[h(X_u)].$$

*Proof.* Let $\theta_1 = \mathrm{argmin}_{\theta \in [q]} h(\theta)$ and $\theta_2 = \mathrm{argmax}_{\theta \in [q]} h(\theta)$. (In the case of a tie, we may choose any of the minimizers or maximizers.) First, we have

$$\mathrm{Var}[h(X_u)] \leq (h(\theta_2) - h(\theta_1))^2.$$

Next, for any $\beta \in [q]$, we have

$$\max\big\{|\mathbb{E}[h(X_u) \mid X_{\mathfrak{p}(u)} = \beta] - h(\theta_1)|, \; |\mathbb{E}[h(X_u) \mid X_{\mathfrak{p}(u)} = \beta] - h(\theta_2)|\big\} \geq \frac{1}{2}|h(\theta_2) - h(\theta_1)|.$$

Let $i \in \{1, 2\}$ be the index such that $|\mathbb{E}[h(X_u) \mid X_{\mathfrak{p}(u)} = \beta] - h(\theta_i)| \geq \frac{1}{2}|h(\theta_2) - h(\theta_1)|$, and we will use this together with $c_M > 0$ to give a lower bound on the conditional variance:

$$\mathrm{Var}[h(X_u) \mid X_{\mathfrak{p}(u)} = \beta] \geq \big(\mathbb{E}[h(X_u) \mid X_{\mathfrak{p}(u)} = \beta] - h(\theta_i)\big)^2 \mathbb{P}\{X_u = \theta_i \mid X_{\mathfrak{p}(u)} = \beta\}$$

$$\geq \frac{1}{4}(h(\theta_2) - h(\theta_1))^2 c_M.$$

Since it holds for every $\beta \in [q]$, we conclude that

$$\mathbb{E}\mathrm{Var}[h(X_u) \mid X_{\mathfrak{p}(u)}] \geq \frac{1}{4}(h(\theta_2) - h(\theta_1))^2 c_M \geq \frac{c_M}{4} \mathrm{Var}[h(X_u)].$$

$\square$

*Proof of Proposition C.5.* We adapt the notation from Lemma C.1. For $w \leq \rho'$ with $h(w) > k$, let

$$g_w(x) := \sum_{u \in D_k(w)} g_u(x).$$

Now, we apply Lemma C.1 and Lemma C.6 to get

$$\mathrm{Var}[g(X)] = \mathrm{Var}\big[(\mathbb{E}_{\rho'} g_{\rho'})(X_{\rho'})\big] + \sum_{w \in T_{\rho'} \backslash \{\rho'\}: h(w) \geq k} \mathbb{E}\mathrm{Var}\big[(\mathbb{E}_w g_w)(X_w) \mid X_{\mathfrak{p}(w)}\big]$$

$$+ \sum_{v \in D_k(\rho')} \mathbb{E}\mathrm{Var}\big[g_v(X) \mid X_v\big]$$

$$\geq \sum_{u \in D_k(\rho')} \mathbb{E}\mathrm{Var}\big[(\mathbb{E}_u g_u)(X_u) \mid X_{\mathfrak{p}(u)}\big]$$

$$\geq \frac{1}{C_{C.6}} \sum_{u \in D_k(\rho')} \mathrm{Var}[g_u(X_u)],$$

where we used the fact that all the terms are non-negative, the first inequality is obtained by looking at the second terms for the summands with $h(w) = k$ and $C_{C.6}$ is the $M$-dependent constant introduced in Lemma C.6.

$\square$

**Lemma C.7.** *Suppose $d\lambda^2 < 1$ and the growth factor is at most $R$. There exists a constant $C = C(M, d) \geq 1$ so that the following holds. Fix $\rho' \in T$ and $0 \leq m \leq \mathrm{h}(\rho')$, if $f_m(x)$ is a function in the form*

$$f_m(x) = \sum_{v \in D_m(\rho')} f_v(x_{\leq v}).$$

*with*

$$\mathbb{E} f_m(X) = 0.$$

*Then, there exists $\bar{f}_v(x_{\leq v})$ for $v \in D_m(\rho')$ such that their sum $\bar{f}_m(x) = \sum_{v \in D_m(\rho')} \bar{f}_v(x_{\leq v})$ satisfies*

$$\bar{f}_m(X) = f_m(X) \text{ almost surely,}$$

*and for $u \leq \rho'$ with $\mathrm{h}(u) \geq m$,*

$$\frac{1}{CR^3} \sum_{v \in D_m(u)} \mathbb{E}\bar{f}_v^2(X) \leq \mathbb{E}\Big( \sum_{v \in D_m(u)} \bar{f}_v(X) \Big)^2 \leq CR \sum_{v \in D_m(u)} \mathbb{E}\bar{f}_v^2(X). \tag{24}$$

The statement of the lemma using $\bar{f}_m$ and $\bar{f}_b$ covers also the case $c_M = 0$. We will prove the Lemma by using either Proposition C.3 or Proposition C.5 with the assumption $c_M > 0$. In the later case, it suffice to simply take $f_v(x_{\leq v}) = \bar{f}_v(x_{\leq v})$.

**Remark C.8.** Note that the lemma implies the following: For any $u \leq \rho'$ with $\mathrm{h}(u) \geq m$, let

$$\bar{f}_{m,u}(x) := \sum_{v \in D_m(u)} \bar{f}_v(x). \tag{25}$$

Then, for any given $m \leq k < k' \leq \mathrm{h}(\rho')$ and $u \in D_{k'}(\rho')$, we have

$$\mathbb{E}\bar{f}_{m,u}^2(X) \leq CR \sum_{v \in D_m(u)} \mathbb{E}\bar{f}_v^2(X) = CR \sum_{w \in D_k(u)} \sum_{v \in D_m(w)} \mathbb{E}\bar{f}_v^2(X) \leq C^2R^4 \sum_{w \in D_k(u)} \bar{f}_{m,w}^2(X),$$

where the first inequality follows from the second inequality of (24) and the second inequality follows from the first inequality of (24).

*Proof of Lemma C.7.* Let

$$h(x) := (\mathbb{E}_m f_m)(x) = \sum_{v \in D_m(\rho')} (\mathbb{E}_v f_v)(x_v).$$

Note that $h$ is a degree one function of the variables $(x_v : v \in D_m(\rho'))$. Thus, we could apply Proposition C.3 to show the existence of 1-variable functions $h_v(x_v)$ for $v \in D_m(\rho')$ such that

$$h(X) = \sum_{v \in D_m(\rho')} h_v(X_v) \quad \text{almost surely} \tag{26}$$

and for any $u \in T(\rho')$ with $\mathrm{h}(u) \geq m$,

$$\sum_{v \in D_m(u)} \mathrm{Var}[h_v(X_v)] \leq C_{C.3}R^3 \mathrm{Var}\Big[ \sum_{v \in D_m(u)} h_v(X_v) \Big], \tag{27}$$

where $C_{C.3} \geq 1$ is the constant introduced in Proposition C.3.

Since $\mathbb{E}h(X) = \mathbb{E}(\mathbb{E}_m f_m)(X) = 0$, we may also assume that

$$\mathbb{E}h_v(X) = 0$$

for $v \in D_m(\rho')$, as a constant shift of the functions will not affect (26) and (27). Now, consider the following functions: For $v \in D_m(\rho')$, let

$$\bar{f}_v(x_{\leq v}) = f_v(x_{\leq v}) - \mathbb{E}f_v(X_{\leq v}) + h_v(x_v)$$

and

$$\bar{f}_m(x) = \sum_{v \in D_m(\rho')} \bar{f}_v(x).$$

First, since $\bar{f}_v(x_{\leq v})$ is defined as the sum of three terms with mean 0, we have $\mathbb{E}\bar{f}_v(X_{\leq v}) = 0$.

Second,

$$\bar{f}_m(X) = \sum_{v \in D_m(\rho')} \left( f_v(X_{\leq v}) - (\mathbb{E}_v f_v)(X_v) + h_v(X_v) \right)$$

$$= f_m(X) - h(X) + \sum_{v \in D_m(\rho')} h_v(X_v)$$

$$\overset{=}{\text{a.s.}} f_m(X).$$

By Remark C.4,

$$\mathrm{Var}\Big[\sum_{v \in D_m(u)} \bar{f}_v(X)\Big] = \mathrm{Var}\Big[\Big(\mathbb{E}_m \sum_{v \in D_m(u)} \bar{f}_v\Big)(X)\Big] + \sum_{v \in D_m(u)} \mathbb{E}\mathrm{Var}[\bar{f}_v(X) \mid X_v]$$

$$= \mathrm{Var}\Big[\sum_{v \in D_m(u)} (\mathbb{E}_v \bar{f}_v)(X)\Big] + \sum_{v \in D_m(u)} \mathbb{E}\mathrm{Var}[f_v(X) - (\mathbb{E}_v f_v)(X_v) + h_v(X_v) \mid X_v]$$

$$= \mathrm{Var}\Big[\sum_{v \in D_m(u)} h_v(X)\Big] + \sum_{v \in D_m(u)} \mathbb{E}\mathrm{Var}[f_v(X) \mid X_v] \qquad (28)$$

To estimate the lower bound, we rely on own choice of $h_v$. By (27) we have

$$(28) \geq \frac{1}{C_{C.3}R^3} \sum_{v \in D_m(u)} \mathrm{Var}\big[h_v(X)\big] + \sum_{v \in D_m(u)} \mathbb{E}\mathrm{Var}[f_v(X) \mid X_v]$$

$$\geq \frac{1}{C_{C.3}R^3} \sum_{v \in D_m(u)} \Big(\mathrm{Var}\big[h_v(X)\big] + \mathbb{E}\mathrm{Var}[f_v(X) \mid X_v]\Big)$$

$$= \frac{1}{C_{C.3}R^3} \sum_{v \in D_m(u)} \Big(\mathrm{Var}\big[(\mathbb{E}_v \bar{f}_v)(X_v)\big] + \mathbb{E}\mathrm{Var}[\bar{f}_v(X) \mid X_v]\Big)$$

$$= \frac{1}{C_{C.3}R^3} \sum_{v \in D_m(u)} \mathrm{Var}[\bar{f}_v(X)].$$

As for the upper bound, we can apply Lemma C.2 and repeat the same derivation as above to get

$$(28) \leq C_{C.2}R \sum_{v \in D_m(u)} \mathrm{Var}\big[h_v(X)\big] + \sum_{v \in D_m(u)} \mathbb{E}\mathrm{Var}[f_v(X) \mid X_v]$$

$$\leq C_{C.2}R \sum_{v \in D_m(u)} \mathrm{Var}[\bar{f}_v(X)].$$

Therefore, by taking the constant $C$ stated in the lemma to be the maximum of $C_{C.3}$ and $C_{C.2}$, the proof follows.

$\square$

## C.2 Proof of the base Case of Proposition B.5

We now prove the base case of Proposition B.5:

**Lemma C.9.** *There exists a constant $C = C(M, d) \geq 1$ so that the following holds. For any degree 1 function $f$ with variables $(x_u : u \in D_k(\rho'))$ for some $\rho' \in T$ with $k \leq \mathrm{h}(\rho')$,*

$$\mathrm{Var}\big[\mathbb{E}\big[f(X) \mid X_{\rho'}\big]\big] \leq CR^4 (\mathrm{h}')^{2q} (d\lambda^2)^{\mathrm{h}'} \mathrm{Var}[f(X)]. \qquad (29)$$

*where $\mathrm{h}' = \mathrm{h}(\rho') - k$.*

*Proof.* Let $f_u$ for $u \in D_k(\rho')$ be the functions from Proposition C.3 so that

$$f(X) = \sum_{u \in D_k(\rho')} f_u(X_u) \text{ almost surely.} \qquad (30)$$

We can assume $\mathbb{E}f(X) = 0$ and $\mathbb{E}f_u(X) = 0$ for every $u \in D_k(\rho')$ without affecting (30). From Proposition C.3, we have

$$\sum_{u \in D_k(\rho')} \mathbb{E}[f_u^2(X)] \leq C_{C.3} R^3 \mathbb{E}[f^2(X)],$$

where $C_{C.3}$ denotes the constant $C$ introduced in the Proposition.

We could apply Lemma A.5 to get

$$\mathbb{E}\big[(\mathbb{E}_{\rho'}f)^2(X_{\rho'})\big] \leq |D_k(\rho')| \sum_{v \in D_k(\rho')} \big[(\mathbb{E}_{\rho'}f_v)^2(X_{\rho'})\big]$$

$$\leq |D_k(\rho')| C_{A.5} \mathrm{h}'^{2q} \lambda^{2\mathrm{h}'} \sum_{u \in D_k(\rho')} \mathbb{E}[f_u^2(X)] \leq C_{C.3} C_{A.5} R^4 (\mathrm{h}')^{2q} (d\lambda^2)^{\mathrm{h}'} \mathbb{E}[f^2(X)]$$

where $C_{A.5}$ denotes the constant $C$ stated in the Lemma. $\qquad\square$

*Proof of the Base Case Proposition B.5.* Given $\rho' \in T$ and $0 \leq m \leq \mathrm{h}(\rho')$ described in the Proposition. Let $\mathrm{h}' = \mathrm{h}(\rho') - m$. By Lemma C.9, any function $f(x) = \sum_{v \in D_m(\rho')} f_v(x_v)$ satisfies

$$\mathrm{Var}\big[(\mathbb{E}_{\rho'}f)(X)\big] \leq C_{C.9} R^4 (\mathrm{h}')^q (d\lambda^2)^{\mathrm{h}'} \mathrm{Var}[f(X)],$$

where $C_{C.9}$ denotes the $M$-dependent constant introduced in the Lemma. With

$$C_{C.9} R^4 (\mathrm{h}')^q (d\lambda^2)^{\mathrm{h}'} \leq C_{C.9} R^4 (\mathrm{h}')^q \exp(-1.1\varepsilon \mathrm{h}') \leq \exp\big(-\varepsilon(\mathrm{h}' - C_1(\log(R) + 1))\big),$$

for some $C_1 \geq 1$ which depends on $M, d$. $\qquad\square$

# D   Decomposition of Polynomials

In this section we study the representation of functions in terms of $\phi_\sigma$ and $\psi_\sigma$. Roughly speaking $\psi_\sigma$ are "more orthogonal" than the $\phi_\sigma$. More formally we will show that under appropriate conditioning expectations of $\psi_\sigma$ factorize. Thus some of the effort in the proof and particularly in this section is devoted to relating the $\phi$ and $\psi$ representations and bounding moments of such representations.

**Lemma D.1.** *Assuming $d\lambda^2 < 1$ and growth factor of $R$, there exists $C = C(M, d) \geq 1$ so that the following holds. Let $\mathcal{A}_1 \subseteq \mathcal{B} \subseteq \mathbf{2}^L \backslash \{\emptyset\}$ be a collection of subsets which is closed under decomposition. Fix a positive integer $k_1$ and $\rho' \in T$ with $l' := \mathrm{h}(\rho') > k_1$. For every function $f$ of the form*

$$f(x) = \sum_{\sigma : \sigma \neq 0 \in \mathcal{F}(\mathcal{B}_{\leq \rho'})} c_\sigma \phi_\sigma(x)$$

*with*

$$\mathbb{E}f(X) = 0,$$

*here exists a decomposition of $f$*

$$f(X) = \sum_{u \leq \rho' : \mathrm{h}(u) \geq k} \tilde{f}_u(X) \text{ almost surely,}$$

*where, for each $u \leq \rho'$ with $\mathrm{h}(u) \geq k_1$, we have a function $f_u(x) = f_u(x_{\leq u})$ and*

1. *For $u \in T_{\rho'}$ with $\mathrm{h}(u) > k_1$, $f_u(x)$ is a linear combination of $\psi_\sigma(x)$ with $\sigma \in \mathcal{F}(\mathcal{B}_u)$ and $\tilde{f}_u(x) = f_u(x) - \mathbb{E}f_u(X)$.*

2. *For $w \leq \rho'$ with $\mathrm{h}(w) \geq k_1$, we have*

$$\frac{1}{CR^3} \mathbb{E}\Big[\sum_{u \in D_{k_1}(w)} f_u^2(X)\Big] \leq \mathbb{E}\big(\sum_{u \in D_{k_1}(w)} f_u(X)\big)^2 \leq CR\mathbb{E}\Big[\sum_{v \in D_{k_1}(w)} f_u^2(X)\Big]. \quad (31)$$

*We may group the $f_u$ according to* $\mathrm{h}(u)$ *and define for* $k_1 \leq k \leq \mathrm{h}(\rho')$,

$$f_k(x) := \sum_{u \in D_k(\rho')} \tilde{f}_u(x).$$

*In other words,*

$$f(X) = \sum_{k \in [k_1, \mathrm{h}(\rho')]} f_k(X) \text{ almost surely.}$$

To prove the main lemma, let us begin by comparing $\phi_\sigma(x)$ and $\psi_\sigma(x)$ (See Definition A.11).

**Lemma D.2.** *For $\sigma \in \mathcal{F}(\mathcal{B}_u)$, $\phi_\sigma(x)$ can be expressed in the form*

$$\phi_\sigma(x) = \prod_{i \in I(\sigma)} \psi_{P_i\sigma}(x) - a_{\subset,\sigma}(x) - a_{<,\sigma}(x) - a_{c,\sigma}, \tag{32}$$

*where:*

- *$a_{\subset,\sigma}(x)$ is a linear combination of $\phi_{\sigma'}(x)$ for $\sigma' \in \mathcal{F}(\mathcal{B}_u)$ such that $I(\sigma')$ is a proper subset of $I(\sigma)$.*

- *$a_{<,\sigma}(x)$ is a linear combination of $\phi_{\sigma'}(x)$ for $\sigma' \in \mathcal{F}(\mathcal{B}_{<u})$ (recall that $\mathcal{B}_{<u} = \{S \in \mathcal{B} : \rho(S) < u\}$), and*

- *$a_{c,\sigma}$ is a constant.*

*Proof.* Fix $\sigma \in \mathcal{F}(\mathcal{B})$ and let $u = \rho(S)$ and $S = S(\sigma)$. Recall that $P_i\sigma \in [0, q-1]^T$ is the projection of $\sigma$ to $S_i$. We can also decompose the function $\phi_\sigma$ according to $\{P_i\sigma\}_{i \in I(\sigma)}$:

$$\phi_\sigma(x) = \prod_{i \in I(S)} \phi_{P_i\sigma}(x). \tag{33}$$

Before we proceed, let us note that by Lemma 1.12 and the definition of $\mathcal{B}$, we have $P_i\sigma \in \mathcal{F}(\mathcal{B}_{\leq u_i})$. Now, let us expand the function $\psi_\sigma$ according to its definition:

$$\prod_{i \in I(S)} \psi_{P_i\sigma}(x) = \prod_{i \in I(S)} \left( \phi_{P_i\sigma}(x) - \mathbb{E}\phi_{P_i\sigma}(X) \right) = \sum_{I_1, I_2} \left( \prod_{i \in I_1} \phi_{P_i\sigma}(x) \right) \left( \prod_{i \in I_2} (-\mathbb{E}\phi_{P_i\sigma}(X)) \right),$$

where the summation is taken over all possible partition $I_1 \sqcup I_2 = I(\sigma)$. Next, we can group the summands into four types based on $|I_1|$ and $|I_2|$:

**Type 1** $|I_1| = |I(\sigma)|$. The summand is simply $\phi_\sigma(x)$.

**Type 2** $2 \leq |I_1| \leq |I(\sigma)| - 1$.

Each summand is a constant multiple of $\phi_{\sigma'}(x)$ where $\sigma'$ is the projection of $\sigma$ to the indices $\sqcup_{i \in I_1} S_i$. Clearly, $S(\sigma') = \sqcup_{i \in I_1} S_i$. With $|I_1| \geq 2$, we have $\rho(\sigma') = u$. Further, each $S_i \in \mathcal{A}_{\leq u_i}$ for $i \in I(\sigma)$, it follows that $S(\sigma') \in \mathcal{B}_u$, which in turn implies $\sigma' \in \mathcal{F}(\mathcal{B}_u)$.

We denote the sum of summands of this type by $a_{\subset,\sigma}(x)$.

**Type 3** $|I_1| = 1$. Each summand is a constant multiple of $\phi_{P_i\sigma}(x)$, where $i$ is the element in $I_1$. Notice that $P_i\sigma \in \mathcal{F}(\mathcal{A}_{<u}) \subset \mathcal{F}(\mathcal{B}_{<u})$ where the inclusion follows from Lemma 1.12. We denote the sum of summands of this type as $a_{<,\sigma}(x)$.

**Type 4** $|I_1| = 0$ There is only one summand, which is a constant. We denote this constant by $a_{c,P_i\sigma}$.

With this decomposition, (32) follows. $\square$

Given the expressions for $\psi_\sigma(x)$ in terms $\phi_\sigma(x)$ and vice-versa, for any given $u \in T \backslash L$, we can convert a linear combination of $\phi_\sigma(x)$ with $\sigma \in \mathcal{F}(\mathcal{B}_u)$ to that of $\psi_\sigma(x)$ with $\sigma \in \mathcal{F}(\mathcal{B}_u)$.

**Lemma D.3.** *For $u \in T\backslash L$, consider any function of the form*

$$p_u(x) = \sum_{\sigma \in \mathcal{F}(\mathcal{B}_u)} c_\sigma \phi_\sigma(x).$$

*Then there exists a decomposition*

$$p_u(x) = \tilde{f}_u(x) + p_{<,u}(x) + c_u,$$

*where:*

- $\tilde{f}_u(x) = f_u(x) - \mathbb{E}f_u(X)$ *and* $f_u(x)$ *is a linear combination of* $\psi_\sigma(x)$ *for* $\sigma \in \mathcal{F}(\mathcal{B}_u)$,
- $p_{<,u}(x)$ *is a linear combination of* $\phi_\sigma(x)$ *with* $\sigma \in \mathcal{F}(\mathcal{B}_{<u})$, *and*
- $c_u$ *is a constant.*

*Proof.* The decomposition is constructed through recursion on the following expression:

$$r(p_u) := \operatorname{argmax}\{|I(\sigma)| \ : \ \sigma \in \mathcal{F}(\mathcal{B}_u), \ c_\sigma \neq 0\}.$$

Suppose $r(p_u) = 2$. Then the statement of simply follows from Lemma D.2.

Suppose the statement of the lemma holds whenever $r(p_u) \leq r$ for $2 \leq r < Rd$. Consider any function $p_u$ with $r(p_u) = r+1$:

$$p_u(x) = \underbrace{\sum_{\sigma \in \mathcal{F}(\mathcal{B}_u) \,:\, |I(\sigma)| \leq r+1} c_\sigma \phi_\sigma(x)}_{} = \underbrace{\sum_{\sigma \in \mathcal{F}(\mathcal{B}_u) \,:\, |I(\sigma)|=r+1} c_\sigma \phi_\sigma(x)}_{:=p_{u,r+1}(x)} + \underbrace{\sum_{\sigma \in \mathcal{F}(\mathcal{B}_u) \,:\, |I(\sigma)| \leq r} c_\sigma \phi_\sigma(x)}_{:=p_{u,\leq r}(x)}.$$

According to the decomposition of $\phi_\sigma(x)$ in Lemma D.2, let

$$f_{u,r+1}(x) := \sum_{\sigma \in \mathcal{F}(\mathcal{B}_u) \,:\, |I(\sigma)|=r+1} c_\sigma \psi_\sigma(x)$$

$$p_{*,u,r+1}(x) := \sum_{\sigma \in \mathcal{F}(\mathcal{B}_u) \,:\, |I(\sigma)|=r+1} c_\sigma a_{*,\sigma}(x)$$

where $*$ can be $\subset$, $<$, or $c$. Then,

$$p_{u,r+1}(x) = f_{u,r+1}(x) + p_{\subset,u,r+1}(x) + p_{<,u,r+1}(x) + p_{c,u,r+1}(x). \tag{34}$$

Observe that $p_{u,\leq r}(x) + p_{\subset,u,r+1}(x)$ is a linear combination of $\phi_\sigma(x)$ with $\sigma \in \mathcal{F}(\mathcal{B}_u)$ and $|I(\sigma)| \leq r$. Thus, by the inductive assumption, the summation can be expressed in the form

$$p_{u,\leq r}(x) + p_{\subset,u,r+1}(x) = \tilde{f}'_u(x) + p'_{<,u}(x) + c'_u.$$

Finally, let

$$f_u(x) = f'_u(x) + f_{u,r+1}(x),$$
$$p_{<,u}(x) = p'_{<,u}(x) + p_{<,u,r+1}(x),$$
$$c_u = c'_u + p'_{c,u} + \mathbb{E}\big[f_{u,r+1}(X)\big],$$

and we have

$$p_u(x) = f_{u,r+1}(x) + p_{<,u,r+1}(x) + p_{c,u,r+1}(x) + \tilde{f}'_u(x) + p'_{<,u}(x) + c'_u$$
$$= \tilde{f}_u(x) + p_{<,u}(x) + c_u.$$

$\square$

*Proof of Lemma D.1.* We will construct $f_u(x)$ for $u$ starting from top layer ($u = \rho'$) to bottom layer. For $k \in [k_1, l' - 1]$, when $f_u(x)$ is constructed for $u \in T_{\rho'}$ with $\mathrm{h}(u) > k$, we define

$$f_{\leq k}(x) = f(x) - \sum_{u \,:\, \mathrm{h}(u) > k+1} \tilde{f}_u(x), \tag{35}$$

where $\tilde{f}_u(x) = f_u(x) - \mathbb{E} f_u(X)$. Without lose of generality, let $f_{\leq l'}(x) = f(x)$.

Fix $k \in [k_1 + 1, l']$. For the induction step, suppose $\{f_u(x)\}_{u \in T_{\rho'} \,:\, \mathrm{h}(u) > k}$ $\{f_{\leq s}(x)\}_{s \in [k, l']}$ have been constructed such that $f_{\leq k}(x)$ can be expressed in the form

$$f_{\leq k}(x) = c' + \sum_{\sigma \in \mathcal{F}(\mathcal{B}) \,:\, k_1 < \mathrm{h}(\rho(\sigma)) \leq k} c'_\sigma \phi_\sigma(x) + \sum_{\sigma \in \mathcal{F}(\mathbf{2}^L) \,:\, \mathrm{h}(\rho(\sigma)) \leq k_1} c'_\sigma \phi_\sigma(x). \tag{36}$$

Clearly, this holds when $k = l'$.

For each $u$ with $\mathrm{h}(u) = k$, let $p_u(x) = \sum_{\sigma \in \mathcal{F}(\mathcal{B}_u)} c'_\sigma \phi_\sigma(x)$, and define $\tilde{f}_u(x), p_{<,u}(x)$, and $c_u$ according to Lemma D.3. Then,

$$
\begin{aligned}
f_{\leq k-1}(x) =\,& f_{\leq k}(x) - \sum_{u \,:\, \mathrm{h}(u) = k_1} \tilde{f}_u(x) \\
=\,& c' + \sum_{\sigma \in \mathcal{F}(\mathcal{B}) \,:\, k_1 < \mathrm{h}(\rho(\sigma)) \leq k-1} c'_\sigma \phi_\sigma(x) + \sum_{\sigma \in \mathcal{F}(\mathbf{2}^L) \,:\, \mathrm{h}(\rho(\sigma)) \leq k_1} c'_\sigma \phi_\sigma(x) \\
&+ \sum_{u \,:\, \mathrm{h}(u) = k} (p_{<,u}(x) + c_u).
\end{aligned}
$$

Recall from Lemma D.3 that $p_{<,u}(x)$ is a linear combination of $\phi_\sigma(x)$ with $\sigma \in \mathcal{F}(\mathcal{B}_{<u})$, the function $f_{\leq k-1}(x)$ satisfies (36) as well (with $k$ been replaced by $k - 1$).

Once the induction terminated at layer $k_1$, we obtain

$$f_{k_1}(x) = c + \sum_{u \in D_{k_1}(\rho')} \sum_{\sigma \in \mathcal{F}(\mathbf{2}^{L_u} \setminus \{\emptyset\})} c_\sigma \phi_\sigma(x).$$

Now, observe that for $k_1 < k \leq \mathrm{h}(\rho')$, we have $f_k(x)$ is defined as the sum of $\tilde{f}_u$ for $u \in D_k(\rho')$, which are functions of mean 0. Together with the assumption that $\mathbb{E} f(X) = 0$, we have

$$\mathbb{E} f_{k_1}(X) = \mathbb{E} f(X) - \sum_{k=k_1+1}^{\mathrm{h}(\rho')} \mathbb{E} f_k(X) = 0.$$

Notice that $f_{k_1}$ satisfies the assumption of the function stated in Lemma C.7 with $m = k_1$. By replacing $f_{k_1}$ by $\bar{f}_{k_1}$ and $f_u$ by $\bar{f}_u$ for each $u \in D_{k_1}(\rho')$, the second statement follows while the third statement of the Lemma remains true. Hence, the proof is completed. $\qquad \square$

# E    Induction Step 1: Decay of $f_u$

The goal this section and next section is to prove Theorem B.6. Let us restate the theorem here.

**Theorem.** *Given the rooted tree $T$ and the transition matrix $M$ described in Theorem 1.6, and under the additional assumption that $c_M = \min_{i,j \in [q]} M_{ij} > 0$. Suppose $\mathcal{A}$ is a collection of subsets satisfying Assumption B.3 with parameters $\mathrm{h}^*$ and $c^*$. Then, there exists $C = C(M, d, c^*) \geq 1$ such that $\mathcal{B} = \mathcal{B}(\mathcal{A})$ satisfies Assumption B.3 with parameters $\mathrm{h}^* + C(\log(R) + 1)$ and $c^*$.*

**In this and the following section**, we will fix a collection $\mathcal{A}$ that meets Assumption B.3 with some parameters $l^*$ and $c^*$. Additionally, we abbreviate

$$\mathcal{B} = \mathcal{B}(\mathcal{A}).$$

Further, we will fix $\rho' \in T$ and a function $f$ described in the Assumption B.3, and assume

$$\mathbb{E}f(X) = 0.$$

The proof is grounded in the decomposition of $f$ as described in Lemma D.1, which splits $f$ into summation of $f_k$ and subsequently into summations of $\tilde{f}_u$. Accordingly, this section is devoted to derive the variance decay properties of $f_u$ stated as Proposition E.1 below. The proposition will be used to derive variance decay properties of $f_k$, and toward the proof of Theorem B.6 in next section.

**Proposition E.1.** *There exists $C = C(M, c^*) \geq 1$ so that the following holds. For any $u \in T$, consider a function $f_u$ of the form*

$$f_u(x) = \sum_{0 \neq \sigma \in \mathcal{F}(\mathcal{B}_u)} c_\sigma \psi_\sigma(x).$$

*Then, for $\theta \in [q]$, we have the following bounds on $(\mathbb{E}_u f_u)(x)$ (recall that that by the Markov Property, $(\mathbb{E}_u f_u)(x)$ is a function of $x_u$):*

$$(\mathbb{E}_u f_u)^2(\theta) \leq \exp(-2\varepsilon(\mathrm{h}(u) - C(\log(R) + 1) - \mathrm{h}^*))(\mathbb{E}_u f_u^2)(\theta) \tag{37}$$

*and*

$$\mathbb{E}\big[(\mathbb{E}_u \tilde{f}_u)^2(X_u)\big] \leq \exp(-2\varepsilon(\mathrm{h}(u) - C(\log(R) + 1) - \mathrm{h}^*))\mathbb{E}\tilde{f}_u^2(X). \tag{38}$$

*Additionally, for any function $a(x)$ having inputs involving only $(x_v : v \in T_{u_i})$ for some $i \in [d_u]$:*

$$\mathbb{E}\big[|\tilde{f}_u(X)a(X)|\big] \leq \exp\Big(-\frac{\varepsilon}{2}(\mathrm{h}(u) - C(\log(R) + 1) - \mathrm{h}^*)\Big)(\mathbb{E}\tilde{f}_u^2(X))^{1/2}(\mathbb{E}a^2(X))^{1/2}. \tag{39}$$

**Remark E.2.** The statement of the Proposition E.1 is exactly the statement of Theorem B.6 restricted to functions all of whose non-zero $c_\sigma$ have $\rho(\sigma) = \rho'$. Thus in some sense in this section we prove the Theorem for the most complex terms. And in the next section we will control the correlations between different terms.

This is an analogue in our setting to the classical fact in Fourier analysis that high amplitude functions have sharp decay under noise.

**Remark E.3.** We remark that the proposition holds immediately whenever $|d_u| \leq 1$, since $\mathcal{B}_u = \emptyset$.

Before we proceed further, we need to decompose $f_u(x)$.

**Definition E.4.** *For $u \in T \backslash L$ and $f_u(x) = \sum_{\sigma \in \mathcal{F}(\mathcal{B}_u)} c_\sigma \psi_\sigma(x)$, let*

$$f_{u,I}(x) := \sum_{\sigma \in \mathcal{F}(\mathcal{B}_u) : I(\sigma) = I} c_\sigma \psi_\sigma(x), \quad \text{and} \tag{40}$$

$$\tilde{f}_{u,I}(x) := f_{u,I}(x) - \mathbb{E}f_{u,I}(X) \tag{41}$$

*for each $I \subseteq [d_u]$ with $|I| \geq 2$.*

Given the above definition, we have

$$f_u(x) = \sum_{I \subseteq [d_u] : |I| \geq 2} f_{u,I}(x).$$

**Proposition E.5.** *There exists $C = C(M, c^*) \geq 1$ so that the following holds. For any $u \in T \setminus L$ and $I \subseteq [d_u]$ with $|I| \geq 2$. Consider a function of the form*

$$a(x) = \sum_{\sigma \in \mathcal{F}(\mathcal{B}_u) : I(\sigma) = I} c_\sigma \psi_\sigma(x).$$

*Then, for $I' \subseteq I$, let*

$$U = T \setminus \Big( \bigcup_{i \in I'} T_{u_i} \Big),$$

*and we have*

$$\big((\mathbb{E}_U a)(x)\big)^2 \leq \exp\big(-\varepsilon|I'|(\mathrm{h}(u) - C - \mathrm{h}^*)\big)(\mathbb{E}_U a^2)(x). \tag{42}$$

Roughly speaking the proposition states that under the decay of correlation in Assumption B.3, for functions all of whose coefficient $c_\sigma$ have $S(\sigma) = I$ for some large set $I$ we get a variance decay of the form $\exp(-\epsilon |I| \mathrm{h}(u))$. For later applications the statement is more general allowing to condition on some of the subtrees. This is an analogue in our setting to the classical fact in Fourier analysis that high amplitude functions have sharp decay under noise.

*Proof.* We fix $u \in T \backslash L$ and $I \subseteq [d_u]$. Without lose of generality, we assume $I' = [s]$.

Let $C_0 = C_0(M, c^*)$ denote the constant described in the statement of the Proposition. The precise value of $C_0$ will be determined during the proof.

For brevity, we introduce some notations that are only used in this proof.

1. Decomposition of $x \in [q]^T$: Consider the representation of $x$ as

$$x = (x_u, x_0, x_1, \ldots, x_s),$$

where, $\forall k \in [s]$, $x_k := (x_v : v \leq u_k)$, and $x_0 = (x_v : v \in U \backslash \{u\})$.

Further, let

$$x_{\leq k} = (x_0, x_1, \ldots, x_k).$$

For $k \in [0, s]$,

$$a_{\leq k}(x_{\leq k}) := \mathbb{E}\Big[a(X)\,\Big|\,X_u = x_u \text{ and } X_{\leq k} = x_{\leq k}\Big].$$

Before we proceed to the proof, observe that applying Jenson's inequality on conditional expectation, we can form a chain of inequalities

$$(\mathbb{E}_U a)^2(x) = (\mathbb{E}_U a_{\leq 0}^2)(x) \leq (\mathbb{E}_U a_{\leq 1}^2)(x) \leq (\mathbb{E}_U a_{\leq 2}^2)(x) \leq \ldots \leq (\mathbb{E}_U a_{\leq s}^2)(x) = (\mathbb{E}_U a^2)(x).$$

If $\mathrm{h}(u) \leq C_0 + \mathrm{h}^*$, then the statement of the Proposition is weaker than the inequality $(\mathbb{E}_U a)^2(x) \leq (\mathbb{E}_U a^2)(x)$ stated above. So the lemma follows immediately in that case. From now on we assume

$$\mathrm{h}(u) > C_0 + \mathrm{h}^*. \tag{43}$$

We will improve each inequality in the above chain by leveraging the assumption (15).

Given the definition of $a(x)$,

$$a(x) = \sum_\sigma c_\sigma \prod_{i \in I \backslash [s]} \tilde{\phi}_{P_i \sigma}(x_0) \prod_{i \in [s]} \tilde{\phi}_{P_i \sigma}(x_i)$$

By the Markov Property, the random variables $(X_i | X_u = x_u)_{i \in [0, s]}$ are independent. This gives rise to:

$$
\begin{aligned}
a_{\leq k}(x) =& \mathbb{E}\Big[\sum_\sigma c_\sigma \prod_{i \in I \backslash [s]} \tilde{\phi}_{P_i \sigma}(X_0) \prod_{i \in [s]} \tilde{\phi}_{P_i \sigma}(X_i)\,\Big|\,X_u = x_u \text{ and } X_{\leq k} = x_{\leq k}\Big] \\
=& \sum_\sigma c_\sigma \underbrace{\prod_{i \in I \backslash [s]} \tilde{\phi}_{P_i \sigma}(x_0) \prod_{i \in [k]} \tilde{\phi}_{P_i \sigma}(x_k)}_{\text{This part is freezed.}} \underbrace{\prod_{i \in [k+1, s]} (\mathbb{E}_u \tilde{\phi}_{P_i \sigma})(x_u)}_{\text{This part is a function of } x_u}.
\end{aligned}
$$

Now, fix $k \in [s]$ and express $a_{\leq k}(x) = a_{\leq k}(x_u, x_{\leq k-1}, x_k)$. An essence of this proof is that the mapping:

$$y_k \mapsto a_{\leq k}(x_u, x_{\leq k-1}, y_k)$$

is a linear combination of $\tilde{\phi}_\sigma(y_k)$ with $\sigma \in \mathcal{F}(\mathcal{A}_{u_k})$ and the coefficients are functions of $(x_u, x_{\leq k-1})$, which gives us room to apply the inductive assumption, or (15) from Assumption B.3.

To aid our analysis, we introduce $Y_k$, an independent copy of $X_k$. By (15), we have

$$\mathbb{E}\Big[\big(\mathbb{E}[a_{\leq k}(x_u, x_{\leq k-1}, Y_k)|Y_u]\big)^2\Big] \leq \exp(-\varepsilon(\mathrm{h}(u) - \mathrm{h}^*))\mathbb{E}_{Y_k}\big[a_{\leq k}^2(x_u, x_{\leq k-1}, Y_k)\big]. \qquad (44)$$

The reason we introduce $Y_k$ is that the L.H.S. and R.H.S. of the above inequality are not related to (any moments of) conditional expectation of $a(X)$. However, it can still be used with some adjustment, relying on (16) from Assumption B.3.

Given the assumption on $C_0$ being greater than or equal to 1, we have

$$\mathrm{h}(u_k) = \mathrm{h}(u) - 1 \overset{(43)}{\geq} \mathrm{h}^* + C_0 - 1 \geq \mathrm{h}^*.$$

Applying (16) to our function $y_k \mapsto a_{\leq k}(x_u, x_{\leq k-1}, y_k)$ we get

$$\mathbb{E}_{Y_k}\big[a_{\leq k}^2(x_u, x_{\leq k-1}, Y_k)\big] \leq \frac{1}{c^*} \min_{\theta \in [q]} \mathbb{E}_{Y_k}\big[a_{\leq k}^2(x_u, x_{\leq k-1}, Y_k)\big|Y_u = \theta\big]$$

$$\leq \frac{1}{c^*} \mathbb{E}_{Y_k}\big[a_{\leq k}^2(x_u, x_{\leq k-1}, Y_k)\big|Y_u = x_u\big]. \qquad (45)$$

On the other hand,

$$\pi(x_u)\big(\mathbb{E}_{Y_k}[a_{\leq k}(x_u, x_{\leq k-1}, Y_k)|Y_u = x_u]\big)^2 \leq \mathbb{E}\Big[\big(\mathbb{E}_{Y_k}[a_{\leq k}(x_u, x_{\leq k-1}, Y_k)|Y_u]\big)^2\Big].$$

Combining the above expression, (44), and (45), we conclude that

$$\big(\mathbb{E}\big[a_{\leq k}(x_u, x_{\leq k-1}, Y_k)\,\big|\,Y_u = x_u\big]\big)^2 \leq \frac{1}{c^*\pi(x_u)}\exp(-\varepsilon(\mathrm{h}(u) - \mathrm{h}^*))\mathbb{E}_{Y_k}\big[a_{\leq k}^2(x_u, x_{\leq k-1}, Y_k)\,\big|\,Y_u = x_u\big].$$

$$(46)$$

Notice that the expression inside the square in L.H.S. is

$$\mathbb{E}\big[a_{\leq k}(x_u, x_{\leq k-1}, Y_k)\,\big|\,Y_u = x_u\big]$$
$$=\mathbb{E}\big[a_{\leq k}(x_u, x_{\leq k-1}, X_k)\,\big|\,X_u = x_u\big]$$
$$=\mathbb{E}\big[a_{\leq k}(X_u, X_{\leq k-1}, X_k)\,\big|\,X_u = x_u, X_{\leq k-1} = x_{\leq k-1}\big]$$
$$=a_{\leq k-1}(x).$$

Similarly,

$$\mathbb{E}_{Y_k}\big[a_{\leq k}^2(x_u, x_{\leq k-1}, Y_k)\,\big|\,Y_u = x_u\big] =\mathbb{E}\big[a_{\leq k}^2(x_u, x_{\leq k-1}, X_k)\,\big|\,X_u = x_u\big]$$
$$=\mathbb{E}\big[a_{\leq k}^2(X_u, X_{\leq k-1}, X_k)\,\big|\,X_u = x_u, X_{\leq k-1} = x_{\leq k-1}\big]$$
$$=a_{\leq k-1}^2(x).$$

By imposing the **first assumption** on $C_0$ that

$$C_0 \geq \frac{1}{\varepsilon}\log\Big(\frac{1}{c^*\min_{j\in[q]}\pi(j)}\Big),$$

it follows from (46) that

$$a_{\leq k-1}^2(x) \leq \exp(-\varepsilon(\mathrm{h}(u) - C_0 - \mathrm{h}^*))\mathbb{E}\big[a_{\leq k}^2(X)\,\big|\,X_u = x_u \text{ and } X_{\leq k-1} = x_{\leq k-1}\big]$$

By taking Conditional Expectation on both sides,

$$(\mathbb{E}_U a_{\leq k-1}^2)(x) \leq \exp(-\varepsilon(\mathrm{h}(u) - C_0 - \mathrm{h}^*))\big(\mathbb{E}_U a_{\leq k}^2\big)(x).$$

Finally, we apply this inequality consecutively for $k \in [s]$ we obtain

$$(\mathbb{E}_U a)^2(x) \leq \exp(-\varepsilon|I'|(\mathrm{h}(u) - C_0 - \mathrm{h}^*))(\mathbb{E}_U a^2)(x).$$

$\square$

**Corollary E.6.** *Fix $u \in T \backslash L$ and a function $f_u(x)$ following the form described in Definition E.4. If $I, J \subseteq [d_u]$ are subsets of $[d_u]$ of size at least 2, then for every $\theta \in [q]$,*

$$|(\mathbb{E}_u f_{u,I})(\theta)| \le \exp\left(-\frac{\varepsilon|I|}{2}(\mathrm{h}(u) - C - \mathrm{h}^*)\right)\sqrt{(\mathbb{E}_u f_{u,I}^2)(\theta)} \tag{47}$$

$$|(\mathbb{E}_u f_{u,I} \cdot f_{u,J})(\theta)| \le \exp\left(-\frac{\varepsilon|I\Delta J|}{2}(\mathrm{h}(u) - C - \mathrm{h}^*)\right)\sqrt{(\mathbb{E}_u f_{u,I}^2)(\theta)} \cdot \sqrt{(\mathbb{E}_u f_{u,J}^2)(\theta)}, \tag{48}$$

*where $C = C(M, c^*)$ is the constant introduced in Proposition E.5, and $I\Delta J := (I \setminus J) \cup (J \setminus I)$.*

*Proof.* For the first statement, it follows from Proposition E.5 with $a(x) = f_{u,I}(x)$ and $I = I'$.

To prove the second statement, we begin by noting that the inputs of $f_{u,I}(x)$ and $f_{u,J}(x)$ do not include $(x_v : v \in \bigcup_{i \in J \setminus I} T_{u_i})$ and $(x_v : v \in \bigcup_{i \in I \setminus J} T_{u_i})$, respectively. Thus, we can apply the Markov Property and that fact that if $Y, Z, W$ are ind pendent then:

$$\mathbb{E}[g(Y,Z)h(Z,W)] = \mathbb{E}[\mathbb{E}[g(Y,Z)h(Z,W)|Z]] = \mathbb{E}[\mathbb{E}[g(Y,Z)|Z]h(Z,W)]$$

and this in turn becomes:

$$\mathbb{E}[\mathbb{E}[g(Y,Z)|Z]h(Z,W)|W] = \mathbb{E}[\mathbb{E}[g(Y,Z)|Z]\mathbb{E}[h(Z,W)|W]],$$

to obtain

$$\begin{aligned} &(\mathbb{E}_u f_{u,I} f_{u,J})(x) \\ =&\mathbb{E}\left[\mathbb{E}\left[f_{u,I}(X) \;\Big|\; X_v : v \notin \bigcup_{i\in I\setminus J} T_{u_i}\right] \cdot \mathbb{E}\left[f_{u,J}(X) \;\Big|\; X_v : v \notin \bigcup_{i\in J\setminus I} T_{u_i}\right]\;\Big|\; X_v = x_v : v \not< u\right]. \end{aligned}$$

In terms of absolute value, by Proposition E.5 we have

$$\begin{aligned} &\left|(\mathbb{E}_u f_{u,I} f_{u,J})(x)\right| \\ \le&\mathbb{E}\left[\left|\mathbb{E}\left[f_{u,I}(X)f_{u,J}(X) \;\Big|\; X_v : v \notin \bigcup_{i\in I\Delta J} T_{u_i}\right]\right| \;\Big|\; X_v = x_v : v \not< u\right] \\ =&\mathbb{E}\left[\left|\mathbb{E}\left[f_{u,I}(X) \;\Big|\; X_v : v \notin \bigcup_{i\in I\setminus J} T_{u_i}\right]\right| \cdot \left|\mathbb{E}\left[f_{u,J}(X) \;\Big|\; X_v : v \notin \bigcup_{i\in J\setminus I} T_{u_i}\right]\right| \;\Big|\; X_v = x_v : v \not< u\right] \\ \le&\mathbb{E}\left[\sqrt{\exp(-\varepsilon|I\setminus J|(\mathrm{h}(u) - C - \mathrm{h}^*)) \cdot \mathbb{E}\left[f_{u,I}^2(X) \;\Big|\; X_v : v \notin \bigcup_{i\in I\setminus J} T_{u_i}\right]} \right. \\ &\left. \cdot \sqrt{\exp(-\varepsilon|J\setminus I|(\mathrm{h}(u) - C - \mathrm{h}^*)) \cdot \mathbb{E}\left[f_{u,J}^2(X) \;\Big|\; X_v : v \notin \bigcup_{i\in J\setminus I} T_{u_i}\right]} \;\Big|\; X_v = x_v : v \not< u\right] \\ \le&\exp\left(-\frac{\varepsilon}{2}|I\Delta J|(\mathrm{h}(u) - C - \mathrm{h}^*)\right)\sqrt{\mathbb{E}\left[\mathbb{E}\left[f_{u,I}^2(X) \;\Big|\; X_v : v \notin \bigcup_{i\in I\setminus J} T_{u_i}\right] \;\Big|\; X_v = x_v : v \not< u\right]} \\ &\cdot \sqrt{\mathbb{E}\left[\mathbb{E}\left[f_{u,J}^2(X) \;\Big|\; X_v : v \notin \bigcup_{i\in J\setminus I} T_{u_i}\right] \;\Big|\; X_v = x_v : v \not< u\right]} \\ =&\exp\left(-\frac{\varepsilon}{2}|I\Delta J|(\mathrm{h}(u) - C - \mathrm{h}^*)\right)\sqrt{(\mathbb{E}_u f_{u,I}^2)(x) \cdot (\mathbb{E}_u f_{u,J}^2)(x)}, \end{aligned}$$

where the second to last inequality follows from Hölder's inequality. $\square$

**Corollary E.7.** *There exists $C = C(M, d, c^*) \ge 1$ so that the following holds. If $u \in T\backslash L$ with $\mathrm{h}(u) \ge \mathrm{h}^* + C(1 + \log(R))$, then for any $f_u(x)$ in the form as described in Definition E.4,*

$$\forall \theta \in [q], \; \frac{1}{2} \cdot \sum_{I \subset [d_u] : |I| \ge 2} (\mathbb{E}_u f_{u,I}^2)(\theta) \le (\mathbb{E}_u f_u^2)(\theta). \tag{49}$$

*Proof.* Let $C_0 = C_0(M, d, c^*)$ denote the constant introduced in the statement of the Lemma. Its value will be determined along the proof.

The statement of the Corollary is trivial when $d_u < 2$ since in that case $\mathcal{B}_u = \emptyset$, implying $f_u = 0$. From now on, we assume $d_u \geq 2$.

First,

$$(\mathbb{E}_u f_u^2)(x) - \sum_{I \in [d_u]\,:\,|I| \geq 2} (\mathbb{E}_u f_{u,I}^2)(x) = \sum_{\{I,J\}} 2(\mathbb{E}_u f_{u,I} \cdot f_{u,J})(x)$$

where $\sum_{\{I,J\}}$ refers to the sum over all unordered pairs $\{I, J\}$ with $I$ and $J$ being distinct subsets of $[d_u]$ of size at least 2.

We can apply (48) to estimate the absolute value of the difference.

$$\left| \sum_{\{I,J\}} 2(\mathbb{E}_u f_{u,I} \cdot f_{u,J})(x) \right|$$

$$\leq \sum_{\{I,J\}} 2 \exp\left( - \frac{\varepsilon |I \Delta J|}{2} (\mathrm{h}(u) - C_{E.5} - \mathrm{h}^*) \right) \sqrt{(\mathbb{E}_u f_{u,I}^2)(x)} \cdot \sqrt{(\mathbb{E}_u f_{u,J}^2)(x)}, \qquad (50)$$

where the constant $C_{E.5}$ is the constant $C_{E.5}$ introduced in Proposition E.5. By $2|ab| \leq a^2 + b^2$ for $a, b \in \mathbb{R}$,

$$2\sqrt{(\mathbb{E}_u f_{u,I}^2)(x)} \cdot \sqrt{(\mathbb{E}_u f_{u,J}^2)(x)} \leq (\mathbb{E}_u f_{u,I}^2)(x) + (\mathbb{E}_u f_{u,J}^2)(x).$$

Hence,

$$(50) \leq \sum_{I \subset [d_u]\,:\,|I| \geq 2} (\mathbb{E}_u f_{u,I}^2)(x) \left( \sum_{J \subset [d_u]\,:\,I \neq J} \exp\left( - \frac{\varepsilon |I \Delta J|}{2} (\mathrm{h}(u) - C_{E.5} - \mathrm{h}^*) \right) \right).$$

With $|\{J \subseteq [d_u]\,:\,|I \Delta J| = i\}| = \binom{d_u}{i} \leq d_u^i$,

$$\sum_{J \subset [d_u]\,:\,I \neq J} \exp\left( - \frac{\varepsilon |I \Delta J|}{2} (\mathrm{h}(u) - C_{E.5} - \mathrm{h}^*) \right) \leq \sum_{i=1}^{\infty} d_u^i \exp\left( - \frac{\varepsilon i}{2} (\mathrm{h}(u) - C_{E.5} - \mathrm{h}^*) \right) \leq 1/4,$$

$$(51)$$

provided that $\mathrm{h}(u) - C_{E.5} - \frac{\log(d_u)}{\varepsilon} - \mathrm{h}^* \geq \frac{16}{\varepsilon}$.

Now, we impose the **first assumption** on $C_0$ that

$$C_0 \geq C_{E.5} + \frac{\log(d)}{\varepsilon} + \frac{1}{\varepsilon} + \frac{16}{\varepsilon},$$

then

$$\mathrm{h}(u) \geq \mathrm{h}^* + C_0(\log(R) + 1) \geq C_{E.5} + \frac{\log(d_u)}{\varepsilon} + \frac{16}{\varepsilon} + \mathrm{h}^*.$$

Hence,

$$\left| (\mathbb{E}_u f_u^2)(x) - \sum_{I \in [d]\,:\,|I| \geq 2} (\mathbb{E}_u f_{u,I}^2)(x) \right| \leq \frac{1}{4} \sum_{I \in [d]\,:\,|I| \geq 2} (\mathbb{E}_u f_{u,I}^2)(x) \qquad (52)$$

and the proof follows. $\qquad \square$

*Proof of Proposition E.1.* Let $C_0 = C_0(M, d, c^*)$ denote the constant introduced in the statement of the Proposition. Its precise value will be determined along the proof. From Remark E.3, it is sufficient to consider the case when $|d_u| \geq 2$. Further, it suffices to prove in the case when

$$\mathrm{h}(u) \geq \mathrm{h}^* + C_0(\log(R) + 1), \qquad (53)$$

since otherwise the statements follow from either Cauchy-Schwarz or Jenson's inequality.

**Part I: Derivation of** (37) **and** (38).

First, by (47),

$$
(\mathbb{E}_u f_u)^2(\theta)) = \Big( \sum_{I \subseteq [d_u]\,:\,|I| \geq 2} (\mathbb{E}_u f_{u,I})(\theta) \Big)^2 \tag{54}
$$

$$
\leq \Big( \sum_{I \subseteq [d_u]\,:\,|I| \geq 2} \exp\Big( -\frac{\varepsilon |I|}{2}(\mathrm{h}(u) - C_{E.5} - \mathrm{h}^*) \Big) \cdot \sqrt{(\mathbb{E}_u f_{u,I}^2)(\theta)} \Big)^2
$$

$$
\leq \Big( \sum_{I \subseteq [d_u]:|I| \geq 2} \exp\Big( -\varepsilon |I|(\mathrm{h}(u) - C_{E.5} - \mathrm{h}^*)) \Big) \cdot \Big( \sum_{I \subseteq [d_u]:|I| \geq 2} (\mathbb{E}_u f_{u,I}^2)(\theta) \Big),
$$

where we applied Cauchy-Schwarz inequality in the last inequality; the constant $C_{E.5}$ is the constant $C$ introduced in Proposition E.5.

With the coarse estimate

$$
\big|\{ I \subseteq [d_u] : |I| = t \}\big| = \binom{d_u}{i} \leq d_u^t \leq (Rd)^t,
$$

we have

$$
\Big( \sum_{I \subseteq [d_u]:|I| \geq 2} \exp\Big( -\varepsilon |I|(\mathrm{h}(u) - C_{E.5} - \mathrm{h}^*) \Big) \Big)
$$

$$
\leq \sum_{t=2}^{\infty} \exp\Big( -\varepsilon t\Big( \mathrm{h}(u) - C_{E.5} - \frac{\log(R) + \log(d)}{\varepsilon} - \mathrm{h}^* \Big) \Big). \tag{55}
$$

The geometric series above is finite if $\mathrm{h}(u)$ is large enough, and this can be achieved by imposing assumption of $C_0$ and relying on (53). Now, let us impose the **first assumption** on $C_0$:

$$
C_0 \geq C_{E.5} + (2 + 2\log(d) + 100)/\varepsilon. \tag{56}
$$

Then, by (53) we have

$$
\mathrm{h}(u) \geq \mathrm{h}^* + C_0(\log(R) + 1) \geq \mathrm{h}^* + C_{E.5} + 2\frac{\log(R) + \log(d)}{\varepsilon} + \frac{100}{\varepsilon},
$$

which in term implies the R.H.S. of (55) is

$$
\frac{\exp\Big( -2\varepsilon\Big( \mathrm{h}(u) - C_{E.5} - \frac{\log(R)+\log(d)}{\varepsilon} - \mathrm{h}^* \Big) \Big)}{1 - \exp\Big( -\varepsilon\Big( \mathrm{h}(u) - C_{E.5} - \frac{\log(R)+\log(d)}{\varepsilon} - \mathrm{h}^* \Big) \Big)}
$$

$$
\leq \frac{\exp\Big( -2\varepsilon\Big( \mathrm{h}(u) - C_{E.5} - \frac{\log(R)+\log(d)}{\varepsilon} - \mathrm{h}^* \Big) \Big)}{1 - e^{-100}}
$$

$$
\leq 2\exp\Big( -2\varepsilon\Big( \mathrm{h}(u) - C_{E.5} - \frac{\log(R) + \log(d)}{\varepsilon} - \mathrm{h}^* \Big) \Big)
$$

$$
= \frac{1}{4}\exp\Big( -2\varepsilon\Big( \mathrm{h}(u) - C_{E.5} - \frac{\log(R) + \log(d)}{\varepsilon} - \mathrm{h}^* - \frac{\log(8)}{2\varepsilon} \Big) \Big)
$$

$$
\leq \frac{1}{4}\exp\Big( -2\varepsilon\big( \mathrm{h}(u) - C_0(\log(R) + 1) - \mathrm{h}^* \big) \Big)
$$

Substituting the above estimate into (54), together with (49) we have

$$
(\mathbb{E}_u f_u)^2(\theta) \leq \frac{1}{4}\exp\Big( -2\varepsilon\big( \mathrm{h}(u) - C_0(\log(R) + 1) - \mathrm{h}^* \big) \Big) \cdot \Big( \sum_{I \subseteq [d_u]:|I| \geq 2} (\mathbb{E}_u f_{u,I}^2)(\theta) \Big)
$$

$$
\leq \frac{1}{2}\exp\Big( -2\varepsilon\big( \mathrm{h}(u) - C_0(\log(R) + 1) - \mathrm{h}^* \big) \Big)(\mathbb{E}_u f_u^2)(\theta). \tag{57}
$$

Therefore, we have derived an inequality which is slightly stronger than (37).

To derive (38), let us first show $\mathbb{E}f_u(X)$ is relatively small using (57) and Jesnon's inequality:

$$\big(\mathbb{E}[f_u(X)]\big)^2 \leq \mathbb{E}\big[(\mathbb{E}_u f_u)^2(X)\big] \leq \frac{1}{2}\exp\Big(-2\varepsilon\big(\mathrm{h}(u) - C_0(\log(R) + 1) - \mathrm{h}^*\big)\Big)\mathbb{E}\big[f_u^2(X)\big]$$

$$\leq \frac{1}{2}\mathbb{E}\big[f_u^2(X)\big].$$

Thus, the variance and the second moment of $f_u(X)$ are the same up to a factor of 2:

$$\mathbb{E}\big[\tilde{f}_u^2(X)\big] = \mathbb{E}\big[f_u^2(X)\big] - \big(\mathbb{E}[f_u(X)]\big)^2 \geq \frac{1}{2}\mathbb{E}\big[f_u^2(X)\big]. \tag{58}$$

We conclude that

$$\mathbb{E}\big[(\mathbb{E}_u \tilde{f}_u)^2(X)\big] \leq \mathbb{E}\big[(\mathbb{E}_u f_u)^2(X)\big]$$

$$\leq \frac{1}{2}\exp\Big(-2\varepsilon\big(\mathrm{h}(u) - C_0(\log(R) + 1) - \mathrm{h}^*\big)\Big)\mathbb{E}\big[f_u^2(X)\big]$$

$$\leq \exp(-2\varepsilon(\mathrm{h}(u) - C_0(1 + \log(R)) - \mathrm{h}^*))\mathbb{E}\big[\tilde{f}_u^2(X)\big].$$

Therefore, we complete the proof of (38).

**Part II: Derivation of** (39).

It remains to show (39) and the proof is similar. Fix $I \subset [d_u]$ with $|I| \geq 2$, let $I' = I\backslash\{i\}$ and we represent $x \in [q]^T$ as $(x_0, x_1)$, where

$$x_0 := \big(x_v \,:\, v \notin \bigcup_{j \in I'} T_{u_j}\big) \qquad \text{and} \qquad x_1 := \big(x_v \,:\, v \in \bigcup_{j \in I'} T_{u_j}\big).$$

With this notation, we have $a(x) = a(x_0)$. Thus,

$$\mathbb{E}\big[|\tilde{f}_{u,I}(X)a(X)|\big] = \mathbb{E}\Big[\big|\mathbb{E}\big[\tilde{f}_{u,I}(X)\,|\,X_0\big]\cdot a(X_0)\big|\Big]$$

$$\leq \sqrt{\mathbb{E}\Big[\big(\mathbb{E}\big[\tilde{f}_{u,I}(X)\,|\,X_0\big]\big)^2\Big]} \cdot \sqrt{\mathbb{E}\big[a^2(X_0)\big]}$$

$$\leq \sqrt{\mathbb{E}\Big[\big(\mathbb{E}\big[f_{u,I}(X)\,|\,X_0\big]\big)^2\Big]} \cdot \sqrt{\mathbb{E}\big[a^2(X_0)\big]}$$

$$\leq \exp\Big(-\frac{\varepsilon}{2}|I\backslash\{i\}|(\mathrm{h}(u) - C_{E.5} - \mathrm{h}^*)\Big)\sqrt{\mathbb{E}\big[f_{u,I}^2(X)\big]}\cdot\sqrt{\mathbb{E}\big[a^2(X)\big]},$$

where the last inequality follows from Proposition E.5. Hence,

$$\mathbb{E}\big[|\tilde{f}_u(X)a(X)|\big]$$

$$\leq \sum_{I \subseteq [d_u]\,:\,|I| \geq 2} \exp\Big(-\frac{\varepsilon}{2}|I\backslash\{i\}|(\mathrm{h}(u) - C_{E.5} - \mathrm{h}^*)\Big)\sqrt{\mathbb{E}\big[f_{u,I}^2(X)\big]}\cdot\sqrt{\mathbb{E}\big[a^2(X)\big]}$$

$$\leq \Big(\sum_{I \subseteq [d_u]\,:\,|I| \geq 2} \exp\big(-\varepsilon|I\backslash\{i\}|(\mathrm{h}(u) - C_{E.5} - \mathrm{h}^*)\big)\Big)^{1/2} \cdot \sqrt{\sum_{I \subseteq [d_u]\,:\,|I| \geq 2}\mathbb{E}\big[f_{u,I}^2(X)\big]}\cdot\sqrt{\mathbb{E}\big[a^2(X)\big]}. \tag{59}$$

Next, we impose the **second assumption** on $C_0$ that

$$C_0 \geq C_{E.7},$$

where $C_{E.7}$ is the constant introduced in Corollary E.7. Together our assumption $\mathrm{h}(u) \geq \mathrm{h}^* + C_0(\log(R) + 1)$ at the beginning of the proof, we can apply the Corollary and (58) to get

$$\sum_{I \subseteq [d_u]\,:\,|I| \geq 2} \mathbb{E}f_{u,I}^2(X) \leq 2\mathbb{E}(f_u(X))^2 \leq 4\mathbb{E}(\tilde{f}_u(X))^2. \tag{60}$$

Repeating the same argument as in the proof of (38) and relying on the assumption (56) of $C_0$,

$$\sum_{I \subseteq [d_u] : |I| \geq 2} \exp(-\varepsilon |I \setminus \{i\}| (\mathrm{h}(u) - C_{E.5} - \mathrm{h}^*)) \leq \sum_{t=1}^{\infty} \exp\left(-\varepsilon t \left(\mathrm{h}(u) - C_{E.5} - \mathrm{h}^* - 2\frac{\log(Rd)}{\varepsilon}\right)\right)$$

$$\leq \frac{1}{4} \exp\left(-\varepsilon \left(\mathrm{h}(u) - C_0(\log(R) + 1) - \mathrm{h}^*\right)\right). \tag{61}$$

Therefore, combining (60), (61), and (59) we get

$$\mathbb{E}\left[|\tilde{f}_u(X) a(X)|\right] \leq \exp\left(-\frac{\varepsilon}{2}\left(\mathrm{h}(u) - C_0(\log(R) + 1) - \mathrm{h}^*\right)\right) \cdot \sqrt{\mathbb{E}\left[\tilde{f}_u^2(X)\right]} \cdot \sqrt{\mathbb{E}\left[a^2(X)\right]}.$$

$\square$

# F   Induction Step 2: Decay of $f_k$ and the proof of Theorem B.6

As a continuation of the inductive step, we adapt the notation introduced in the previous section. Building on the properties of an single component $f_u$ from Proposition E.1, our objective is to deduce variance and covariance decay of $f_k$, which is stated in Proposition F.1 below. Once it is established, we will be ready to prove Theorem B.6.

## F.1   Properties of $f_k$

The main goal of this subsection is to derive the following Proposition.

**Proposition F.1.** *There exists $C = C(M, d, c^*) \geq 1$ so that the following holds. For any $\rho' \in T$ satisfying*

$$\mathrm{h}(\rho') \geq \mathrm{h}^* + C(1 + \log(R)).$$

*Fix a positive integer $k_1$ such that*

$$\mathrm{h}(\rho') \geq k_1 \geq \mathrm{h}^* + C(1 + \log(R)).$$

*Consider a function*

$$f(x) = c + \sum_{\sigma \in \mathcal{F}(\mathcal{B}_{\leq \rho'})} c_\sigma \phi_\sigma(x)$$

*with $\mathbb{E}f(X) = 0$. We decompose $f$ according to Lemma D.1 with the given $k_1$. Then, the following holds:*

- *for $k \in [k_1 + 1, \mathrm{h}(\rho')]$,*

$$\mathbb{E}\left[(\mathbb{E}_{\rho'} f_k)^2 (X_{\rho'})\right] \leq \exp\left(-\varepsilon \left(\mathrm{h}(\rho') + k - C(\log(R) + 1) - 2\mathrm{h}^*\right)\right) \mathbb{E}f_k^2(X), \tag{62}$$

- *for $k = k_1$,*

$$\mathbb{E}\left[(\mathbb{E}_{\rho'} f_{k_1})^2 (X_{\rho'})\right] \leq \exp\left(-\varepsilon \left(\mathrm{h}(\rho') - k - C(\log(R) + 1)\right)\right) \mathbb{E}f_{k_1}^2(X), \text{ and} \tag{63}$$

- *for $k_1 \leq m < k \leq \mathrm{h}(\rho')$,*

$$\left|\mathbb{E}\left[f_k(X) f_m(X)\right]\right| \leq \exp\left(-\frac{\varepsilon}{2}\left(k - C(\log(R) + 1) - \mathrm{h}^*\right)\right) \sqrt{\mathbb{E}\left[f_k^2(X)\right] \mathbb{E}\left[f_m^2(X)\right]}, \tag{64}$$

Before we prove the Proposition, let us prove the following second moment bounds for the partial sums of $\tilde{f}_u$.

**Lemma F.2.** *There exists a constant $C = C(M, d, c^*) \geq 1$ so that the following holds. Consider the same description as stated in Proposition F.1 and $k_1 \geq \mathrm{h}^* + C(\log(R) + 1)$. Let $(h, k)$ be a pair of integers satisfying $k_1 < h \leq k \leq l'$. For $u \in D_k(\rho')$, let*

$$f_{h,u}(x) = \sum_{v \in D_h(u)} \tilde{f}_v(x).$$

*In other words,*

$$f_h(x) = \sum_{u \in D_k(\rho')} f_{h,u}(x).$$

*The following holds: First, for $u \in D_k(\rho')$,*

$$\frac{1}{2} \sum_{v \in D_h(u)} \mathbb{E}\tilde{f}_v^2(X) \leq \mathbb{E}f_{h,u}^2(X) \leq 2 \sum_{v \in D_h(u)} \mathbb{E}\tilde{f}_v^2(X).$$

*Second,*

$$\frac{1}{4} \sum_{u \in L_k(\rho')} \mathbb{E}f_{h,u}^2(X) \leq \mathbb{E}f_h^2(X) \leq 4 \sum_{u \in L_k(\rho')} \mathbb{E}f_{h,u}^2(X).$$

*Proof.* Let $C_0 = C_0(M, d, \varepsilon')$ denote the constant introduced in the statement of the Lemma. Its precise value will be determined along the proof.

Let us fix $u \in D_k(\rho')$. Consider the following conditional expectation of $f_{h,u}(x)$.

$$(\mathbb{E}_h f_{h,u})(x) = \mathbb{E}\big[f_{h,u}(X) \,\big|\, X_v = x_v \,:\, \mathrm{h}(v) \geq h\big] = \sum_{v \in D_h(\rho')} (\mathbb{E}_v \tilde{f}_v)(x_v).$$

Comparing the second moments of $f_{h,u}(x) = \sum_{v \in D_h(u)} \tilde{f}_v(x)$ and $\sum_{v \in D_h(u)} (\mathbb{E}_v \tilde{f}_v)(x_v)$ we get

$$
\begin{aligned}
\mathbb{E}\bigg[\bigg(\sum_{v \in D_h(u)} \tilde{f}_v(X)\bigg)^2\bigg] &= \sum_{v \in D_h(u)} \mathbb{E}\big[\tilde{f}_v^2(X)\big] + \sum_{v,v' \in D_h(u)\,:\,v \neq v'} \mathbb{E}\big[\tilde{f}_v(X)\tilde{f}_{v'}(X)\big] \\
&= \sum_{v \in D_h(u)} \mathbb{E}\big[\tilde{f}_v^2(X)\big] + \sum_{(v,v') \in (D_h(u))^2\,:\,v \neq v'} \mathbb{E}\bigg[\mathbb{E}\Big[(\mathbb{E}_v \tilde{f}_v)(X)(\mathbb{E}_{v'} \tilde{f}_{v'})(X) \,\Big|\, X_{\mathfrak{p}(v,v')}\Big]\bigg] \\
&= \mathbb{E}\bigg[\bigg(\sum_{v \in D_h(u)} (\mathbb{E}_v \tilde{f}_v)(X)\bigg)^2\bigg] + \sum_{v \in D_h(u)} \Big(\mathbb{E}\big[\tilde{f}_v^2(X)\big] - \mathbb{E}\big[(\mathbb{E}_v \tilde{f}_v)^2(X)\big]\Big) \\
&\geq \sum_{v \in D_h(u)} \Big(1 - \exp\Big(-\varepsilon\big(h - C_{E.1}(1 + \log(R)) - \mathrm{h}^*\big)\Big)\Big) \mathbb{E}\big[\tilde{f}_v^2(X)\big],
\end{aligned}
$$
(65)

where the last inequality follow from Proposition E.1 and $C_{E.1}$ is the constant $C$ introduced in Proposition E.1.

Here we impose the **first assumption** on $C_0$:

$$C_0 > 10 \max\{\varepsilon^{-1}, C_{E.1}\}.$$

Then, due to $k_1 \geq \mathrm{h}^* + C_0(\log(R) + 1)$, we have

$$\exp(-\varepsilon(h - C_{E.1}(1 + \log(R)) - \mathrm{h}^*))) \leq \exp(-\varepsilon(k_1 - C_{E.1}(1 + \log(R)) - \mathrm{h}^*))) \leq \exp(-\varepsilon \cdot 0.9 C_0) \leq 1/2,$$

and thus (65) can be simplified to

$$\mathbb{E}\big[f_{h,u}^2(X)\big] \geq \frac{1}{2} \sum_{v \in D_h(u)} \mathbb{E}\big[\tilde{f}_v^2(X)\big]. \tag{66}$$

With the lower bound been established, the upper bound can also be derived in the same fashion. Let us first recycle the first three lines of (65):

$$\mathbb{E}\left[\left(\sum_{v\in D_h(u)}\tilde{f}_v(X)\right)^2\right] = \mathbb{E}\left(\sum_{v\in D_h(u)}(\mathbb{E}_v\tilde{f}_v)(X)\right)^2 + \sum_{v\in D_h(u)}\mathbb{E}(\tilde{f}_v(X))^2 - \mathbb{E}(\mathbb{E}_v\tilde{f}_v)^2(X)$$

$$\leq \mathbb{E}\left(\sum_{v\in D_h(u)}(\mathbb{E}_v\tilde{f}_v)(X)\right)^2 + \sum_{v\in D_h(u)}\mathbb{E}(\tilde{f}_v(X))^2.$$

Notice that we can apply Lemma C.2 for the first summand in the above expression.

$$\mathbb{E}\left(\sum_{v\in D_h(u)}(\mathbb{E}_v\tilde{f}_v)(X)\right)^2 = \mathbb{E}\left(\sum_{v\in D_h(u)}(\mathbb{E}_v\tilde{f}_v)(X_v)\right)^2 \leq C_{C.2}R\sum_{v\in D_h(u)}(\mathbb{E}_v\tilde{f}_v)^2(X_v)$$

where $C_{C.2}$ is the constant introduced in Lemma C.2. Again, applying the estimate from Proposition E.1 we have

$$C_{C.2}R\sum_{v\in D_h(u)}(\mathbb{E}_v\tilde{f}_v)^2(X_v) \leq C_{C.2}R\exp\left(-2\varepsilon\big(h - C_{E.1}(1+\log(R)) - \mathrm{h}^*\big)\right)\sum_{v\in D_h(u)}\mathbb{E}\tilde{f}_v^2(X).$$

Here we impose the **second assumption** on $C_0$ that

$$C_0 \geq C_{E.1} + \frac{1+\log(C_{C.2})}{2\varepsilon}. \tag{67}$$

Then, relying on $h > k_1 \geq \mathrm{h}^* + C_0(\log(R)+1)$,

$$C_{C.2}R\exp\left(-2\varepsilon\big(h - C_{E.1}(1+\log(R)) - \mathrm{h}^*\big)\right)$$

$$\leq \exp\left(-2\varepsilon\Big(h - C_{E.1}(1+\log(R)) - \mathrm{h}^* - \frac{\log(C_{C.2}) + \log(R)}{2\varepsilon}\Big)\right)$$

$$\leq 1,$$

which in turn implies

$$\mathbb{E}(f_{h,u}(X))^2 \leq 2\sum_{v\in D_h(u)}\mathbb{E}(\tilde{f}_v(X))^2.$$

Now it remains to show the second statement. Notice that $f_h = f_{h,\rho'}$, we immediately have

$$\frac{1}{2}\sum_{v\in D_h(\rho')}\mathbb{E}\tilde{f}_v^2(X) \leq \mathbb{E}f_h^2(X) \leq 2\sum_{v\in D_h(\rho')}\mathbb{E}\tilde{f}_v^2(X)$$

Together with

$$\frac{1}{2}\sum_{u\in D_k(\rho')}\mathbb{E}f_{k,u}^2(X) \leq \sum_{v\in D_h(\rho')}\mathbb{E}\tilde{f}_v^2(X) \leq 2\sum_{u\in D_k(\rho')}\mathbb{E}f_{k,u}^2(X),$$

the second statement of the lemma follows.

$$\square$$

*Proof of Proposition F.1.* Let $C_0 = C_0(M, d, c^*)$ denote the constant introduced in the statement of the Lemma. Its precise value will be determined along the proof. Let us make the **first assumption** on $C_0$ that

$$C_0 \geq C_{F.2},$$

where $C_{F.2}$ is the constant introduced in Lemma F.2. Now, we could apply the statements of the Lemma.

**Part 1: Derivation of** (62).

Fix $k \in [k_1 + 1, l']$. Applying Lemma F.2 with the parameters $h$ and $k$ in the Lemma setting to be $k$,

$$\mathbb{E}(f_k(X))^2 \geq \frac{1}{2} \sum_{u \in D_k(\rho')} \mathbb{E}(\tilde{f}_u(X))^2. \tag{68}$$

The next step is to compare the sum of $\mathbb{E}\tilde{f}_u^2(X)$ with $\mathbb{E}\big[(\mathbb{E}_{\rho'} f_k)^2(X_{\rho'})\big]$. By Jenson's inequality,

$$\begin{aligned}
\mathbb{E}\big[(\mathbb{E}_{\rho'} f_k)^2(X_{\rho'})\big] =& \mathbb{E}\Big[\Big(\sum_{u \in D_k(\rho')} (\mathbb{E}_{\rho'} \tilde{f}_u)(X_{\rho'})\Big)^2\Big] \\
\leq& \mathbb{E}\Big[|D_k(\rho')| \sum_{u \in D_k(\rho')} (\mathbb{E}_{\rho'} \tilde{f}_u)^2(X_{\rho'})\Big] \\
=& |D_k(\rho')| \sum_{u \in D_k(\rho')} \mathbb{E}\big[(\mathbb{E}_{\rho'} \tilde{f}_u)^2(X_{\rho'})\big].
\end{aligned}$$

For each summand, we can apply (8) from Lemma A.5 to get the following estimate.

$$\mathbb{E}\big[(\mathbb{E}_{\rho'} \tilde{f}_u)^2(X_{\rho'})\big] \leq C_{A.5}(l' - k)^{2q} \lambda^{2(l'-k)} \mathbb{E}\big[(\mathbb{E}_u \tilde{f}_u)^2(X_u)\big]$$

where $C_{A.5}$ is the constant introduced in the Lemma. Together with $|D_k(\rho')| \leq R d^{l'-k}$ from the assumption on $T$ and $d\lambda^2 \leq \exp(-1.1\varepsilon)$ from the definiton of $\varepsilon$,

$$\begin{aligned}
\mathbb{E}\big[(\mathbb{E}_{\rho'} f_k)^2(X_{\rho'})\big] \leq& C_{A.5}(l' - k)^{2q} R \exp(-1.1\varepsilon(l' - k)) \sum_{u \in D_k(\rho')} \mathbb{E}\big[(\mathbb{E}_u \tilde{f}_u)^2(X_u)\big] \\
\leq& \frac{1}{2} \exp\Big(-\varepsilon\Big(l' - C'(1 + \log(R)) - k\Big)\Big) \sum_{u \in D_k(\rho')} \mathbb{E}\big[(\mathbb{E}_u \tilde{f}_u)^2(X_u)\big] \tag{69}
\end{aligned}$$

where we set

$$C' = 1 + \log\big(1 + 2C_{A.5} \max_{n \in \mathbb{N}} n^{2q} \exp(-0.1\varepsilon n)\big) < +\infty.$$

By Proposition E.1 we have

$$\mathbb{E}\big[(\mathbb{E}_u \tilde{f}_u)^2(X)\big] \leq \exp(-2\varepsilon(k - C_{E.1}(1 + \log(R)) - h^*))\mathbb{E}(\tilde{f}_u(X))^2, \tag{70}$$

where $C_{E.1}$ is the constant $C$ introduced in the Proposition. Substituting this inequality into (69), together with (68) from first step,

$$\begin{aligned}
\mathbb{E}\big[(\mathbb{E}_{\rho'} f_k)^2(X_{\rho'})\big] \leq& \frac{1}{2} \exp\Big(-\varepsilon\Big(l' + k - (C' + C_{E.1})(\log(R) + 1) - 2h^*\Big)\Big) \sum_{u \in D_k(\rho')} \mathbb{E}(\tilde{f}_u(X))^2 \\
\leq& \exp(-\varepsilon(l' + k - (C' + C_{E.1})(\log(R) + 1) - 2h^*))\mathbb{E}(f_k(X))^2.
\end{aligned}$$

Now, we impose the **second assumption** on $C_0$ that

$$C_0 \geq (C' + C_{E.1}),$$

we finished the proof of (62).

**Part 2: Derivation of** (63).

Let us consider

$$h_{k_1}(x) := (\mathbb{E}_{k_1} f_{k_1})(x) = \mathbb{E}\big[f_{k_1}(X) \,\big|\, X_u = x_u \,:\, u \in D_{k_1}(\rho')\big].$$

In other words, we may view $h_{k_1}(x)$ as a linear function with variables $x_u$ for $u \in D_{k_1}(\rho')$ with $\mathbb{E}h_{k_1}(X) = \mathbb{E}f_{k_1}(X) = 0$. Then,

$$\begin{aligned}
\mathbb{E}\big[(\mathbb{E}_{\rho'} f_{k_1})^2(X_{\rho'})\big] = \mathbb{E}\big[(\mathbb{E}_{\rho'} h_{k_1})^2(X_{\rho'})\big] \leq& \exp\big(-\varepsilon(h(\rho') - k_1 - C_{B.5})\big)\mathbb{E}h_{k_1}^2(X) \\
\leq& \exp\big(-\varepsilon(h(\rho') - k_1 - C_{B.5})\big)\mathbb{E}f_{k_1}^2(X)
\end{aligned}$$

The first inequality follows from Proposition B.5. The second inequality follows from Jensen's inequality. Here we impose the **third assumption** on $C_0$ that

$$C_0 \geq C_{B.5},$$

the derivation of (63) follows.

**Part 3: Derivation of** (64)

For $w \leq \rho'$ with $m \leq \mathrm{h}(w) \leq k$, let

$$f_{m,w}(x) = \sum_{v \in D_k(w)} \tilde{f}_v(x).$$

Let us make a remark that either by second property of $f$ from Lemma D.1 when $m = k_1$ or by Lemma F.2 in the case when $m > k_1$, we have the following: For $w \leq \rho'$ and $m \leq k' \leq \mathrm{h}(w)$,

$$\Big( \sum_{u \in D_{k'}(w)} \mathbb{E} f_{m,u}^2(X) \Big)^{1/2} \leq C_{D.1} R^2 \big( \mathbb{E} f_{m,w}^2(X) \big)^{1/2}. \tag{71}$$

With this notation,

$$\mathbb{E} f_k(X) f_m(X)$$
$$= \sum_{u \in D_k(\rho')} \mathbb{E} \tilde{f}_u(X) f_{m,u}(X) + \sum_{u,u' \in D_k(\rho') \,:\, u \neq u'} \mathbb{E} \tilde{f}_u(X) f_{m,u'}(X)$$
$$= \sum_{u \in D_k(\rho')} \mathbb{E} \tilde{f}_u(X) f_{m,u}(X) + \sum_{u,u' \in D_k(\rho') \,:\, u \neq u'} \mathbb{E} \Big[ \mathbb{E} \big[ (\mathbb{E}_u \tilde{f}_u)(X) (\mathbb{E}_{u'} f_{m,u'})(X) \,\big|\, X_{\mathfrak{p}(u,u')} \big] \Big]$$
$$= \mathbb{E} \Big[ \Big( \sum_{u \in D_k(\rho')} (\mathbb{E}_u \tilde{f}_u)(X) \Big) \Big( \sum_{u \in D_k(\rho')} (\mathbb{E}_u f_{m,u})(X) \Big) \Big] + \sum_{u \in D_k(\rho')} \mathbb{E} \tilde{f}_u(X) f_{m,u}(X) \tag{72}$$
$$\quad - \sum_{u \in D_k(\rho')} \mathbb{E} \Big[ (\mathbb{E}_u \tilde{f}_u)(X) (\mathbb{E}_u f_{m,u})(X) \Big].$$

We will estimate the three summands individually.

**Part 3.1: Estimating first summand of** (72) First, we apply Cauchy-Schwarz inequality,

$$\Big| \mathbb{E} \Big[ \Big( \sum_{u \in D_k(\rho')} (\mathbb{E}_u \tilde{f}_u)(X) \Big) \Big( \sum_{u \in D_k(\rho')} (\mathbb{E}_u f_{m,u})(X) \Big) \Big] \Big| = \Big| \mathbb{E} \Big[ (\mathbb{E}_k f_k)(X) (\mathbb{E}_k f_m)(X) \Big] \Big|$$
$$\leq \sqrt{\mathbb{E} \big[ (\mathbb{E}_k f_k)^2(X) \big]} \sqrt{\mathbb{E} \big[ (\mathbb{E}_k f_m)^2(X) \big]}. \tag{73}$$

Now, combining (70) and (68), we have

$$\sqrt{\mathbb{E} \big[ (\mathbb{E}_k f_k)^2(X) \big]} \leq \sqrt{2} \exp \big( - \varepsilon \big( k - C_{E.1}(1 + \log(R)) - \mathrm{h}^* \big) \big) (\mathbb{E} f_k^2(X))^{1/2}. \tag{74}$$

By setting

$$C_1 = C_{E.1} + \frac{1}{\varepsilon} \Big( \frac{1}{2} \log(2) + \log(2 C_{D.1}) + 2 \Big),$$

we can conclude that

$$\Big| \mathbb{E} \Big[ \Big( \sum_{u \in D_k(\rho')} (\mathbb{E}_u \tilde{f}_u)(X) \Big) \Big( \sum_{u \in D_k(\rho')} (\mathbb{E}_u f_{m,u})(X) \Big) \Big] \Big|$$
$$\leq \exp(-\varepsilon(k - C_1(\log(R) + 1) - \mathrm{h}^*)) \sqrt{\mathbb{E} f_k^2(X) \mathbb{E} f_m^2(X)}, \tag{75}$$

**Part 3.2: Estimating second summand of** (72) For the second summand of (72), we begin with the estimate for each $u \in D_k(\rho')$:

$$\mathbb{E} |\tilde{f}_u(X) f_{m,u}(X)| \leq \sum_{i \in [d_u]} \mathbb{E} |\tilde{f}_u(X) f_{m,u_i}(X)|.$$

Since for each $i \in [d_u]$ we have $f_{m,u_i}(x) = f_{m,u_i}(x_{\leq u_i})$, we apply (39) from Proposition E.1 to $\tilde{f}_u$ and $a(x) = f_{m,u_i}(x)$ to get

$$\sum_{i \in [d_u]} \mathbb{E}|\tilde{f}_u(X)f_{m,u_i}(X)| \leq \sum_{i \in [d_u]} \exp\Big(-\frac{\varepsilon}{2}\big(k - C_{E.1}(\log(R)+1) - \mathrm{h}^*\big)\Big)(\mathbb{E}\tilde{f}_u^2(X))^{1/2}(\mathbb{E}f_{m,u_i}^2(X))^{1/2},$$

where $C_{E.1}$ is the constant introduced in the Proposition. Applying Jensen's inequality and (71) with $w = u$ and $k' = k - 1$,

$$\sum_{i \in [d_u]} (\mathbb{E}f_{m,u_i}^2(X))^{1/2} \leq d_u^{1/2}\Big(\sum_{i \in [d_u]} \mathbb{E}f_{m,u_i}^2(X)\Big)^{1/2} \leq (Rd)^{1/2}C_{D.1}R^2\big(\mathbb{E}f_{m,u}^2(X)\big)^{1/2}.$$

Hence,

$$\mathbb{E}|\tilde{f}_u(X)f_{m,u}(X)| \leq (Rd)^{1/2}C_{D.1}R^2 \exp\Big(-\frac{\varepsilon}{2}\big(k - C_{E.1}(\log(R)+1) - \mathrm{h}^*\big)\Big)(\mathbb{E}\tilde{f}_u^2(X))^{1/2}\big(\mathbb{E}f_{m,u}^2(X)\big)^{1/2}$$

$$\leq \exp\Big(-\frac{\varepsilon}{2}\big(k - C_2(\log(R)+1) - \mathrm{h}^*\big)\Big)(\mathbb{E}\tilde{f}_u^2(X))^{1/2}\big(\mathbb{E}f_{m,u}^2(X)\big)^{1/2}, \quad (76)$$

where

$$C_2 = \frac{2}{\varepsilon}\Big(\frac{3}{2} + \frac{1}{2}\log(d) + \log(C_{D.1})\Big) + C_{E.1}.$$

Now, returning to the summation, we apply (76) and Cauchy-Schwarz inequality to get

$$\Big|\sum_{u \in D_k(\rho')} \mathbb{E}\tilde{f}_u(X)f_{m,u}(X)\Big|$$

$$\leq \sum_{u \in D_k(\rho')} \exp\Big(-\frac{\varepsilon}{2}(k - C_2(\log(R)+1) - \mathrm{h}^*)\Big)(\mathbb{E}\tilde{f}_u^2(X))^{1/2}(\mathbb{E}f_{m,u}^2(X))^{1/2}$$

$$\leq \exp\Big(-\frac{\varepsilon}{2}(k - C_2(\log(R)+1) - \mathrm{h}^*)\Big)\Big(\sum_{u \in D_k(\rho')} \mathbb{E}\tilde{f}_u^2(X)\Big)^{1/2}\Big(\sum_{u \in D_k(\rho')} \mathbb{E}f_{m,u}^2(X)\Big)^{1/2}$$

$$\leq 2C_{D.1}R^2 \exp\Big(-\frac{\varepsilon}{2}(k - C_2(\log(R)+1) - \mathrm{h}^*)\Big)\big(\mathbb{E}f_k^2(X)\big)^{1/2}\big(\mathbb{E}f_m^2(X)\big)^{1/2}$$

$$\leq \exp\Big(-\frac{\varepsilon}{2}(k - C_3(\log(R)+1) - \mathrm{h}^*)\Big)\sqrt{\mathbb{E}f_k^2(X)\mathbb{E}f_m^2(X)}. \quad (77)$$

In the derivation above, we applied Lemma F.2 for the term $\Big(\sum_{u \in D_k(\rho')} \mathbb{E}\tilde{f}_u^2(X)\Big)^{1/2}$ and (71) for the term $\Big(\sum_{u \in D_k(\rho')} \mathbb{E}f_{m,u}^2(X)\Big)^{1/2}$ in the secont to last inequality. The constant $C_2$ in the last inequality is defined as

$$C_3 = \frac{2}{\varepsilon}(\log(2C_{D.1}) + 2) + C_2.$$

**Part 3.3: Estimating third summand of** (72) It remains to bound the third summand, it can be reduced to the upper bound for first summand. Applying the Cauchy-Schwarz inequality and Hölder's inequality we have

$$\Big|\sum_{u \in D_k(\rho')} \mathbb{E}\Big[(\mathbb{E}_u\tilde{f}_u)(X)(\mathbb{E}_u f_{m,u})(X)\Big]\Big|$$

$$\leq \mathbb{E}\Big[\Big|\sum_{u \in D_k(\rho')} (\mathbb{E}_u\tilde{f}_u)(X)(\mathbb{E}_u f_{m,u})(X)\Big|\Big]$$

$$\leq \mathbb{E}\Big[\Big(\sum_{u \in D_k(\rho')} (\mathbb{E}_u\tilde{f}_u)^2(X)\Big)^{1/2}\Big(\sum_{u \in D_k(\rho')} (\mathbb{E}_u f_{m,u})^2(X)\Big)^{1/2}\Big]$$

$$\leq \Big(\mathbb{E}\sum_{u \in D_k(\rho')} (\mathbb{E}_u\tilde{f}_u)^2(X)\Big)^{1/2}\Big(\mathbb{E}\sum_{u \in D_k(\rho')} (\mathbb{E}_u f_{m,u})^2(X)\Big)^{1/2}$$

$$\leq \sqrt{2}\exp\big(-\varepsilon\big(k - C_{E.1}(1 + \log(R)) - \mathrm{h}^*\big)\big) \cdot C_{D.1}R^2\sqrt{\mathbb{E}f_k^2(X)\mathbb{E}f_m^2(X)}.$$

where in the last inequality we applied (74) and (71).

By setting
$$C_4 = \frac{1}{\varepsilon}\left(\frac{1}{2}\log(2) + \log(C_{D.1}) + 2\right),$$
we conclude that
$$\left|\sum_{u \in D_k(\rho')} \mathbb{E}\left[(\mathbb{E}_u \tilde{f}_u)(X)(\mathbb{E}_u f_{m,u})(X)\right]\right| \le \exp\left(-\varepsilon\left(k - C_4(1 + \log(R)) - \mathrm{h}^*\right)\right)\sqrt{\mathbb{E}f_k^2(X)\mathbb{E}f_m^2(X)}.$$
$$(78)$$

Now, combining the three estimates of the summands (75), (77), and (78) for (72) we conclude that
$$|\mathbb{E}f_k(X)f_m(X)| \le \exp\left(-\frac{\varepsilon}{2}\left(k - C_5(\log(R) + 1) - \mathrm{h}^*\right)\right)\sqrt{\mathbb{E}f_k^2(X)\mathbb{E}f_m^2(X)},$$
where
$$C_5 := \frac{2}{\varepsilon}\log(3) + \max\{C_1, C_3, C_4\}.$$

Now we impose the **forth assumption** on $C_0$ that
$$C_0 \ge C_5,$$
and (64) follows.

$\square$

## F.2 Proof of Theorem B.6

Let $C_0 = C_0(M, d, c^*)$ denote the constant introduced in the statement of the Lemma. Its precise value will be determined along the proof.

Let $k_1$ be a positive integer with the precise value to be determined later. Here we impose our **first assumption** on $k_1$ that
$$k_1 \ge C_{F.1}(\log(R) + 1) + \mathrm{h}^*$$
where $C_{F.1}$ is a constant that appears in Proposition F.1.

Now, let us consider a function $f$ described in the Theorem. Without lose of generality, we may assume $\mathbb{E}f(X) = 0$. Then, it is equivalent to estimate the second moments.

Further, let us assume $\mathrm{h}(\rho') \ge k_1$ and the decomposition of $f$ according to Lemma D.1:
$$f(x) = \sum_{k \in [k_1, \mathrm{h}(\rho')]} f_k(X). \tag{79}$$

Our first goal is to show
$$\mathbb{E}f(X)^2 \simeq \sum_{k \in [k_1, \mathrm{h}(\rho')]} f_k^2(X),$$
by showing $\mathbb{E}f_k(X)f_m(X)$ is insignificant whenever $k \ne m$.

For $k_1 \le m < k$, by Propostion F.1,
$$2|\mathbb{E}f_k(X)f_m(X)| \le 2\exp\left(-\frac{\varepsilon}{2}\left(k - C_{F.1}(\log(R) + 1) - \mathrm{h}^*\right)\right)(\mathbb{E}f_k^2(X))^{1/2}(\mathbb{E}f_m^2(X))^{1/2}$$
$$\le \exp\left(-\frac{\varepsilon}{2}\left(k - C_{F.1}(\log(R) + 1) - \mathrm{h}^*\right)\right)\mathbb{E}f_k^2(X)$$
$$+ \exp\left(-\frac{\varepsilon}{2}\left(k - C_{F.1}(\log(R) + 1) - \mathrm{h}^*\right)\right)\mathbb{E}f_m^2(X).$$

Applying the above inequality to bound the second moment of $f(X)$ we get
$$\mathbb{E}f^2(X) = \mathbb{E}\sum_{k,m \in [k_1, \mathrm{h}(\rho')]} \mathbb{E}f_k(X)f_m(X)$$
$$\ge \sum_{k \in [k_1, \mathrm{h}(\rho')]} \mathbb{E}f_k^2(X) \cdot \left(1 - \sum_{s \in [k_1, \mathrm{h}(\rho')]} \exp\left(-\frac{\varepsilon}{2}\left(s - C_{F.1}(\log(R) + 1) - \mathrm{h}^*\right)\right)\right).$$

Notice that there exists $t_0$ which depends on $\varepsilon$ so that

$$\sum_{t=t_0}^{\infty} \exp\left(-\frac{\varepsilon}{2}t\right) \leq \frac{1}{2}.$$

By setting

$$k_1 := \lceil \mathrm{h}^* + C_{F.1}(\log(R) + 1) + t_0 \rceil,$$

we get

$$\mathbb{E}f^2(X) \geq \frac{1}{2} \sum_{k \in [k_1, \mathrm{h}(\rho')]} \mathbb{E}f_k^2(X). \tag{80}$$

Our second goal is comparing $\mathbb{E}\left[(\mathbb{E}_{\rho'} f_k)^2 (X_{\rho'})\right]$ and $\sum_{k \in [k_1, \infty]} \mathbb{E}f_k^2(X)$. Starting with the variance and $\ell_\infty$ norm comparison from Lemma A.5,

$$
\begin{aligned}
\mathbb{E}\left[(\mathbb{E}_{\rho'} f)^2 (X_{\rho'})\right] \leq & \mathbb{E}\left[\left(\sum_{k \in [k_1, \mathrm{h}(\rho')]} \max_{\theta_k \in [q]} \left|(\mathbb{E}_{\rho'} f_k)(\theta_k)\right|\right)^2\right] \\
= & \left(\sum_{k \in [k_1, \mathrm{h}(\rho')]} \max_{\theta_k \in [q]} \left|(\mathbb{E}_{\rho'} f_k)(\theta_k)\right|\right)^2 \\
\leq & C_{A.5}\left(\sum_{k \in [k_1, \mathrm{h}(\rho')]} \sqrt{\mathbb{E}(\mathbb{E}_{\rho'} f_k)^2(X_\rho)}\right)^2,
\end{aligned}
$$

where $C_{A.5}$ is the constant introduced in the Lemma.

By (62), from Proposition F.1, for $k \in [k_1 + 1, \mathrm{h}(\rho')]$,

$$\mathbb{E}\left[(\mathbb{E}_{\rho'} f_k)^2 (X_{\rho'})\right] \leq \exp\left(-\varepsilon(k - \mathrm{h}^*)\right) \cdot \exp\left(-\varepsilon(\mathrm{h}(\rho') - C_{F.1}(\log(R) + 1) - \mathrm{h}^*)\right) \mathbb{E}f_k^2(X)$$

And for $k = k_1 = \lceil \mathrm{h}^* + C_{F.1}(\log(R) + 1) + t_0 \rceil$, we apply (63) to get

$$
\begin{aligned}
\mathbb{E}\left[(\mathbb{E}_{\rho'} f_{k_1})^2 (X_{\rho'})\right] \leq & \exp\left(-\varepsilon(\mathrm{h}(\rho') - k_1 - C_{F.1}(\log(R) + 1))\right) \mathbb{E}f_{k_1}^2(X) \\
\leq & \exp\left(-\varepsilon(-t_0 - 1)\right) \\
& \cdot \exp\left(-\varepsilon(\mathrm{h}(\rho') - 2C_{F.1}(\log(R) + 1) - \mathrm{h}^*)\right) \mathbb{E}f_{k_1}^2(X).
\end{aligned}
$$

Substituting these estimate and by Cauchy-Schwarz inequality we have

$$
\begin{aligned}
\mathbb{E}\left[(\mathbb{E}_{\rho'} f)^2 (X_{\rho'})\right] \leq & C_{A.5} \exp\left(-\varepsilon(\mathrm{h}(\rho') - 2C_{F.1}(\log(R) + 1) - \mathrm{h}^*)\right) \\
& \cdot \underbrace{\left(\exp(\varepsilon(t_0 + 1)) + \sum_{t=0}^{\infty} \exp(-\varepsilon t)\right)}_{:=C_1} \cdot \sum_{k \in [k_1, \mathrm{h}(\rho')]} \mathbb{E}f_k^2(X) \\
\leq & C_{A.5} C_1 2 \exp\left(-\varepsilon(\mathrm{h}(\rho') - 2C_{F.1}(\log(R) + 1) - \mathrm{h}^*)\right) \mathbb{E}f^2(X).
\end{aligned}
$$

Now, by taking

$$C_0 \geq \max\left\{2C_{F.1} + \frac{1}{\varepsilon}\log(C_{A.5} C_1 2), \, C_{F.1} + t_0 + 1\right\},$$

we conclude that

$$\mathbb{E}\left[(\mathbb{E}_{\rho'} f)^2 (X_{\rho'})\right] \leq \exp\left(-\varepsilon(\mathrm{h}(\rho') - C_0(\log(R) + 1) - \mathrm{h}^*)\right) \mathbb{E}f^2(X).$$

It remains to show the case when $\mathrm{h}(\rho') \leq k_1$. From the assumption that $C_0 \geq C_{F.1} + t_0 + 1$ and $k_1 \leq \mathrm{h}^* + C_{F.1}(\log(R) + 1) + t_0 + 1$, we have

$$\exp\left(-\varepsilon(\mathrm{h}(\rho') - C_0(\log(R) + 1) - \mathrm{h}^*\right) \geq 1.$$

Hence, the statement follows directly from Jensen's inequality.

# G General Case: Base Case

Now, we want to establish Theorem 1.6, which does not rely on the assumption $c_M > 0$. Let us first establish analogues of Assumption B.3 (the inductive assumption), Proposition B.5 (the base case), and Theorem B.1 (the inductive step) in the general case.

**Assumption G.1.** By stating that $\mathcal{A}$ satisfies this assumption with given parameter $\mathrm{h}^\circ$, we mean $\mathcal{A}_1 \subseteq \mathcal{A} \subseteq \mathbf{2}^L \setminus \{\emptyset\}$ is closed under decomposition, and the following holds:

For every $u \in T$ and any $\mathcal{A}_{\leq u}$-polynomials functions $f$ and $g$, we have

$$\mathrm{Var}\big[(\mathbb{E}_u f)(X)\big] \leq \exp(-\varepsilon(\mathrm{h}(u) - \mathrm{h}^\circ))\mathrm{Var}\big[f(X)\big]. \tag{81}$$

Further, suppose $\mathbb{E}f = \mathbb{E}g = 0$ and $\mathrm{h}(u) \geq \mathrm{h}^\circ$. Notice that by the Markov Property, $(\mathbb{E}_u fg)(x)$, $(\mathbb{E}_u f^2)(x)$, and $(\mathbb{E}_u g^2)(x)$ are functions of $x_u$. Then,

$$\max_{\theta \in [q]} |(\mathbb{E}_u fg)(\theta) - \mathbb{E}fg| \leq \exp\left(-\frac{\varepsilon}{2}(\mathrm{h}(u) - \mathrm{h}^\circ)\right)\sqrt{\min_\theta (\mathbb{E}_u f^2)(\theta) \min_{\theta'}(\mathbb{E}_u g^2)(\theta)}. \tag{82}$$

The main difference of this assumption and Assumption B.3 is the difference of (82) and (16).

**Proposition G.2.** *Consider the rooted tree $T$ and transition matrix $M$ described in Theorem 1.6. There exists $C = C(M, d) \geq 1$ such that $\mathcal{A}_1$ satisfies Assumption G.1 with some parameter $\mathrm{h}^\circ$ satisfying*

$$\mathrm{h}^\circ \leq C(\log(R) + 1). \tag{83}$$

**Theorem G.3.** *Consider the rooted tree $T$ and transition matrix $M$ described in Theorem 1.6. There exists $C = C(M, d) > 1$ so that the following holds. Suppose $\mathcal{A}$ satisfies Assumption G.1 with some parameter $\mathrm{h}^\circ$. Let $\mathcal{B} = \mathcal{B}(\mathcal{A})$ (see Definition 1.11). Then, $\mathcal{B}$ satisfies Assumption G.1 with parameter*

$$\mathrm{h}^\circ + C(\log(R) + 1).$$

*Proof of Theorem 1.6.* The proof of Theorem 1.6 is analogous to that of Theorem B.1, employing a similar strategy by leveraging Proposition G.2 and Theorem G.3 in the former, and Proposition B.5 and Theorem B.6 in the latter.

$\square$

In this section we will prove the Base Case Proposition G.2.

**Lemma G.4.** *There exists a constant $C = C(M, \varepsilon) \geq 1$ so such that for any $\rho' \in T$ and $0 \leq m \leq \mathrm{h}(\rho')$:*

*Consider two degree 1 polynomials $f$ and $g$ with variables $(x_u : u \in D_m(\rho'))$. Suppose*

$$f(X) = \sum_{u \in D_m(\rho')} f_u(X) \text{ almost surely,}$$

*where $f_u(x) = f_u(x_u)$ and $\mathbb{E}[f_u(X)] = 0$, and we assume the same conditions for the polynomial $g$ and $g_u$. Then,*

$$\max_{\theta \in [q]} \big|(\mathbb{E}_{\rho'} fg)(\theta) - \mathbb{E}fg\big| \leq CR \exp(-\varepsilon(\mathrm{h}(\rho') - m))\sqrt{\sum_{u \in D_m(\rho')} \mathbb{E}f_u^2(X)}\sqrt{\sum_{u \in D_m(\rho')} \mathbb{E}g_u^2(X)}.$$

*Proof.* Let $C_0 = C_0(M, d)$ denote the constant introduced in the statement of the Lemma. Its value will be determined along the proof.

First of all,

$$\max_{\theta \in [q]} \big|(\mathbb{E}_{\rho'} fg)(\theta) - \mathbb{E}fg\big| = \max_{\theta \in [q]} \bigg|\sum_{u,v \in D_m(\rho')} \big((\mathbb{E}_{\rho'} f_u g_v)(\theta) - \mathbb{E}f_u g_v\big|\bigg)$$

$$\leq \sum_{u,v \in D_m(\rho')} \max_{\theta \in [q]} \big|(\mathbb{E}_{\rho'} f_u g_v)(\theta) - \mathbb{E}f_u g_v\big|.$$

Our proof will be carried out by estimating each summand. Fix any pair $u, v \in D_m(\rho')$ and consider

$$\|(\mathbb{E}_{\rho'} f_u g_v) - \mathbb{E} f_u g_v\|_\infty = \max_\theta \left| \mathbb{E}_{\rho'} \left[ f_u(X) g_v(X) - \mathbb{E} f_u g_v \,\middle|\, X_{\rho'} = \theta \right] \right|.$$

Let $w = \rho(u, v)$. Since $f_u$ and $g_v$ are functions of $x_{\leq w}$, relying on the Markov Property we know the function $(\mathbb{E}_w f_u \cdot g_v)(x)$ is a function of $x_w$ with expected value $\mathbb{E} f_u(X) g_v(X)$. With

$$(\mathbb{E}_{\rho'} f_u g_v)(x_{\rho'}) = \mathbb{E}\left[ (\mathbb{E}_w f_u g_v)(X_w) \,\middle|\, X_{\rho'} = x_{\rho'} \right],$$

applying (10) from Lemma A.5,

$$\|(\mathbb{E}_{\rho'} f_u g_v) - \mathbb{E} f_u g_v\|_\infty \leq C_{A.5}(\mathrm{h}(\rho') - \mathrm{h}(w))^q \lambda^{\mathrm{h}(\rho') - \mathrm{h}(w)} \|(\mathbb{E}_w f_u g_v)(\theta) - \mathbb{E} f_u g_v\|_\infty,$$

where $C_{A.5}$ is the $M$-dependent constant introduced in the Lemma.

Next, we will estimate $\|(\mathbb{E}_w f_u g_v)(\theta) - \mathbb{E} f_u g_v\|_\infty$. In the case $u \neq v$, there exists $i \neq j$ such that $u \leq w_i$ and $v \leq w_j$, which in turn implies that $(X_{\leq u} \mid X_w = x_w)$ and $(X_{\leq v} \mid X_w = x_w)$ are jointly independent by the Markov Property. Thus,

$$(\mathbb{E}_w f_u g_v)(\theta) = (\mathbb{E}_w f_u)(\theta)(\mathbb{E}_w g_v)(\theta),$$

which implies

$$\max_{\theta \in [q]} |(\mathbb{E}_w f_u g_v)(\theta)| \leq \max_{\theta \in [q]} |(\mathbb{E}_w f_u)(\theta)| \cdot \max_{\theta \in [q]} |(\mathbb{E}_w g_v)(\theta)|$$

$$\leq C_{A.5}^3 (\mathrm{h}(w) - m)^{2q} \lambda^{2(\mathrm{h}(w) - m)} \sqrt{\mathbb{E} f_u^2(X) \mathbb{E} g_v^2(X)},$$

where we applied (10) and (9) from Lemma A.5 in the last inequality. If $u = v$, then the same estimate follows immediately without relying on (10).

Now, we convert the above estimate to that of $\|(\mathbb{E}_w f_u g_v)(\theta) - \mathbb{E} f_u g_v\|_\infty$, which relies on the simple bound that $|\mathbb{E} f_u(X) g_v(X)| \leq \max_{\theta \in [q]} |(\mathbb{E}_w f_u g_v)(\theta)|$. Thus,

$$\|(\mathbb{E}_w f_u g_v)(\theta) - \mathbb{E} f_u g_v\|_\infty \leq 2 \max_{\theta \in [q]} |(\mathbb{E}_w f_u g_v)(\theta)|$$

$$\leq 2 C_{A.5}^3 (\mathrm{h}(w) - m)^{2q} \lambda^{2(\mathrm{h}(w) - m)} \sqrt{\mathbb{E} f_u^2(X) \mathbb{E} g_v^2(X)}.$$

Together we conclude that for a pair $u, v \in D_m(\rho')$ with $w = \rho(u, v)$,

$$\|(\mathbb{E}_{\rho'} f_u g_v) - \mathbb{E} f_u g_v\|_\infty \leq 2 C_{A.5}^4 (\mathrm{h}(w) - m)^{2q} (\mathrm{h}(\rho') - \mathrm{h}(w))^q \lambda^{\mathrm{h}(\rho') + \mathrm{h}(w) - 2m} \sqrt{\mathbb{E} f_u^2(X) g_v^2(X)}$$

$$\leq 2 C_{A.5}^4 (\mathrm{h}(\rho') - m)^{3q} \lambda^{\mathrm{h}(\rho') + \mathrm{h}(w) - 2m} \sqrt{\mathbb{E} f_u^2(X) g_v^2(X)}.$$

Relying on this estimate, we are ready to bound the $l_\infty$-norm of $(\mathbb{E}_{\rho'} fg)(x_{\rho'}) - \mathbb{E} fg$.

$$\max_{\theta \in [q]} \left| (\mathbb{E}_{\rho'} fg)(\theta) - \mathbb{E} fg \right|$$

$$\leq \sum_{u, v \in D_m(\rho')} \max_{\theta \in [q]} \left| (\mathbb{E}_{\rho'} f_u g_v)(\theta) - \mathbb{E} f_u g_v \right|$$

$$= \sum_{k \in [m, \mathrm{h}(\rho')]} \sum_{w \in D_k(\rho')} \sum_{u, v \,:\, \rho(u,v) = w} \max_{\theta \in [q]} \left| (\mathbb{E}_\rho f_u g_v)(\theta) - \mathbb{E} f_u g_v \right|$$

$$\leq \sum_{k \in [m, \mathrm{h}(\rho')]} \sum_{w \in D_k(\rho')} \sum_{u, v \,:\, \rho(u,v) = w} 2 C_{A.5}^4 (\mathrm{h}(\rho') - m)^{3q} \lambda^{\mathrm{h}(\rho') + k - 2m} \sqrt{\mathbb{E} f_u^2(X) g_v^2(X)}. \tag{84}$$

Next, relaxing the condition $\rho(u, v) = w$ in the summation,

$$(*) \leq \sum_{k \in [m, \mathrm{h}(\rho')]} \sum_{w \in D_k(\rho')} \sum_{u, v \in D_m(w)} 2 C_{A.5}^4 (\mathrm{h}(\rho') - m)^{3q} \lambda^{\mathrm{h}(\rho') + k - 2m} \sqrt{\mathbb{E} f_u^2(X) g_v^2(X)}$$

$$= \sum_{k \in [m, \mathrm{h}(\rho')]} \sum_{w \in D_k(\rho')} 2 C_{A.5}^4 (\mathrm{h}(\rho') - m)^{3q} \lambda^{\mathrm{h}(\rho') + k - 2m} \left( \sum_{u \in D_m(w)} \sqrt{\mathbb{E} f_u^2(X)} \right) \left( \sum_{u \in D_m(w)} \sqrt{\mathbb{E} g_u^2(X)} \right).$$

Notice the inequality $\sum_{i\in[n]} \frac{|t_i|}{n} \leq \sqrt{\sum_{i\in[n]} \frac{|t_i|^2}{n}}$ follows from Jenson's inequality applying to the function $t \mapsto t^2$ and the uniform measure on $[n]$. Now apply this inequality to the collection $\{\sqrt{\mathbb{E}f_u^2(X)}\}$ and $\{\sqrt{\mathbb{E}g_u^2(X)}\}$ respectively, together with $|D_m(w)| \leq Rd^{\mathrm{h}(w)-m}$, from our tree asscumption, we have

$$(*) \leq \sum_{k\in[m,\mathrm{h}(\rho')]} \sum_{w\leq\rho'\,:\,w\in D_k(\rho')} 2C_{A.5}^4 (\mathrm{h}(\rho')-m)^{3q} \lambda^{\mathrm{h}(\rho')+k-2m} Rd^{k-m}$$

$$\cdot \sqrt{\sum_{u\in D_m(w)} \mathbb{E}f_u^2(X)} \sqrt{\sum_{u\in D_m(w)} \mathbb{E}g_u^2(X)}$$

$$\leq \sum_{k\in[m,\mathrm{h}(\rho')]} 2C_{A.5}^4 (\mathrm{h}(\rho')-m)^{3q} \lambda^{\mathrm{h}(\rho')+k-2m} Rd^{k-m}$$

$$\cdot \sqrt{\sum_{w\leq\rho'\,:\,\mathrm{h}(w)=k} \sum_{u\in D_m(w)} \mathbb{E}f_u^2(X)} \cdot \sqrt{\sum_{w\leq\rho'\,:\,\mathrm{h}(w)=k} \sum_{u\in D_m(w)} \mathbb{E}g_u^2(X)}$$

$$= \sum_{k\in[m,\mathrm{h}(\rho')]} 2C_{A.5}^4 (\mathrm{h}(\rho')-m)^{3q} \lambda^{\mathrm{h}(\rho')+k-2m} Rd^{k-m} \sqrt{\sum_{u\in D_m(\rho')} \mathbb{E}f_u^2(X)} \sqrt{\sum_{u\in D_m(\rho')} \mathbb{E}g_u^2(X)},$$
$$(85)$$

where the last inequality follows from Cauchy-Schwarz inequality. Finally,

$$\sum_{k\in[m,\mathrm{h}(\rho')]} 2C_{A.5}^4 (\mathrm{h}(\rho')-m)^{3q} \lambda^{\mathrm{h}(\rho')+k-2m} Rd^{k-m}$$

$$\leq 2C_{A.5}^4 R(\mathrm{h}(\rho')-m)^{3q} \cdot (\mathrm{h}(\rho')-m)\lambda^{\mathrm{h}(\rho')-m} \max_{k\in[m,\mathrm{h}(\rho')]} \lambda^{k-m} d^{k-m}$$

$$= 2C_{A.5}^4 R(\mathrm{h}(\rho')-m)^{3q} \cdot (\mathrm{h}(\rho')-m)\big(\max\{d\lambda^2,\,\lambda\}\big)^{\mathrm{h}(\rho')-m}$$

$$= 2C_{A.5}^4 R(\mathrm{h}(\rho')-m)^{3q+1} \exp(-1.1\varepsilon(\mathrm{h}(\rho')-m))$$

$$\leq C_0 R \exp(-\varepsilon(\mathrm{h}(\rho')-m)),$$

where

$$C_0 = 2C_{A.5}^4 \max_{n\in\mathbb{N}} n^{3q+1} \exp(-0.1\varepsilon n) < +\infty$$

is a constant depending on $M$ and $\varepsilon$. Combining the above estimate with (85) we conclude that

$$\max_{\theta\in[q]} \big|(\mathbb{E}_{\rho'} fg)(\theta) - \mathbb{E}fg\big| \leq C_0 R \exp\big(-\varepsilon(\mathrm{h}(\rho')-m)\big) \sqrt{\sum_{u\in D_m(\rho')} \mathbb{E}f_u^2(X)} \sqrt{\sum_{u\in D_m(\rho')} \mathbb{E}g_u^2(X)},$$

and the lemma follows. $\qquad\square$

The statement of Lemma G.4 together with Proposition C.3 implies the following:

**Corollary G.5.** *There exists a constant $C = C(M,\varepsilon) \geq 1$ so that the following holds. For $\rho' \in T$ and $0 \leq m \leq \mathrm{h}(\rho')$, consider two degree 1 polynomials $f$ and $g$ with variables $(x_u\,:\,u\in D_m(\rho'))$ with $\mathbb{E}f(X) = \mathbb{E}g(X) = 0$. Notice that by the Markov Property, $(\mathbb{E}_{\rho'} fg)(x)$ is a function of $x_{\rho'}$. Then,*

$$\max_{\theta\in[q]} \big|(\mathbb{E}_{\rho'} fg)(\theta) - \mathbb{E}fg\big| \leq CR^4 \exp(-\varepsilon(\mathrm{h}(\rho')-m)) \sqrt{\mathbb{E}f^2(X)} \sqrt{\mathbb{E}g^2(X)}.$$

**Remark G.6.** By taking the degree 1 polynomial $f = g$ with the assumption that $\mathbb{E}f(X) = 0$, we get

$$\mathbb{E}\big[(\mathbb{E}_{\rho'} f^2)(X) - \mathbb{E}f^2(X)\big]^2 \leq \Big(\max_{\theta\in[q]} \big|(\mathbb{E}_{\rho'} fg)(\theta) - \mathbb{E}fg\big|\Big)^2 \leq C^2 R^6 \exp(-2\varepsilon\mathrm{h}(\rho'))(\mathbb{E}f^2(X))^2.$$
$$(86)$$

In other words, if $\mathrm{h}(\rho')$ is sufficiently large, $(\mathbb{E}_{\rho'} f^2)(X_{\rho'})$ is almost the same as $\mathbb{E}f^2(X)$ with a small fluctuation. Let us state this as a seperate lemma.

**Lemma G.7.** *There exists $C = C(M, d)$ so that the following holds. For $\rho' \in T$ with*

$$\mathrm{h}(\rho') \geq \frac{C(\log(R) + 1)}{\varepsilon},$$

*any degree 1 polynomial $f$ of variables $(x_u \; : \; u \in L_{\rho'})$ with $\mathbb{E}f(X) = 0$ satisfies*

$$\max_{\theta \in [q]} (\mathbb{E}_{\rho'} f^2)(\theta) \leq 2 \min_{\theta \in [q]} (\mathbb{E}_{\rho'} f^2)(\theta).$$

*Proof.* By Corollary G.5, for every $\theta \in [q]$,

$$\left| (\mathbb{E}_{\rho'} f^2)(\theta) - \mathbb{E}f^2(X) \right| \leq C_{G.5} R^4 \exp(-\varepsilon \mathrm{h}(\rho')) \mathbb{E}f^2(X).$$

where $C_{G.5}$ is the constant introduced in Lemma G.5. Now, we set the constant described in the lemma as

$$C = \frac{1}{\varepsilon} \Big( \log(4 C_{G.5}) + 4 \Big),$$

which implies

$$C_{G.4} R \exp(-\varepsilon \mathrm{h}(\rho')) \leq \frac{1}{4} \exp \big( - \varepsilon(\mathrm{h}(\rho') - C(\log(R) + 1)) \big).$$

Then, with $\mathrm{h}(\rho') \geq C(\log(R) + 1)$

$$|(\mathbb{E}_{\rho'} f^2)(\theta) - \mathbb{E}f^2(X)| \leq \frac{1}{4} \mathbb{E}f^2(X),$$

which in term implies

$$\frac{\max_{\theta \in [q]} (\mathbb{E}_{\rho'} f^2)(\theta)}{\min_{\theta \in [q]} (\mathbb{E}_{\rho'} f^2)(\theta)} \leq \frac{\frac{5}{4} \mathbb{E}f^2(X)}{\frac{3}{4} \mathbb{E}f^2(X)} < 2.$$

$\square$

*Proof of Proposition G.2.* Let $C_0$ denote the constant introduced in the statement of the Proposition. Its precise value will be determined along the proof.

Let $\rho' \in T$ with $\mathrm{h}' := \mathrm{h}(\rho')$. By Lemma C.9, any degree-1 polynomial $f(x)$ with variables $(x_u \; : \; u \in L_{\rho'})$ satisfies

$$\mathrm{Var}\big[(\mathbb{E}_{\rho'} f)(X)\big] \leq C_{C.9} R^4 (\mathrm{h}')^{2q} (d\lambda^2)^{\mathrm{h}'} \mathrm{Var}[f(X)],$$

where $C_{C.9}$ denotes the $M$-dependent constant introduced in the Lemma. For the term in front of $\mathrm{Var}[f(X)]$,

$$C_{C.9} R^4 (\mathrm{h}')^{2q} (d\lambda^2)^{\mathrm{h}'} \leq C_{C.9} R^4 (\mathrm{h}')^{2q} \exp(-1.1\varepsilon \mathrm{h}') \leq \exp \big( - \varepsilon(\mathrm{h}' - C_1(\log(R) + 1)) \big),$$

where

$$C_1 := \frac{1}{\varepsilon} \Big( \log(C_{C.9}) + 4 + \max_{n \in \mathbb{N}} n^{2q} \exp(-0.1\varepsilon q) \Big).$$

Thus, if we impose the **first assumption** on $C_0$ that

$$C_0 \geq C_1,$$

then the first condition (81) in Assumption G.1 holds for $\mathcal{A}_1$ if we take $\mathrm{h}^\circ \geq C_0(1 + \log(R))$.

It remains to establish (82). Let $f, g$ be two degree-1 polynomials in the variables $(x_u \; : \; u \in L_{\rho'})$ satisfying $\mathbb{E}f(X) = \mathbb{E}g(X) = 0$. First, by Corollary G.5,

$$\max_{\theta \in [q]} |(\mathbb{E}_{\rho'} fg)(\theta) - \mathbb{E}fg| \leq C_{G.5} R^4 \exp(-\varepsilon \mathrm{h}(\rho')) \sqrt{\mathbb{E}f^2(X)} \sqrt{\mathbb{E}g^2(X)},$$

where $C_{G.5}$ is the constant introduced in Corollary G.5. Next, we would like to apply Lemma G.7. Assuming

$$\mathrm{h}(u) \geq \frac{C_{G.7}(\log(R) + 1)}{\varepsilon}$$

where $C_{G.7}$ is the constant introduced in the Lemma, we can apply the lemma to get

$$\mathbb{E}f^2(X) \leq 2\min_{\theta}(\mathbb{E}_{\rho'}f^2)(\theta)$$

and the same holds for $g$. Together we may conclude that

$$\max_{\theta \in [q]} |(\mathbb{E}_{\rho'}fg)(\theta) - \mathbb{E}fg| \leq 2C_{G.5}R^4 \exp(-\varepsilon\mathrm{h}(\rho'))\sqrt{\min_{\theta}(\mathbb{E}_{\rho'}f^2)(\theta)\min_{\theta}(\mathbb{E}_{\rho'}g^2)(\theta)}$$

Now, we impose the **second assumption** on $C_0$ that

$$C_0 \geq \max\left\{\frac{1}{\varepsilon}\Big(\log(2C_{G.5}) + 4\Big), \frac{C_{G.7}}{\varepsilon}\right\}.$$

Then, we conclude that

$$\max_{\theta \in [q]} |(\mathbb{E}_u fg)(\theta) - \mathbb{E}fg| \leq \exp\Big(-\varepsilon\big(\mathrm{h}(\rho') - C_0(\log(R) + 1)\big)\Big)\sqrt{\min_{\theta}(\mathbb{E}_u f^2)(\theta)\min_{\theta}(\mathbb{E}_u g^2)(\theta)}$$

provided that

$$\mathrm{h}(\rho') \geq C_0(\log(R) + 1).$$

Therefore, we can conclude that $\mathcal{A}_1$ satisfies Assumption G.1 with

$$\mathrm{h}^\circ = C_0(\log(R) + 1).$$

$\square$

# H  Inductive Step in General Case

The goal in this section is to prove Theorem G.3. Let us restate the theorem here:

**Theorem.** *Consider the rooted tree $T$ and transition matrix $M$ described in Theorem 1.6. There exists $C = C(M, d) > 1$ so that the following holds. Suppose $\mathcal{A}$ satisfies Assumption G.1 with some parameter $\mathrm{h}^\circ$. Let $\mathcal{B} = \mathcal{B}(\mathcal{A})$ (see Definition 1.11). Then, $\mathcal{B}$ satisfies Assumption G.1 with parameter*

$$\mathrm{h}^\circ + C(\log(R) + 1).$$

**In this section, we fix a subcollection $\mathcal{A}$ satisfying Assumption G.1 with a given parameter $\mathrm{h}^\circ$ and let $\mathcal{B} = \mathcal{B}(\mathcal{A})$.**

We begin with the following lemma, which allows us to recycle some of the results from the case $c_M > 0$.

**Lemma H.1.** *Suppose $\mathcal{A}$ satisfies Assumption G.1 with parameter $\mathrm{h}^\circ$. Then, then $\mathcal{A}$ satisfies Assumption B.3 with $\mathrm{h}^* = \mathrm{h}^\circ + \frac{2}{\varepsilon}\log(2)$ and $c^* = \frac{1}{2}$.*

*Proof.* Let $f$ be a $\mathcal{A}_{\leq v}$-polynomial. If we set $\mathrm{h}^* \geq \mathrm{h}^\circ$, then (15) follows immediately from (81).

Now, we assume that $\mathbb{E}f(X) = 0$ and $\mathrm{h}(v) \geq \mathrm{h}^\circ$. We could apply (82) with $g = f$ to get

$$\max_{\theta \in [q]} \big|(\mathbb{E}_v f^2)(\theta) - \mathbb{E}f^2(X)\big| \leq \exp\Big(-\frac{\varepsilon}{2}(\mathrm{h}(v) - \mathrm{h}^\circ)\Big)\mathbb{E}f^2(X).$$

If $\exp\Big(-\frac{\varepsilon}{2}(\mathrm{h}(v) - \mathrm{h}^\circ)\Big) \leq \frac{1}{2}$, or equivalently,

$$\mathrm{h}(v) \geq \mathrm{h}^\circ + \frac{2}{\varepsilon}\log(2),$$

then, for every $\theta \in [q]$,

$$\frac{1}{2}\mathbb{E}h^2(X) \leq (\mathbb{E}_v h^2)(\theta) \leq \frac{3}{2}\mathbb{E}h^2(X).$$

Therefore, if we set $\mathrm{h}^* \geq \mathrm{h}^\circ + \frac{2}{\varepsilon}\log(2)$ and $c^* = \frac{1}{2}$, both (15) and (16) hold. $\square$

In the remainning of this section, we set

$$\mathrm{h}^* = \mathrm{h}^\circ + \frac{2}{\varepsilon}\log(2) \text{ and } c^* = \frac{1}{2}, \tag{87}$$

and we will rely on the fact that $\mathcal{A}$ satisfies Assumption B.3 with these two parameters. In particular, we could apply Theorem B.6 to show the existence of $C_{B.6} = C(M, \varepsilon, 1/2)$ such that for any $\mathcal{B}_{\leq v}$-polynomial $f$,

$$\mathrm{Var}\big[(\mathbb{E}_v f)(X)\big] \leq \exp\Big( - \varepsilon\big(\mathrm{h}(v) - \mathrm{h}^\circ + C_{B.6}(\log(R) + 1)\big)\Big)\mathrm{Var}\big[f(X)\big].$$

Therefore, to establish Theorem G.3, it remains to show the existence of $C = C(M, d)$ so that any $\mathcal{B}_{\leq v}$-polynomials $f$ and $g$ with $\mathrm{h}(v) \geq \mathrm{h}^\circ + C(\log(R) + 1)$ and $\mathbb{E}f(X) = \mathbb{E}g(X) = 0$ satisfy

$$\max_{\theta \in [q]}|(\mathbb{E}_v fg)(\theta) - \mathbb{E}fg| \leq \exp\Big( - \frac{\varepsilon}{2}(\mathrm{h}(v) - \mathrm{h}^\circ - C(\log(R) + 1))\Big)\sqrt{\min_\theta(\mathbb{E}_v f^2)(\theta)\min_{\theta'}(\mathbb{E}_v g^2)(\theta)}.$$

To establish the above inequality, the higher level structure is essentially the same as that for deriving Theorem B.6. We again decompose $f$ and $g$ according to Lemma D.1. To the proof of the theorem, similarly it contains three steps:

1. Establish properties of $\tilde{f}_u$ and $\tilde{g}_u$, see Proposition H.2.
2. Estalbish properties of $f_k$ and $g_k$, see Proposition H.6.
3. Establish Theorem G.3.

## H.1 Properties of $f_u$

The main goal we want to prove in this subsection is the following Proposition.

**Proposition H.2.** *There exsits $C = C(M, d) \geq 1$ so that the following holds. For a given $u \in T\backslash L$ with*

$$\mathrm{h}(u) \geq \mathrm{h}^\circ + C(\log(R) + 1),$$

*suppose $f_u$ and $g_u$ are two functions which are linear combination of $\psi_\sigma(x)$ with $\sigma \in \mathcal{F}(\mathcal{B}_u)$. Then, for any $\theta, \theta' \in [q]$,*

$$\big|(\mathbb{E}_u f_u g_u)(\theta) - (\mathbb{E}_u f_u g_u)(\theta')\big|$$
$$\leq \exp\Big( - \frac{\varepsilon}{2}(\mathrm{h}(u) - C(\log(R) + 1) - \mathrm{h}^\circ)\Big)\sqrt{\min_\theta(\mathbb{E}_u f_u^2)(\theta)\min_\theta(\mathbb{E}_u g_u^2)(\theta)}.$$

With a minor modification to our approach, we are able to obtain an analogous result wherein $f_u$ and $g_u$ are substituted by $\tilde{f}_u$ and $\tilde{g}_u$, respectively:

**Corollary H.3.** *There exsits $C = C(M, d) \geq 1$ so that the following holds. For a given $u \in T\backslash L$ with*

$$\mathrm{h}(u) \geq \mathrm{h}^\circ + C(\log(R) + 1),$$

*suppose $f_u$ and $g_u$ are two functions which are linear combination of $\psi_\mathbf{S}(x)$ with $\mathbf{S} \in \mathcal{F}(\mathcal{B}_u)$. Then, for any $\theta, \theta' \in [q]$,*

$$\big|(\mathbb{E}_u \tilde{f}_u \tilde{g}_u)(\theta) - (\mathbb{E}_u \tilde{f}_u \tilde{g}_u)(\theta')\big|$$
$$\leq \exp\Big( - \frac{\varepsilon}{2}(\mathrm{h}(u) - C(\log(R) + 1) - \mathrm{h}^\circ)\Big)\sqrt{\min_\theta(\mathbb{E}_u \tilde{f}_u^2)(\theta)\min_\theta(\mathbb{E}_u \tilde{g}_u^2)(\theta)}.$$

Let us prove Corollary first.

*Proof.* Let $C_0$ denote the constant introduced in the Corollary. Its value will be dervied during the proof.

From the identity

$$\tilde{f}_u(x)\tilde{g}_u(x) = f_u(x)g_u(x) - f_u(x)\mathbb{E}g_u(X) - \mathbb{E}f_u(X)g_u(x) + \mathbb{E}f_u(X)\mathbb{E}g_u(X),$$

it follows that

$$\left|(\mathbb{E}_u \tilde{f}_u \tilde{g}_u)(\theta) - (\mathbb{E}_u \tilde{f}_u \tilde{g}_u)(\theta')\right| \leq \left|(\mathbb{E}_u f_u g_u)(\theta) - (\mathbb{E}_u f_u g_u)(\theta')\right| + |\mathbb{E} g_u(X)| \left|(\mathbb{E}_u f_u)(\theta) - (\mathbb{E}_u f_u)(\theta')\right|$$
$$+ |\mathbb{E} f_u(X)| \left|(\mathbb{E}_u g_u)(\theta) - (\mathbb{E}_u g_u)(\theta')\right|$$
$$\leq \left|(\mathbb{E}_u f_u g_u)(\theta) - (\mathbb{E}_u f_u g_u)(\theta')\right| + 4 \max_{\theta'} |\mathbb{E}_u f_u(\theta)| \max_{\theta'} |\mathbb{E}_u g_u(\theta)|.$$

First, we apply Propostion E.1 with the fact that $\mathcal{A}$ satisfies Assumption G.1 with parameter $h^* = h^\circ + \frac{2}{\varepsilon}\log(2)$ and $c^* = \frac{1}{2}$,

$$\max_{\theta'}(\mathbb{E}_u f_u)^2(\theta) \leq \exp(-2\varepsilon(h(u) - C_{E.1}(\log(R) + 1) - h^*)) \max_{\theta'}(\mathbb{E}_u f_u^2)(\theta')$$

where $C_{E.1} = C(M, d, \frac{1}{2})$ is the constant introduced in the Proposition.

Second, applying Proposition H.2 with $f_u = g_u$ we have

$$\left| \max_\theta (\mathbb{E}_u f_u^2)(\theta) - \min_{\theta'}(\mathbb{E}_u f_u^2)(\theta')\right|$$
$$\leq \exp\left( -\frac{\varepsilon}{2}(h(u) - C_{H.2}(\log(R) + 1) - h^\circ)\right) \min_\theta (\mathbb{E}_u f_u^2)(\theta)$$

where $C_{H.2}$ is the constant introduced in Proposition H.2.

Let us impose the **first assumption** that $C_0 \geq C_{H.2}$. Then, with $h(u) \geq h^\circ + C_0(\log(R) + 1)$, we can conclude that

$$\max_{\theta'}(\mathbb{E}_u f_u^2)(\theta') \leq 2 \min_{\theta'}(\mathbb{E}_u f_u^2)(\theta').$$

Clearly, the same derivation also holds for $g_u$.

Therefore, we conclude that

$$\left|(\mathbb{E}_u \tilde{f}_u \tilde{g}_u)(\theta) - (\mathbb{E}_u \tilde{f}_u \tilde{g}_u)(\theta')\right|$$
$$\leq \left|(\mathbb{E}_u f_u g_u)(\theta) - (\mathbb{E}_u f_u g_u)(\theta')\right|$$
$$+ 8\exp(-2\varepsilon(h(u) - C_1(\log(R) + 1) - h^*))\sqrt{\min_\theta(\mathbb{E}_u f_u^2)(\theta) \min_\theta(\mathbb{E}_u g_u^2)(\theta)}$$
$$\leq \exp\left( -\frac{\varepsilon}{2}(h(u) - C_{H.2}(\log(R) + 1) - h^\circ)\right)\sqrt{\min_\theta(\mathbb{E}_u f_u^2)(\theta) \min_\theta(\mathbb{E}_u g_u^2)(\theta)}$$
$$+ 8\exp\left( -2\varepsilon\left(h(u) - C_{E.1}(\log(R) + 1) - h^\circ - \frac{2}{\varepsilon}\log(2)\right)\right)\sqrt{\min_\theta(\mathbb{E}_u f_u^2)(\theta) \min_\theta(\mathbb{E}_u g_u^2)(\theta)}$$
$$\leq \exp\left( -\frac{\varepsilon}{2}(h(u) - C_0(\log(R) + 1) - h^\circ)\right)\sqrt{\min_\theta(\mathbb{E}_u f_u^2)(\theta) \min_\theta(\mathbb{E}_u g_u^2)(\theta)},$$

where the last inequality follows by imposing the **second assumption** on $C_0$ that

$$C_0 \geq \frac{2}{\varepsilon}\log(2) + \max\left\{ C_{H.2}, C_{E.1} + \frac{2}{\varepsilon}\log(2) + \frac{1}{2\varepsilon}\log(8)\right\}.$$

This completes the proof of the Corollary. $\qquad\square$

The main technical part for proving Proposition H.2 is the following:

**Lemma H.4.** *For any $u \in T$ with $\exp\left( -\frac{\varepsilon}{2}(h(u) - h^\circ)\right) \leq \frac{1}{4Rd}$, the following holds: Let $I \subset [d_u]$ be a subset of size at least 2. For any $a(x)$ and $b(x)$ which are linear combinations of $\psi_\sigma(x)$ with $\sigma \in \mathcal{B}_u$ satisfying $I(\sigma) = I$, we have*

$$\max_{\theta,\theta' \in [q]} \left|(\mathbb{E}_u ab)(\theta) - (\mathbb{E}_u ab)(\theta')\right| \leq 4dR\exp\left( -\frac{\varepsilon}{2}(h(u) - h^\circ)\right)\sqrt{\min_\theta(\mathbb{E}_u a^2)(\theta) \cdot \min_\theta(\mathbb{E}_u b^2)(\theta)}.$$

**Remark H.5.** From the assumption that $h(u)$ satisfies

$$4dR\exp\left( -\frac{\varepsilon}{2}(h(u) - h^\circ)\right) \leq 1 \Leftrightarrow h(u) \geq h^\circ + \frac{2}{\varepsilon}\log(4dR).$$

By taking $a(x) = b(x)$ we have

$$\max_\theta(\mathbb{E}_u a^2)(\theta) \leq 2\min_\theta(\mathbb{E}_u a^2)(\theta). \tag{88}$$

*Proof.* Let $u$, $a(x)$, and $b(x)$ be the vertex and functions described in the Lemma. Let us introduce some notations for the ease of expressing the calculation later. For brevity, let

$$\delta = \exp\Big(-\frac{\varepsilon}{2}(\mathrm{h}(u) - \mathrm{h}^\circ)\Big).$$

For $x \in [q]^T$, let

$$x_{u,I} = (x_{u_i})_{i \in I}.$$

For any given function $h(x)$ with variables in $(x_v \,:\, v \in \bigcup_{i \in I} T_{u_i})$, we define

$$(\mathbb{E}_{u,I} h)(x) := \mathbb{E}\Big[h(X) \,\Big|\, \forall v \notin \bigcup_{i \in I}\{w < u_i\}, \, X_v = x_v\Big].$$

Observe that

$$(\mathbb{E}_{u,I} a)(x), \; (\mathbb{E}_{u,I} b)(x), \text{ and } (\mathbb{E}_{u,I} ab)(x)$$

are functions with input $x_{u,I}$. This is due to the fact that $a$ and $b$ –and consequently $ab$– are functions of variables $(x_v \,:\, x_v \in \bigcup_{i \in I} L_{u_i})$ and Markov Property.

**Claim:** The function $x_{u,I} \mapsto (\mathbb{E}_{u,I} ab)(x_{u,I})$ is Lipschitz continuous with respect to the Hamming Distance with Lipschitz constant

$$2\delta \sqrt{\max_{x_{u,I}} (\mathbb{E}_{u,I} a^2)(x_{u,I})} \sqrt{\max_{x_{u,I}} (\mathbb{E}_{u,I} b^2)(x_{u,I})}. \tag{89}$$

We begin with the proof of the claim. Fix an index $i_0 \in I$. Without lose of generality, we assume $I = [k]$ and $i_0 = 1$. For $x \in [q]^T$, let

$$x_i = x_{\leq u_i}$$

for $i \leq [d_u]$, and set

$$x_0 = (x_2, \ldots, x_k).$$

With this notation above, we can express

$$a(x) = a(x_0, x_1) \qquad\qquad \text{and} \qquad\qquad b(x) = b(x_0, x_1).$$

Fix any value of $x_0$, the function

$$x_1 \mapsto a(x_0, x_1)$$

is a linear combination of $\tilde{\phi}_{\sigma_1}(x_1)$ with $\sigma_1 \in \mathcal{F}(\mathcal{A}_{\leq u})$. Notably, this implies that $\mathbb{E}a(x_0, X_1) = 0$. The same properties hold for the function $x_1 \mapsto b(x_0, x_1)$.

Now, given the assumption $\exp\big(-\frac{\varepsilon}{2}(\mathrm{h}(u) - \mathrm{h}^\circ)\big) \leq \frac{1}{4Rd}$ implies $\mathrm{h}(u) \geq \mathrm{h}^\circ$, we can apply (82) from Assumption G.1 to get that

$$\max_{\theta, \theta' \in [q]} \Big| \mathbb{E}\big[a(x_0, X_1)b(x_0, X_1) \,\big|\, X_{u_1} = \theta_1\big] - \mathbb{E}\big[a(x_0, X_1)b(x_0, X_1) \,\big|\, X_{u_1} = \theta_2\big]\Big|$$

$$\leq 2 \max_{\theta \in [q]} \Big| \mathbb{E}\big[a(x_0, X_1)b(x_0, X_1) \,\big|\, X_{u_1} = \theta_1\big] - \mathbb{E}a(x_0, X_1)b(x_0, X_1)\Big|$$

$$\leq 2\delta \sqrt{\min_\theta \mathbb{E}\big[a^2(x_0, X_1) \,\big|\, X_{u_1} = \theta\big] \min_\theta \mathbb{E}\big[b^2(x_0, X_1) \,\big|\, X_{u_1} = \theta\big]}.$$

For any $x \in [q]^T$, let $x_{u_0} = (x_{u_2}, x_{u_3}, \ldots, x_{u_{d_u}})$. By the Markov Property, for any $\theta \in [q]$,

$$(X_0 \,|\, X_{u_0} = x_{u_0}, X_{u_1} = \theta) = (X_0 \,|\, X_{u_0} = x_{u_0}) \text{ and}$$
$$(X_1 \,|\, X_{u_0} = x_{u_0}, X_{u_1} = \theta) = (X_{u_1} \,|\, X_{u_1} = \theta)$$

are jointly independent. Hence,

$$\mathbb{E}\big[a(X_0, X_1)b(X_0, X_1) \,\big|\, X_{u_0} = x_{u_0}, X_{u_1} = \theta\big]$$
$$= \mathbb{E}\big[a(Y_0, X_1)b(Y_0, X_1) \,\big|\, X_{u_1} = \theta\big]$$

where $Y_0$ is an independent copy of $(X_0 \mid X_{u_0} = x_{u_0})$. We have

$$\left| \mathbb{E}\big[a(X_0, X_1)b(X_0, X_1) \,\big|\, X_{u_0} = x_{u_0}, X_{u_1} = \theta\big] - \mathbb{E}\big[a(X_0, X_1)b(X_0, X_1) \,\big|\, X_{u_0} = x_{u_0}, X_{u_1} = \theta'\big] \right|$$

$$= \left| \mathbb{E}_{Y_0}\Big[ \mathbb{E}_{X_1}[a(Y_0, X_1)b(Y_0, X_1) \mid X_{u_1} = \theta] - \mathbb{E}_{X_1}[a(Y_0, X_1)b(Y_0, X_1) \mid X_{u_1} = \theta'] \Big] \right|$$

$$\leq \mathbb{E}_{Y_0}\Big[ \left| \mathbb{E}_{X_1}[a(Y_0, X_1)b(Y_0, X_1) \mid X_{u_1} = \theta] - \mathbb{E}_{X_1}[a(Y_0, X_1)b(Y_0, X_1) \mid X_{u_1} = \theta'] \right| \Big]$$

$$\leq 2\delta \mathbb{E}_{Y_0}\Big[ \big( \min_{\theta} \mathbb{E}_{X_1}[a^2(Y_0, X_1) \mid X_{u_1} = \theta] \big)^{1/2} \cdot \big( \min_{\theta'} \mathbb{E}_{X_1}[b^2(Y_0, X_1) \mid X_{u_1} = \theta'] \big)^{1/2} \Big]$$

$$\leq 2\delta \big( \mathbb{E}_{Y_0}\big[ \min_{\theta} \mathbb{E}_{X_1}[a^2(Y_0, X_1) \mid X_{u_1} = \theta] \big] \big)^{1/2} \cdot \big( \mathbb{E}_{Y_0}\big[ \min_{\theta'} \mathbb{E}_{X_1}[b^2(Y_0, X_1) \mid X_{u_1} = \theta'] \big] \big)^{1/2},$$

where the last inequality follows from Hölder's inequality. Further,

$$\mathbb{E}_{Y_0}\big[ \min_{\theta} \mathbb{E}_{X_1}[a^2(Y_0, X_1) \mid X_{u_1} = \theta] \big] \leq \min_{\theta} \mathbb{E}_{Y_0}\big[ \mathbb{E}_{X_1}[a^2(Y_0, X_1) \mid X_{u_1} = \theta] \big]$$

$$= \min_{\theta} \mathbb{E}\big[a^2(X) \mid X_{u_0} = x_{u_0}, X_{u_1} = \theta\big]$$

$$\leq \max_{x_{u,I}} (\mathbb{E}_{u,I} a^2)(x_{u,I}).$$

Applying the same derivation to $b$ we get

$$\mathbb{E}_{Y_0}\big[ \min_{\theta} \mathbb{E}_{X_1}[b^2(Y_0, X_1) \mid X_{u_1} = \theta] \big] \leq \max_{x_{u,I}} (\mathbb{E}_{u,I} b^2)(x_{u,I}).$$

Therefore, our claim (89) follows: For any $\theta, \theta' \in [q]$,

$$\left| \mathbb{E}\big[a(X)b(X) \,\big|\, X_{u_0} = x_{u_0}, x_{u_1} = \theta\big] - \mathbb{E}\big[a(X)b(X) \,\big|\, X_{u_0} = x_{u_0}, x_{u_1} = \theta'\big] \right|$$

$$\leq 2\delta \sqrt{\max_{x_{u,I}} (\mathbb{E}a^2)(x_{u,I}) \max_{x_{u,I}} (\mathbb{E}b^2)(x_{u,I})}.$$

With the Lipschitz continuity been established, essentially the lemma follows when $\delta$ is sufficiently small. Let us proceed with the remaining argument. Let

$$x'_{u,I} = \operatorname{argmin}_{x_{u,I}} (\mathbb{E}_{u,I} a^2)(x_{u,I}) \qquad \text{and} \qquad x''_{u,I} = \operatorname{argmax}_{x_{u,I}} (\mathbb{E}_{u,I} a^2)(x_{u,I}).$$

Applying (89) with the assumption $a(x) = b(x)$ and the fact $|I| \leq d_u$,

$$(\mathbb{E}_{u,I} a^2)(x''_{u,I}) - (\mathbb{E}_{u,I} a^2)(x'_{u,I}) \leq 2d_u \delta (\mathbb{E}_{u,I} a^2)(a''_{u,I}),$$

and hence

$$\max_{x_{u,I}} (\mathbb{E}_{u,I} a^2)(x_{u,I}) \leq \frac{1}{1 - 2d_u \delta} \min_{x_{u,I}} (\mathbb{E}_{u,I} a^2)(x_{u,I}) \leq \frac{1}{1 - 2d_u \delta} \min_{s} (\mathbb{E}_u a^2)(s), \qquad (90)$$

provided that $2d_u \delta < 1$.

Again, the same derivation also holds for $b$. Combining (89) and (90) we conclude that for any $\theta, \theta' \in [q]$,

$$|(\mathbb{E}_u ab)(\theta) - (\mathbb{E}_u ab)(\theta')| \leq |\max_{x_{u,I}} (\mathbb{E}_u ab)(x_{u,I}) - \min_{x'_{u,I}} (\mathbb{E}_u ab)(x'_{u,I})|$$

$$\leq \frac{2d_u \delta}{1 - 2d_u \delta} \sqrt{\min_{\theta} (\mathbb{E}_u a^2)(\theta) \min_{\theta'} (\mathbb{E}_u b^2)(\theta')}.$$

With our assumption on the tree $T$ that $d_u \leq Rd$, our assumption

$$\delta = \exp\left( -\frac{\varepsilon}{2} (\mathrm{h}(u) - \mathrm{h}^\circ) \right) \leq \frac{1}{4Rd},$$

implies that

$$\frac{2d_u \delta}{1 - 2d_u \delta} \leq 4Rd\delta.$$

We conclude that

$$|(\mathbb{E}_u ab)(\theta) - (\mathbb{E}_u ab)(\theta')| \leq 4Rd \exp\left( \frac{\varepsilon}{2} (\mathrm{h}(u) - \mathrm{h}^\circ) \right) \sqrt{\min_{\theta} (\mathbb{E}_u a^2)(\theta) \min_{\theta'} (\mathbb{E}_u b^2)(\theta')}.$$

$$\square$$

*Proof of Proposition H.2.* Let $C_0 = C_0(M, d)$ denote the constant introduced in the statement of the Proposition. Recall the decomposition of $f_u$ into $f_{u,I}$ from Definition E.4, consider the decomposition

$$f_u(x) = \sum_{I \subseteq [d_u] \,:\, |I| \geq 2} f_{u,I}(x) \text{ and } g_u(x) = \sum_{I \subseteq [d_u] \,:\, |I| \geq 2} g_{u,I}(x).$$

The proof of the Proposition will proceed by bounding summands in the formula below:

$$\left|(\mathbb{E}_u f_u g_u)(\theta) - (\mathbb{E}_u f_u g_u)(\theta')\right| \leq \sum_{I,J \subseteq [d_u] \,:\, |I|,|J| \geq 2} \left|(\mathbb{E}_u f_{u,I} g_{u,J})(\theta) - (\mathbb{E}_u f_{u,I} g_{u,J})(\theta')\right|. \quad (91)$$

Estimate of summands in (91): For any $I, J \subseteq [d_u]$ with $|I|, |J| \geq 2$, we have two cases to consider: First, we consider the case $I \neq J$. Notice that, by Lemma H.1, $\mathcal{A}$ satisfies Assumption B.3 with parameters $(\mathrm{h}^\circ + \frac{2}{\varepsilon} \log(2), \frac{1}{2})$. This allows us to invoke Corollary E.6, yielding

$$\left|(\mathbb{E}_u f_{u,I} g_{u,J})(\theta) - (\mathbb{E}_u f_{u,I} g_{u,J})(\theta')\right|$$
$$\leq 2 \max_{\theta \in [q]} \left|(\mathbb{E}_u f_{u,I} g_{u,J})(\theta)\right|$$
$$\leq 2 \exp\left(-\frac{\varepsilon |I \Delta J|}{2}(\mathrm{h}(u) - C_{E.6} - \mathrm{h}^\circ - \frac{2}{\varepsilon} \log(2))\right)\left(\max_{\theta \in [q]}(\mathbb{E}_u f_{u,I}^2)(\theta)\right)^{1/2} \cdot \left(\max_{\theta \in [q]}(\mathbb{E}_u g_{u,J}^2)(\theta)\right)^{1/2},$$

where $C_{E.6} = C_{E.6}(M, d, \frac{1}{2})$ is the constant introduced in the Corollary.

Let us impose the **first assumption** on $C_0$ that

$$C_0 \geq \frac{2}{\varepsilon}(1 + \log(4d)),$$

which implies that $\mathrm{h}(u) \geq \mathrm{h}^\circ + C_0(\log(R) + 1) \geq \mathrm{h}^\circ + \frac{2}{\varepsilon} \log(4dR)$. With this assumption, we could apply the remark (88) of Lemma H.4 to get

$$\left(\max_{\theta \in [q]}(\mathbb{E}_u f_{u,I}^2)(\theta)\right)^{1/2} \leq 2\left(\min_{\theta}(\mathbb{E}_u f_{u,I}^2)(\theta)\right)^{1/2}$$

and the same holds for $g_{u,J}$. Hence, for $I \neq J$ we have

$$\left|(\mathbb{E}_u f_{u,I} g_{u,J})(\theta) - (\mathbb{E}_u f_{u,I} g_{u,J})(\theta')\right|$$
$$\leq 4 \exp\left(-\frac{\varepsilon |I \Delta J|}{2}(\mathrm{h}(u) - C_{E.6} - \mathrm{h}^\circ + \frac{2}{\varepsilon} \log(2))\right)\left(\min_{\theta \in [q]}(\mathbb{E}_u f_{u,I}^2)(\theta)\right)^{1/2} \cdot \left(\min_{\theta \in [q]}(\mathbb{E}_u g_{u,J}^2)(\theta)\right)^{1/2}.$$

Second, we consider the case $I = J$. Here we simply apply Lemma H.4, yielding

$$\left|(\mathbb{E}_u f_{u,I} g_{u,I})(\theta) - (\mathbb{E}_u f_{u,I} g_{u,I})(\theta')\right|$$
$$\leq 4Rd \exp\left(-\frac{\varepsilon}{2}(\mathrm{h}(u) - \mathrm{h}^\circ)\right)\left(\min_{\theta \in [q]}(\mathbb{E}_u f_{u,I}^2)(\theta)\right)^{1/2} \cdot \left(\min_{\theta \in [q]}(\mathbb{E}_u g_{u,J}^2)(\theta)\right)^{1/2}.$$

Let us unify the above two estimates by introducing

$$C_1 = \max\left\{C_{E.6} + \frac{2}{\varepsilon} \log(2) + \frac{2}{\varepsilon} \log(8), \frac{2}{\varepsilon}(1 + \log(24d))\right\}.$$

Then,

$$\left|(\mathbb{E}_u f_{u,I} g_{u,J})(\theta) - (\mathbb{E}_u f_{u,I} g_{u,J})(\theta')\right| \quad (92)$$
$$\leq \underbrace{\frac{1}{6} \exp\left(-\frac{\varepsilon}{2} \max\{|I \Delta J|, 1\}(\mathrm{h}(u) - C_1(\log(R) + 1) - \mathrm{h}^\circ)\right)}_{:=a_{I,J}} \underbrace{\left(\min_{\theta \in [q]}(\mathbb{E}_u f_{u,I}^2)(\theta)\right)^{1/2}}_{:=\alpha_I} \cdot \underbrace{\left(\min_{\theta \in [q]}(\mathbb{E}_u g_{u,J}^2)(\theta)\right)^{1/2}}_{:=\beta_J},$$
$$\quad (93)$$

for every pair $I, J \subseteq [d_u]$ with $|I| \geq 2$ and $|J| \geq 2$.

Using this inequality, (91) becomes

$$\left|(\mathbb{E}_u f_u g_u)(\theta) - (\mathbb{E}_u f_u g_u)(\theta')\right| \leq \sum_{I,J\subseteq[d_u]\,:\,|I|,|J|\geq 2} a_{I,J}\alpha_I\beta_J = \vec{\alpha}^\top A\vec{\beta} \leq \|\vec{\alpha}\|\cdot\|A\|\cdot\|\vec{\beta}\| \quad (94)$$

where $\vec{\alpha} = (\alpha_I)_{I\subseteq[d_u]\,:\,|I|\geq 2}$, $\vec{\beta} = (\beta)_{I\subseteq[d_u]\,:\,|I|\geq 2}$, and $A = (a_{I,J})_{I,J\subseteq[d_u]\,:\,|I|,|J|\geq 2}$. Further, $\|\vec{\alpha}\|$ and $\|\vec{\beta}\|$ are the $\ell_2$ norms of $\vec{\alpha}$ and $\vec{\beta}$, respectively, and $\|A\|$ is the operator norm of $A$.

Estimate of operator norm of $A$: Notice that $A$ is a symmetric matrix. Thus, we can fix a unit vector $\vec{\gamma}$ satisfying $\|A\| = \vec{\gamma}^\top A\vec{\gamma}$. For each pair $I, J \subseteq [d_u]$ with $|I| \geq 2$ and $|J| \geq 2$, since $a_{I,J} \geq 0$,

$$\gamma_I a_{I,J}\gamma_J \leq \frac{a_{I,J}}{2}\gamma_I^2 + \frac{a_{I,J}}{2}\gamma_J^2,$$

and thus,

$$\|A\| = \sum_{I,J\subseteq[d_u]\,:\,|I|,|J|\geq 2} a_{I,J}\gamma_I\gamma_J \leq \sum_{I\subseteq[d_u]\,:\,|I|\geq 2}\gamma_I^2\Big(\sum_{J\subseteq[d_u]\,:\,|J|\geq 2} a_{I,J}\Big). \quad (95)$$

For each $I \subseteq [d_u]$ with $|I| \geq 2$, the number of $J \subseteq [d_u]$ with $|I\Delta J| = k$ is bounded above by $d_u^{k-1} \leq (Rd)^k$. Then, with the given estimate of $a_{I,J}$ in (92),

$$\sum_{J\subseteq[d_u]\,:\,|J|\geq 2} a_{I,J} \leq \frac{1}{6}\exp\Big(-\frac{\varepsilon}{2}(\mathrm{h}(u) - C_1(\log(R) + 1) - \mathrm{h}^\circ)\Big)$$

$$+ \sum_{t\geq 1}\frac{1}{6}(Rd)^t\exp\Big(-\frac{\varepsilon}{2}t(\mathrm{h}(u) - C_1(\log(R) + 1) - \mathrm{h}^\circ)\Big). \quad (96)$$

Now, we impose the **second assumption** on $C_0$ that

$$C_0 \geq C_1 + \frac{2}{\varepsilon}\Big(1 + \log(2d)\Big).$$

With the assumption that $\mathrm{h}(u) \geq \mathrm{h}^\circ + C_0(\log(R) + 1)$, the geometric sum in (96) has a decay rate smaller than $1/2$. Therefore,

$$\sum_{J\subseteq[d_u]\,:\,|J|\geq 2} a_{I,J} \leq \frac{1}{2}\exp\Big(-\frac{\varepsilon}{2}(\mathrm{h}(u) - C_2(\log(R) + 1) - \mathrm{h}^\circ)\Big),$$

where

$$C_2 = C_1 + \frac{2}{\varepsilon}\Big(1 + \log(d)\Big).$$

Now applying the above estimate, together with $\sum_{I\subseteq[d_u]\,:\,|I|\geq 2}\gamma_I^2 = 1$, to (95), we obtain the following bound:

$$\|A\| \leq \frac{1}{2}\exp\Big(-\frac{\varepsilon}{2}(\mathrm{h}(u) - C_1(\log(R) + 1) - \mathrm{h}^\circ)\Big).$$

Comparison of $\sum_I \min_{\theta\in[q]}(\mathbb{E}_u f_{u,I}^2)(\theta)$ and $\min_{\theta\in[q]}(\mathbb{E}_u f_u^2)(\theta)$ (and the same for $g$): Here is the last step toward the proof of the Proposition. Returning to (94), we have

$$\left|(\mathbb{E}_u f_u g_u)(\theta) - (\mathbb{E}_u f_u g_u)(\theta')\right|$$

$$\leq \frac{1}{2}\exp\Big(-\frac{\varepsilon}{2}(\mathrm{h}(u) - C_1(\log(R) + 1) - \mathrm{h}^\circ)\Big)\cdot\sqrt{\sum_I\min_{\theta\in[q]}(\mathbb{E}_u f_{u,I}^2)(\theta)}\cdot\sqrt{\sum_I\min_{\theta\in[q]}(\mathbb{E}_u g_{u,I}^2)(\theta)}.$$

Let us impose the **third assumption** on $C_0$ that

$$C_0 \geq C_{E.7} + \frac{2}{\varepsilon}\log(2)$$

where $C_{E.7}$ introduced in Corollary E.7. Recall that we have $\mathrm{h}^* = \mathrm{h}^\circ + \frac{2}{\varepsilon}\log(2)$ from (87). We can invoke this Corollary to yield:

$$\forall \theta \in [q], \; \sqrt{\sum_I (\mathbb{E}_u f_{u,I}^2)(\theta)} \leq \sqrt{2\mathbb{E}f_u^2(\theta)}.$$

Let $\theta_0 \in [q]$ be the value minimizing $\theta \mapsto \sqrt{\mathbb{E}f_u^2(\theta)}$. Then,

$$\min_{\theta \in [q]} \sqrt{2\mathbb{E}f_u^2(\theta)} = \sqrt{2\mathbb{E}f_u^2(\theta_0)} \geq \sqrt{\sum_I (\mathbb{E}_u f_{u,I}^2)(\theta_0)} \geq \sqrt{\sum_I \min_{\theta \in [q]}(\mathbb{E}_u f_{u,I}^2)(\theta)}.$$

Clearly, the same derivation also holds for $g_u$. Together we conclude that

$$\left|(\mathbb{E}_u f_u g_u)(\theta) - (\mathbb{E}_u f_u g_u)(\theta')\right|$$
$$\leq \exp\left(-\frac{\varepsilon}{2}(\mathrm{h}(u) - C_2(\log(R)+1) - \mathrm{h}^\circ)\right)\left(\min_{\theta \in [q]}(\mathbb{E}_u f_u^2)(\theta)\right)^{1/2}\left(\min_{\theta \in [q]}(\mathbb{E}_u g_u^2)(\theta)\right)^{1/2}.$$

Finally, if we impose the **forth assumption** on $C_0$ that

$$C_0 \geq C_2,$$

then the Proposition follows.

$\square$

## H.2 Properties of $f_k$: Products

The goal of this subsection is to establish the following.

**Proposition H.6.** *There exists $C = C(M,d) \geq 1$ so that the following holds. For any $\rho' \in T$ satisfying*

$$\mathrm{h}(\rho') \geq \mathrm{h}^\circ + C(\log(R)+1)$$

*and a positive integer $\mathrm{h}^\circ + C(\log(R)+1) \leq k_1 \leq \mathrm{h}(\rho')$. Consider a function $f$ and $g$ are $\mathcal{B}_{\leq \rho'}$ polynomials with $\mathbb{E}f(X) = \mathbb{E}g(X) = 0$. We decompose $f$ and $g$ according to Lemma D.1 with the given $k_1$. Then, the following holds: For $k_1 \leq m, k \leq \mathrm{h}(\rho')$,*

- *If $\max\{m,k\} > k_1$,*

$$\max_{\theta,\theta' \in [q]} \left|(\mathbb{E}_{\rho'} f_k g_m)(\theta) - (\mathbb{E}_{\rho'} f_k g_m)(\theta')\right|$$
$$\leq \exp\left(-\frac{\varepsilon}{2}(2\mathrm{h}(\rho') - \max\{k,m\} - C(\log(R)+1) - \mathrm{h}^\circ)\right)(\mathbb{E}f_k^2(X))^{1/2}(\mathbb{E}g_m^2(X))^{1/2}.$$

- *If $k = m = k_1$,*

$$\max_{\theta,\theta' \in [q]} \left|(\mathbb{E}_{\rho'} f_k g_m)(\theta) - (\mathbb{E}_{\rho'} f_k g_m)(\theta')\right|$$
$$\leq \exp\left(-\frac{\varepsilon}{2}(2\mathrm{h}(\rho') - 2k_1 - C(\log(R)+1))\right)(\mathbb{E}f_k^2(X))^{1/2}(\mathbb{E}g_m^2(X))^{1/2}.$$

The proof mirrors the structure used in Proposition F.1. In this case, we rely on both Proposition E.1 and Proposition H.2. Through this subsection, let

$$C^\circ = C^\circ(M,d)$$

be the constant described in the Proposition. The functions $f$, $g$, and $k_1$ are as introduced in the Proposition.

Assuming without lose of generality that $m \le k$, we apply the reasoning from (72) in Proposition F.1, yielding

$$
\begin{aligned}
&(\mathbb{E}_{\rho'} f_k g_m)(\theta) \\
=&\mathbb{E}\Big[\Big(\sum_{u \in D_k(\rho')} (\mathbb{E}_u \tilde{f}_u)(X)\Big)\Big(\sum_{u \in D_k(\rho')} (\mathbb{E}_u g_{m,u})(X)\Big) \Big| X_{\rho'} = \theta\Big] + \sum_{u \in D_k(\rho')} (\mathbb{E}_{\rho'} \tilde{f}_u g_{m,u})(\theta) \\
&- \sum_{u \in D_k(\rho')} \mathbb{E}\Big[(\mathbb{E}_u \tilde{f}_u)(X_u)(\mathbb{E}_u g_{m,u})(X_u) \Big| X_{\rho'} = \theta\Big],
\end{aligned}
$$

and hence,

$$
\begin{aligned}
&(\mathbb{E}_{\rho'} f_k g_m)(\theta) - (\mathbb{E}_{\rho'} f_k g_m)(\theta') \\
=&\big(\mathbb{E}_{\rho'}(\mathbb{E}_k f_k)(\mathbb{E}_k g_m)\big)(\theta) - \big(\mathbb{E}_{\rho'}(\mathbb{E}_k f_k)(\mathbb{E}_k g_m)\big)(\theta') \\
&+ \sum_{u \in D_k(\rho')} \big((\mathbb{E}_{\rho'} \tilde{f}_u g_{m,u})(\theta) - (\mathbb{E}_{\rho'} \tilde{f}_u g_{m,u})(\theta')\big) \\
&- \Big(\sum_{u \in D_k(\rho')} \big((\mathbb{E}_{\rho'}(\mathbb{E}_u \tilde{f}_u)(\mathbb{E}_u g_{m,u}))(\theta) - ((\mathbb{E}_{\rho'}(\mathbb{E}_u \tilde{f}_u)(\mathbb{E}_u g_{m,u}))(\theta')\big)\Big). \quad (97)
\end{aligned}
$$

Similar to the derivation of (64) from Proposition F.1. The proof is dedicated into estimating the above three summands.

We begin with the following estimate:

**Lemma H.7.** *There exists a constant $C = C(M, d)$ so that the following holds. Suppose $C^\circ \ge C_{H.2}$, where $C_{H.2}$ is the constant introduced in Proposition H.2. Then, the following holds: For $u \in D_k(\rho')$,*

1. *if $k > k_1$, then*

$$
\begin{aligned}
&\max_{\theta, \theta' \in [q]} \big|(\mathbb{E}_{\rho'} \tilde{f}_u g_{m,u})(\theta) - \mathbb{E}_{\rho'} \tilde{f}_u g_{m,u})(\theta')\big| \\
&\le \exp\Big(-\frac{\varepsilon}{2}(2\mathrm{h}(\rho') - k - C(\log(R) + 1) - \mathrm{h}^\circ)\Big)(\mathbb{E}\tilde{f}_u^2(X))^{1/2}(\mathbb{E}g_{m,u}^2(X))^{1/2};
\end{aligned}
$$

2. *if $k = m = k_1$, then*

$$
\begin{aligned}
&\max_{\theta, \theta' \in [q]} \big|(\mathbb{E}_{\rho'} \tilde{f}_u g_{m,u})(\theta) - \mathbb{E}_{\rho'} \tilde{f}_u g_{m,u})(\theta')\big| \\
&\le \exp\Big(-\frac{\varepsilon}{2}(2\mathrm{h}(\rho') - 2k - C)\Big)(\mathbb{E}\tilde{f}_u^2(X))^{1/2}(\mathbb{E}g_{m,u}^2(X))^{1/2}.
\end{aligned}
$$

*Proof.* Step 1. Bound $\mathbb{E}|\tilde{f}_u(X)g_{m,u}(X) - \mathbb{E}\tilde{f}_u g_{m,u}|$ from above: By Hölder's inequality,

$$
\underbrace{\mathrm{Var}\big[(\mathbb{E}_u \tilde{f}_u g_{m,u})(X_u)\big]}_{\ell_2\text{-norm}} \le \underbrace{\max_{\theta \in [q]} \big|(\mathbb{E}_u \tilde{f}_u g_{m,u})(\theta) - \mathbb{E}\tilde{f}_u g_{m,u}\big|}_{\ell_\infty\text{-norm}} \cdot \underbrace{\mathbb{E}\big|(\mathbb{E}_u \tilde{f}_u g_{m,u})(X_u) - \mathbb{E}\tilde{f}_u g_{m,u}\big|}_{\ell_1\text{-norm}}
$$

$$
\le \sqrt{C_{A.5}\mathrm{Var}\big[(\mathbb{E}_u \tilde{f}_u g_{m,u})(X_u)\big]} \cdot \mathbb{E}|\tilde{f}_u(X)g_{m,u}(X) - \mathbb{E}\tilde{f}_u g_{m,u}|
$$

$$
\Leftrightarrow \quad \sqrt{\mathrm{Var}\big[(\mathbb{E}_u \tilde{f}_u g_{m,u})(X_u)\big]} \le \sqrt{C_{A.5}}\mathbb{E}|\tilde{f}_u(X)g_{m,u}(X) - \mathbb{E}\tilde{f}_u g_{m,u}|, \quad (98)
$$

where we applied (9) from Lemma A.5 with $C_{A.5}$ is the constant introduced in the Lemma. Further, relying on (98), together with (9) and (8) from the Lemma A.5, we have

$$
\begin{aligned}
&\max_{\theta, \theta' \in [q]} \big|(\mathbb{E}_{\rho'} \tilde{f}_u g_{m,u})(\theta) - \mathbb{E}_{\rho'} \tilde{f}_u g_{m,u})(\theta')\big| \\
\le& 2\max_{\theta \in [q]} \big|(\mathbb{E}_{\rho'} \tilde{f}_u g_{m,u})(\theta) - \mathbb{E}\tilde{f}_u g_{m,u}\big| \\
\le& 2C_{A.5}(\mathrm{h}(\rho') - k)^q \lambda^{\mathrm{h}(\rho')-k} \max_{\theta \in [q]} \big|(\mathbb{E}_u \tilde{f}_u g_{m,u})(\theta) - \mathbb{E}\tilde{f}_u g_{m,u}\big| \\
\le& 2C_{A.5}^2(\mathrm{h}(\rho') - k)^q \lambda^{\mathrm{h}(\rho')-k} \sqrt{C_{A.5}}\mathbb{E}|\tilde{f}_u(X)g_{m,u}(X) - \mathbb{E}\tilde{f}_u g_{m,u}| \\
=& C_1 \exp(-\varepsilon(\mathrm{h}(\rho') - k))\mathbb{E}|\tilde{f}_u(X)g_{m,u}(X) - \mathbb{E}\tilde{f}_u g_{m,u}|, \quad (99)
\end{aligned}
$$

where

$$C_1 = 2C_{A.5}^{5/2} \cdot \max_{n \in \mathbb{N}} n^q \exp(-0.1\varepsilon n).$$

Case 1: $m < k$. Here we can simply recycle the estimate from (76):

$$\mathbb{E}|\tilde{f}_u(X)g_{m,u}(X) - \mathbb{E}\tilde{f}_u g_{m,u}| \leq 2\mathbb{E}|\tilde{f}_u(X)g_{m,u}(X)|$$
$$\leq \exp\Big(-\frac{\varepsilon}{2}\big(k - C_2(\log(R)+1) - \mathrm{h}^\circ\big)\Big)(\mathbb{E}\tilde{f}_u^2(X))^{1/2}(\mathbb{E}g_{m,u}^2(X))^{1/2}, \tag{100}$$

where

$$C_2 = \frac{2}{\varepsilon}\Big(\frac{3}{2} + \frac{1}{2}\log(d) + \log(C_{D.1})\Big) + C_{E.1} + 2 \cdot \frac{2}{\varepsilon}\log(2),$$

where $C_{D.1}$ is the constant introduced in Lemma D.1 and $C_{E.1}$ is the constant introduced in Proposition E.1.

Case 2: $k_1 < m = k$.

This is the case where we need Proposition H.2. With the assumption that $C^\circ \geq C_{H.2}$, where $C_{H.2} \geq 1$ is the constant introduced in the Proposition, we have

$$m = k > k_1 \geq \mathrm{h}^\circ + C^\circ(\log(R)+1) \geq \mathrm{h}^\circ + C_{H.2}(\log(R)+1),$$

so that we could apply the Proposition to get

$$\mathbb{E}|\tilde{f}_u(X)g_{m,u}(X) - \mathbb{E}\tilde{f}_u g_{m,u}|$$
$$\leq \max_{\theta,\theta'}\big|(\mathbb{E}_u f_u g_u)(\theta) - (\mathbb{E}_u f_u g_u)(\theta')\big|$$
$$\leq \exp\Big(-\frac{\varepsilon}{2}(k - C_{H.2}(\log(R)+1) - \mathrm{h}^\circ)\Big)(\mathbb{E}\tilde{f}_u^2(X))^{1/2}(\mathbb{E}g_{m,u}^2(X))^{1/2}.$$

Case 3: $k_1 = m = k$ The last case is straightforward:

$$\mathbb{E}|\tilde{f}_u(X)g_{m,u}(X) - \mathbb{E}\tilde{f}_u g_{m,u}| \leq 2\mathbb{E}|\tilde{f}_u(X)g_{m,u}(X)| \leq 2\sqrt{\mathbb{E}\tilde{f}_u^2(X)}\sqrt{\mathbb{E}g_{m,u}^2(X)}.$$

By taking $C_3 = \max\{C_2, C_{H.2}\} + \frac{2}{\varepsilon}\log(C_1)$, the statement of the Lemma follows with $C = C_3$. $\square$

As an analogue of the above Lemma, we also have

**Lemma H.8.** *There exists a constant $C = C(M, d)$ so that the following holds. Suppose $C^\circ \geq C_{H.2}$ is the constant introduced in Proposition H.2. Then, the following holds: For $u \in D_k(\rho')$,*

1. *if $k > k_1$, then*

$$\max_{\theta,\theta' \in [q]}\big|(\mathbb{E}_{\rho'}(\mathbb{E}_u \tilde{f}_u)(\mathbb{E}_u g_{m,u}))(\theta) - ((\mathbb{E}_{\rho'}(\mathbb{E}_u \tilde{f}_u)(\mathbb{E}_u g_{m,u}))(\theta')\big|$$
$$\leq \exp\Big(-\frac{\varepsilon}{2}(2\mathrm{h}(\rho') - k - C(\log(R)+1) - \mathrm{h}^\circ)\Big)(\mathbb{E}\tilde{f}_u^2(X))^{1/2}(\mathbb{E}g_{m,u}^2(X))^{1/2}.$$

2. *if $k = m = k_1$, then*

$$\big|(\mathbb{E}_{\rho'}(\mathbb{E}_u \tilde{f}_u)(\mathbb{E}_u g_{m,u}))(\theta) - ((\mathbb{E}_{\rho'}(\mathbb{E}_u \tilde{f}_u)(\mathbb{E}_u g_{m,u}))(\theta')\big|$$
$$\leq \exp\Big(-\frac{\varepsilon}{2}(2\mathrm{h}(\rho') - 2k - C)\Big)(\mathbb{E}\tilde{f}_u^2(X))^{1/2}(\mathbb{E}g_{m,u}^2(X))^{1/2}.$$

Since the proof is simpler and the structure is the same as that for Lemma H.7, we will outline a sketch proof in this case.

*Proof.* Let $a_u(x_u) = (\mathbb{E}_u \tilde{f}_u)(x_u)$ and $b_u = (\mathbb{E}_u g_{m,u})(x_u)$. Repeating the first step of the proof of Lemma H.7, we have

$$\max_{\theta,\theta'\in[q]} \left|(\mathbb{E}_{\rho'} a_u b_u)(\theta) - \mathbb{E}_{\rho'} a_u b_u)(\theta')\right| \leq C_1 \exp(-\varepsilon(\mathrm{h}(\rho') - k))\mathbb{E}|a_u(X)b_u(X) - \mathbb{E}a_u b_u|,$$

with

$$C_1 := 2C_{A.5}^{5/2} \cdot \max_{n\in\mathbb{N}} n^q \exp(-0.1\varepsilon n),$$

which is exactly the same constant stated in Lemma H.7. Next,

$$\mathbb{E}|a_u(X)b_u(X) - \mathbb{E}a_u b_u| \leq 2\mathbb{E}|a_u(X)b_u(X)| \leq 2\sqrt{\mathbb{E}a_u^2(X)}\sqrt{\mathbb{E}b_u^2(X)} \leq 2\sqrt{\mathbb{E}a_u^2(X)}\sqrt{\mathbb{E}g_{m,u}^2(X)}.$$

If $k > k_1$, we could apply (38) from Proposition E.1 to $\tilde{f}_u$, and get

$$\sqrt{\mathbb{E}a_u^2(X)} \leq \exp\left( - 2\varepsilon(k - C_{E.1}(\log(R) + 1) \underbrace{-\mathrm{h}^\circ - \frac{2}{\varepsilon}\log(2))}_{-\mathrm{h}^*}\right) \sqrt{\mathbb{E}\tilde{f}_u^2(X)}.$$

Indeed, this tail bound is stronger than what we got from Lemma H.7. The remainning part involves combining these estimates with a suitable constant $C$ so that the lemma holds. Given the argument was already presented in the proof of Lemma H.7, we will omit these details. $\square$

Before bounding the summands in (97), let us bound $\sum_{u\in D_k(\rho')} \mathbb{E}\tilde{f}_u^2(X)$ and $\sum_{u\in D_k(\rho')} \mathbb{E}g_{m,u}^2(X)$ from above by $\mathbb{E}f_k^2(X)$ and $\mathbb{E}g_m^2(X)$, respectively.

**Lemma H.9.** *Suppose*

$$C^\circ \geq C_{F.2} + \frac{2}{\varepsilon}\log(2),$$

*where $C_{F.2}$ is the constant introduced in Lemma F.2. Then,*

$$\sum_{u\in D_k(\rho')} \mathbb{E}g_{m,u}^2(X) \leq \max\left\{4, C_{D.1}^2 R^4\right\} \cdot \mathbb{E}g_m^2(X),$$

*and*

$$\sum_{u\in D_k(\rho')} \mathbb{E}\tilde{f}_u^2(X) \leq \max\left\{4, C_{D.1}^2 R^4\right\} \cdot \mathbb{E}f_k^2(X).$$

*Proof.* Given that $C^\circ \geq C_{F.2} + \frac{2}{\varepsilon}\log(2)$, we have

$$k_1 \geq \mathrm{h}^\circ + C^\circ(\log(R) + 1) > \mathrm{h}^\circ + \frac{2}{\varepsilon}\log(2) + C_{F.2}(\log(R) + 1) = \mathrm{h}^* + C_{F.2}(\log(R) + 1),$$

and thus we could apply Lemma F.2. When $m > k_1$, the lemma yields

$$\sum_{u\in D_k(\rho')} \mathbb{E}g_{m,u}^2(X) = \sum_{u\in D_k(\rho')} \mathbb{E}\left[\left(\sum_{v\in D_m(u)} \tilde{g}_v(X)\right)^2\right] \leq \sum_{u\in D_k(\rho')}\sum_{v\in D_m(u)} 2\mathbb{E}\tilde{g}_v^2(X) \leq 4\mathbb{E}g_m^2(X).$$

And in the case when $m = k_1$, we use the same derivation with Lemma F.2 been replaced by (31) in Lemma D.1 to get

$$\sum_{u\in D_k(\rho')} \mathbb{E}g_{m,u}^2(X) \leq C_{D.1}R^3 \cdot C_{D.1}R\mathbb{E}g_m^2(X).$$

Clearly, the same derivation also holds for the comparison of $\sum_{u\in D_k(\rho')} \mathbb{E}\tilde{f}_u^2(X)$ and $\mathbb{E}f_k^2(X)$. $\square$

Now, relying on the above two lemmas, we will estimate the second and third summand of (97):

**Corollary H.10.** *There exists a constant $C = C(M, d) \geq 1$ so that the following holds. Suppose*

$$C^\circ \geq \max\left\{C_{H.2}, C_{F.2} + \frac{2}{\varepsilon}\log(2)\right\},$$

*where the constants are introduced in Proposition H.2 and Lemma F.2, respectively. Then,*

1. if $k > k_1$, then

$$\left| \sum_{u \in D_k(\rho')} \left( (\mathbb{E}_{\rho'} \tilde{f}_u g_{m,u})(\theta) - (\mathbb{E}_{\rho'} \tilde{f}_u g_{m,u})(\theta') \right) \right.$$

$$\left. - \left( \sum_{u \in D_k(\rho')} \left( (\mathbb{E}_{\rho'} (\mathbb{E}_u \tilde{f}_u)(\mathbb{E}_u g_{m,u}))(\theta) - ((\mathbb{E}_{\rho'} (\mathbb{E}_u \tilde{f}_u)(\mathbb{E}_u g_{m,u}))(\theta') \right) \right) \right|$$

$$\leq \exp\left( -\frac{\varepsilon}{2}(2\mathrm{h}(\rho') - k - C(\log(R) + 1) - \mathrm{h}^\circ) \right) (\mathbb{E} f_k^2(X))^{1/2} (\mathbb{E} g_m^2(X))^{1/2}.$$

2. if $k = m = k_1$, then the above term above can be bounded by

$$\exp\left( -\frac{\varepsilon}{2}(2\mathrm{h}(\rho') - 2k - C) \right) (\mathbb{E} f_k^2(X))^{1/2} (\mathbb{E} g_m^2(X))^{1/2}.$$

*Proof.* Let $C_1$ be the maximum of the two constants introduced in Lemma H.7 and Lemma H.8. For convenience, let

$$U := \left| \sum_{u \in D_k(\rho')} \left( (\mathbb{E}_{\rho'} \tilde{f}_u g_{m,u})(\theta) - (\mathbb{E}_{\rho'} \tilde{f}_u g_{m,u})(\theta') \right) \right.$$

$$\left. - \left( \sum_{u \in D_k(\rho')} \left( (\mathbb{E}_{\rho'} (\mathbb{E}_u \tilde{f}_u)(\mathbb{E}_u g_{m,u}))(\theta) - ((\mathbb{E}_{\rho'} (\mathbb{E}_u \tilde{f}_u)(\mathbb{E}_u g_{m,u}))(\theta') \right) \right) \right|.$$

By the two lemmas together with the triangle inequality, in the case when $k > k_1$, we have

$$U \leq \sum_{u \in D_k(\rho')} 2 \exp\left( -\frac{\varepsilon}{2}(2\mathrm{h}(\rho') - k - C_1(\log(R) + 1) - \mathrm{h}^\circ) \right) (\mathbb{E} \tilde{f}_u^2(X))^{1/2} (\mathbb{E} g_{m,u}^2(X))^{1/2}$$

$$\leq 2 \exp\left( -\frac{\varepsilon}{2}(2\mathrm{h}(\rho') - k - C_1(\log(R) + 1) - \mathrm{h}^\circ) \right) \sqrt{\sum_{u \in D_k(\rho')} \mathbb{E} \tilde{f}_u^2(X)} \sqrt{\sum_{u \in D_k(\rho')} \mathbb{E} g_{m,u}^2(X)}$$

$$\leq 2 \max\left\{ 4, C_{D.1}^2 R^4 \right\} \sqrt{\mathbb{E} f_k^2(X)} \sqrt{\mathbb{E} g_m^2(X)}, \tag{101}$$

where the last inequality follows from Lemma H.9. Similarly, when $k = m = k_1$, we have

$$U \leq 2 \exp\left( -\frac{\varepsilon}{2}(2\mathrm{h}(\rho') - 2k - C_1) \right) \sqrt{\sum_{u \in D_k(\rho')} \mathbb{E} \tilde{f}_u^2(X)} \sqrt{\sum_{u \in D_k(\rho')} \mathbb{E} g_{m,u}^2(X)}. \tag{102}$$

By setting

$$C = \frac{2}{\varepsilon} \left( C_1 + \log(4) + \log(C_{D.1}^2) \right),$$

the corollary follows. $\qquad \square$

It remains to estimate the first summand of (97):

**Lemma H.11.** *There exists a constant $C = C(M, d) \geq 1$ so that the following holds. Suppose*

$$C^\circ \geq C_{F.2} + \frac{2}{\varepsilon} \log(2)$$

*where $C_{F.2}$ is the constant introduced in Lemma F.2. Then,*

1. if $k > k_1$, then

$$\max_{\theta, \theta' \in [q]} \left| (\mathbb{E}_{\rho'} (\mathbb{E}_k f_k)(\mathbb{E}_k g_m))(\theta) - (\mathbb{E}_{\rho'} (\mathbb{E}_k f_k)(\mathbb{E}_k g_m))(\theta') \right|$$

$$\leq \exp\left( -\varepsilon(\mathrm{h}(\rho') - C(\log(R) + 1) - \mathrm{h}^\circ) \right) \sqrt{\mathbb{E} f_k^2(X)} \sqrt{\mathbb{E} g_m^2(X)}.$$

2. if $k = m = k_1$, then the above term is bounded by

$$\exp(-\varepsilon(\mathrm{h}(\rho') - k_1 - C(\log(R) + 1))) \sqrt{\mathbb{E} f_k^2(X)} \sqrt{\mathbb{E} g_m^2(X)}.$$

*Proof.* Observe that both $(\mathbb{E}_k f_k)(x) = \sum_{u \in D_k(\rho')}(\mathbb{E}_u \tilde{f}_u)(x_u)$ and $(\mathbb{E}_k g_m)(x) = \sum_{u \in D_k(\rho')}(\mathbb{E}_k g_{m,u})(x_u)$ are both degree-1 polynomials with variables $(x_u : u \in D_k(\rho'))$ satisfying

$$\mathbb{E}(\mathbb{E}_k g_m)(X) = \mathbb{E}(\mathbb{E}_k f_k) = 0.$$

This allows us to apply Lemma G.4, yielding

$$\max_{\theta, \theta' \in [q]} \left| \big(\mathbb{E}_{\rho'}(\mathbb{E}_k f_k)(\mathbb{E}_k g_m)\big)(\theta) - \big(\mathbb{E}_{\rho'}(\mathbb{E}_k f_k)(\mathbb{E}_k g_m)\big)(\theta') \right|$$

$$\leq 2 \max_{\theta} \left| \big(\mathbb{E}_{\rho'}(\mathbb{E}_k f_k)(\mathbb{E}_k g_m)\big)(\theta) - \mathbb{E}\big(\mathbb{E}_k f_k)(\mathbb{E}_k g_m)\big) \right|$$

$$\leq 2 C_{G.4} R \exp(-\varepsilon(\mathrm{h}(\rho') - k)) \sqrt{\sum_{u \in D_k(\rho')} \mathbb{E}\big[(\mathbb{E}_u \tilde{f}_u)^2(X)\big]} \sqrt{\sum_{u \in D_k(\rho')} \mathbb{E}\big[(\mathbb{E}_u g_{m,u})^2(X)\big]},$$

where $C_{G.4} \geq 1$ is the constant introduced in Lemma G.4. Next, we apply Lemma H.9 (which is why we need the assumption on $C^\circ$) to get

$$\sqrt{\sum_{u \in D_k(\rho')} \mathbb{E}\big[(\mathbb{E}_u g_{m,u})^2(X)\big]} \leq \sqrt{\sum_{u \in D_k(\rho')} \mathbb{E} g_{m,u}^2(X)} \leq \sqrt{\max\big\{4, C_{D.1}^2 R^4\big\} \cdot \mathbb{E} g_m^2(X)}.$$

As for $\sqrt{\sum_{u \in D_k(\rho')} \mathbb{E}\big[(\mathbb{E}_u \tilde{f}_u)^2(X)\big]}$, if $k = m = k_1$, then we can apply the same derivation to get

$$\sqrt{\sum_{u \in D_k(\rho')} \mathbb{E}\big[(\mathbb{E}_u \tilde{f}_u)^2(X)\big]} \leq \sqrt{\max\big\{4, C_{D.1}^2 R^4\big\} \cdot \mathbb{E} f_k^2(X)}.$$

This leads to

$$\max_{\theta, \theta' \in [q]} \left| \big(\mathbb{E}_{\rho'}(\mathbb{E}_k f_k)(\mathbb{E}_k g_m)\big)(\theta) - \big(\mathbb{E}_{\rho'}(\mathbb{E}_k f_k)(\mathbb{E}_k g_m)\big)(\theta') \right|$$

$$\leq 2 C_{G.4} \max\big\{4, C_{D.1}^2 R^4\big\} \exp(-\varepsilon(\mathrm{h}(\rho') - k)) \sqrt{\mathbb{E} f_k^2(X)} \sqrt{\mathbb{E} g_m^2(X)}.$$

If $k > k_1$, then we can apply Proposition E.1 and Lemma H.9 to get

$$\sqrt{\sum_{u \in D_k(\rho')} \mathbb{E}\big[(\mathbb{E}_u \tilde{f}_u)^2(X)\big]}$$

$$\leq \exp\Big(-\varepsilon(k - C_{E.1}(\log(R) + 1) - \mathrm{h}^\circ - \frac{2}{\varepsilon}\log(2))\Big) \sqrt{\sum_{u \in D_k(\rho')} \mathbb{E} \tilde{f}_u^2(X)}$$

$$\leq \sqrt{\max\big\{4, C_{D.1}^2 R^4\big\}} \exp\Big(-\varepsilon(k - C_{E.1}(\log(R) + 1) - \mathrm{h}^\circ - \frac{2}{\varepsilon}\log(2))\Big) \sqrt{\mathbb{E} f_k^2(X)}.$$

In this case, we have

$$\max_{\theta, \theta' \in [q]} \left| \big(\mathbb{E}_{\rho'}(\mathbb{E}_k f_k)(\mathbb{E}_k g_m)\big)(\theta) - \big(\mathbb{E}_{\rho'}(\mathbb{E}_k f_k)(\mathbb{E}_k g_m)\big)(\theta') \right|$$

$$\leq 2 C_{G.4} \max\big\{4, C_{D.1}^2 R^4\big\} \exp\Big(-\varepsilon(\mathrm{h}(\rho') - C_{E.1}(\log(R) + 1) - \mathrm{h}^\circ - \frac{2}{\varepsilon}\log(2))\Big) \sqrt{\mathbb{E} f_k^2(X)} \sqrt{\mathbb{E} g_m^2(X)}.$$

By taking

$$C = C_{E.1} + \frac{2}{\varepsilon}\log(2) + \frac{1}{\varepsilon}\Big(\log(2 C_{G.4}) + \log(C_{D.1}^2) + 4\Big),$$

both statements of the lemma follows. $\qquad\square$

*Proof of Proposition H.6.* Without lose of generality, it is sufficient to prove the case when $m \leq k$.

First, we impose the **first assumption** that

$$C^\circ \geq \max\Big\{C_{H.2}, C_{F.2} + \frac{2}{\varepsilon}\log(2)\Big\},$$

where the constants are introduced in Proposition H.2 and Lemma F.2, respectively. This allows us to apply Corollary H.10 and Lemma H.11. For simplicity, let

$$C_1 := \max\{C_{H.10}, C_{H.11}\}.$$

Then, combining the Corollary and the Lemma to the estimate (97) we can conclude that: For $k_1 \leq m \leq k$ with $k > k_1$,

$$\max_{\theta, \theta' \in [q]} \left| (\mathbb{E}_{\rho'} f_k g_m)(\theta) - (\mathbb{E}_{\rho'} f_k g_m)(\theta') \right|$$
$$\leq 2 \exp \left( - \frac{\varepsilon}{2} (2\mathrm{h}(\rho') - k - C_1 (\log(R) + 1) - \mathrm{h}^\circ) \right) (\mathbb{E} f_k^2(X))^{1/2} (\mathbb{E} g_m^2(X))^{1/2},$$

and in the case where $k = m = k_1$, the above term is bounded by

$$2 \exp \left( - \frac{\varepsilon}{2} (2\mathrm{h}(\rho') - 2k - C_1 (\log(R) + 1)) \right) (\mathbb{E} f_k^2(X))^{1/2} (\mathbb{E} g_m^2(X))^{1/2}.$$

Then, the proof of the proposition follows by making the **second assumption** on $C^\circ$ that

$$C^\circ \geq C_1 + \frac{2}{\varepsilon} \log(2).$$

$\square$

## H.3 Proof of Theorem G.3

*Proof.* Now we are ready to establish the main theorem. As usual, let $C_0 = C_0(M, d)$ denote the constant introduced in the statement of the Theorem. The value of $C_0$ will be determined as the proof proceeds.

Applying Theorem B.6 with $\mathcal{A}$ and $\mathrm{h}^* = \mathrm{h}^\circ + \frac{2}{\varepsilon} \log(2)$ and $c^* = 1/2$, we conclude that

$$\mathrm{Var}\left[ (\mathbb{E}_{\rho'} f)(X) \right] \leq \exp \left( - \varepsilon (\mathrm{h}(\rho') - C_{B.6} (\log(R) + 1) - \mathrm{h}^*) \mathrm{Var} \left[ f(X) \right].$$

for any $\mathcal{B}_{\leq \rho'}$-polynomial $f$, where $C_{B.6} = C(M, d, 1/2)$ is the constant introduced by the theorem. We impose the **first assumption** on $C_0$ that

$$C_0 \geq C_{B.6} + \frac{2}{\varepsilon} \log(2),$$

and conclude that

$$\mathrm{Var}\left[ (\mathbb{E}_{\rho'} f)(X) \right] \leq \exp \left( - \varepsilon (\mathrm{h}(\rho') - C_0 (\log(R) + 1) - \mathrm{h}^\circ) \mathrm{Var} \left[ f(X) \right].$$

Now, it remains to show that with the suitable choice of $C_0$, for any $\rho'$ with $\mathrm{h}(\rho') \geq \mathrm{h}^\circ + C_0 (\log(R) + 1)$ and any two $\mathcal{B}_{\leq \rho'}$-polynomials $f$ and $g$, we have

$$\max_{\theta \in [q]} |(\mathbb{E}_{\rho'} fg)(\theta) - \mathbb{E} fg| \leq \exp \left( - \frac{\varepsilon}{2} (\mathrm{h}(\rho') - \mathrm{h}^\circ - C_0 (\log(R) + 1)) \right) \sqrt{\min_{\theta} (\mathbb{E}_{\rho'} f^2)(\theta) \min_{\theta'} (\mathbb{E}_{\rho'} g^2)(\theta)}.$$

Let

$$C_1 := \max \left\{ C_{H.6}, \frac{2}{\varepsilon} \log(2) + C_{F.1} + t_0 \right\},$$

where

- $t_0$ is the constant such that $\sum_{t=t_0}^{\infty} \exp \left( - \frac{\varepsilon}{2} t \right) \leq \frac{1}{2}$,

- $C_{H.6}$ is the constant introduced in Proposition H.6, and

- $C_{F.1} = C(M, d, 1/2)$ is the constant introduced in Proposition F.1.

Next, let
$$k_1 = \lceil \mathrm{h}^\circ + C_1(\log(R) + 1) \rceil.$$
The chocie of $C_1$ and $k_1$ allow us to apply Proposition H.6 and Proposition F.1 toward both $f$ and $g$.

Next, we impose the **second assumption** on $C_0$ that
$$C_0 \geq 2C_1 + 2.$$
This assumption implies that there is a **gap** between $h(\rho')$ and $k_1$, which is necessary for the proof.

Now, we fix such $\rho'$ and consider two $\mathcal{B}_{\leq\rho'}$-polynomial $f$ and $g$ with $\mathbb{E}f(X) = \mathbb{E}g(X) = 0$. Further, consider the decomposition of $f$ and $g$ according to Lemma D.1 with the above chosen $k_1$.

First, by our choice of $C_1$, we have
$$k_1 \geq \lceil \underbrace{\mathrm{h}^\circ + \frac{2}{\varepsilon}\log(2)}_{=\mathrm{h}^*} + C_{F.1}(\log(R) + 1) + t_0 \rceil.$$
This assumption allow us to recycle the partial step in the proof of Theorem B.6 to obtain (80):
$$\sum_{k\in[k_1,\mathrm{h}(\rho')]} \mathbb{E}f_k^2(X) \leq 2\mathbb{E}f^2(X) \quad \text{and} \quad \sum_{k\in[k_1,\mathrm{h}(\rho')]} \mathbb{E}g_k^2(X) \leq 2\mathbb{E}g^2(X). \tag{103}$$

Second, with our assumption that $k_1 \geq \lceil \mathrm{h}^\circ + C_{H.6}(\log(R) + 1) \rceil$, we can apply Proposition H.6 to get
$$\max_{\theta,\theta'\in[q]} \left| (\mathbb{E}_{\rho'}fg)(\theta) - (\mathbb{E}_{\rho'}fg)(\theta') \right| \leq \sum_{m,k\in[k_1,\mathrm{h}(\rho')]} \max_{\theta,\theta'\in[q]} \left| (\mathbb{E}_{\rho'}f_kg_m)(\theta) - (\mathbb{E}_{\rho'}f_kg_m)(\theta') \right|$$
$$\leq \sum_{m,k\in[k_1,\mathrm{h}(\rho')]} a_{km}\alpha_k\beta_m = \vec{\alpha}^\top A\vec{\beta} \leq \|\vec{\alpha}\|\|A\|\|\vec{\beta}\|,$$
where $\vec{\alpha} = (\alpha_{k_1}, \alpha_{k_2}, \ldots, \alpha_{\mathrm{h}(\rho')})$ with $\alpha_k = \sqrt{\mathbb{E}f_k^2(X)}$, $\vec{\beta} = (\beta_{k_1}, \beta_{k_2}, \ldots, \beta_{\mathrm{h}(\rho')})$ with $\beta_m = \sqrt{\mathbb{E}g_m^2(X)}$, and $A = (a_{km})_{k,m\in[k_1,\mathrm{h}(\rho')]}$ with
$$a_{km} := \begin{cases} \exp\left( -\frac{\varepsilon}{2}(\mathrm{h}(\rho') + \mathrm{h}(\rho') - \max\{k,m\} - C_{H.6}(\log(R) + 1) - \mathrm{h}^\circ) \right) & \max\{k,m\} > k_1 \\ \exp\left( -\frac{\varepsilon}{2}(\mathrm{h}(\rho') + \mathrm{h}(\rho') - 2k_1 - C_{H.6}(\log(R) + 1)) \right) & k = m = k_1. \end{cases}$$

Together with (103), we have
$$\|\vec{\alpha}\|\|A\|\|\vec{\beta}\| \leq 2\|A\|\sqrt{\mathbb{E}f^2(X)}\sqrt{\mathbb{E}g^2(X)}.$$

The next goal is to bound $\|A\|$ from above. Notice the fact that the matrix $A$ is symmetric implies there exists a unit vector $\vec{\gamma}$ such that $\|A\| = \vec{\gamma}^\top A\vec{\gamma}$. Now we fix such vector $\vec{\gamma}$. Relying on the fact that $a_{km} \geq 0$,
$$\|A\| = \sum_{k,m\in[k_1,\mathrm{h}(\rho')]} a_{km}\gamma_k\gamma_m \leq \sum_{k,m\in[k_1,\mathrm{h}(\rho')]} \frac{a_{km}}{2}(\gamma_k^2 + \gamma_m^2) = \sum_{m\in[k_1,\mathrm{h}(\rho')]} \gamma_m^2 \left( \sum_{k\in[k_1,\mathrm{h}(\rho')]} a_{km} \right).$$

Clearly, from the definition of $a_{km}$, the term $\sum_{k\in[k_1,\mathrm{h}(\rho')]} a_{km}$ is maximized when $m = k_1$.
$$\sum_{k\in[k_1,\mathrm{h}(\rho')]} a_{kk_1} = \exp\left( -\frac{\varepsilon}{2}(\mathrm{h}(\rho') + \mathrm{h}(\rho') - 2k_1 - C_{H.6}(\log(R) + 1)) \right)$$
$$+ \sum_{k\in[k_1,\mathrm{h}(\rho')]} \exp\left( -\frac{\varepsilon}{2}(\mathrm{h}(\rho') + \mathrm{h}(\rho') - k - C_{H.6}(\log(R) + 1) - \mathrm{h}^\circ) \right)$$
$$= \exp\left( -\frac{\varepsilon}{2}(\mathrm{h}(\rho') - C_{H.6}(\log(R) + 1) - \mathrm{h}^\circ) \right) \cdot$$
$$\cdot \left( \exp\left( -\frac{\varepsilon}{2}(\mathrm{h}(\rho') - 2k_1 + \mathrm{h}^\circ) \right) + \sum_{k\in[k_1,\mathrm{h}(\rho')]} \exp\left( -\frac{\varepsilon}{2}(\mathrm{h}(\rho') - k) \right) \right).$$

First,

$$\sum_{k \in [k_1, h(\rho')]} \exp\left(-\frac{\varepsilon}{2}(h(\rho') - k)\right) \leq \frac{1}{1 - \exp(-\varepsilon/2)} \leq \frac{4}{\varepsilon}.$$

Second,

$$\exp\left(-\frac{\varepsilon}{2}(h(\rho') - 2k_1 + h^\circ)\right) \leq \exp\left(-\frac{\varepsilon}{2}\Big(C_0(\log(R) + 1) + h^\circ - 2(h^\circ + C_1(\log(R) + 1) + 1) + h^\circ\Big)\right)$$

$$\leq \exp\left(-\frac{\varepsilon}{2}(C_0 - 2C_1 - 2)(\log(R) + 1)\right) \leq 1,$$

which in turn implies that

$$\left(\exp\left(-\frac{\varepsilon}{2}(h(\rho') - 2k_1 + h^\circ)\right) + \sum_{k \in [k_1, h(\rho')]} \exp\left(-\frac{\varepsilon}{2}(h(\rho') - k)\right)\right) \leq \frac{5}{\varepsilon}.$$

Hence, we conclude that

$$\|A\| \leq \frac{5}{\varepsilon} \exp\left(-\frac{\varepsilon}{2}(h(\rho') - C_{H.6}(\log(R) + 1) - h^\circ)\right).$$

Together we conclude that when $h(\rho) \geq h^\circ + C_1(\log(R) + 1)$, any two $\mathcal{B}_{\leq \rho'}$-polynomials $f$ and $g$ with $\mathbb{E}f(X) = \mathbb{E}g(X) = 0$ satisfies

$$\max_{\theta, \theta' \in [q]} \left|(\mathbb{E}_{\rho'}fg)(\theta) - (\mathbb{E}_{\rho'}fg)(\theta')\right| \tag{104}$$

$$\leq \frac{10}{\varepsilon} \exp\left(-\frac{\varepsilon}{2}(h(\rho') - C_{H.6}(\log(R) + 1) - h^\circ)\right)\sqrt{\mathbb{E}f^2(X)}\sqrt{\mathbb{E}g^2(X)}.$$

Now, we impose the **third assumption** on $C_0$ that

$$C_0 \geq C_{H.6} + \frac{2}{\varepsilon}\log(20/\varepsilon),$$

then

$$\frac{10}{\varepsilon} \exp\left(-\frac{\varepsilon}{2}(h(\rho') - C_{H.6}(\log(R) + 1) - h^\circ)\right) \leq \frac{10}{\varepsilon} \exp\left(-\frac{\varepsilon}{2}(C_0 - C_{H.6})(\log(R) + 1)\right) \leq 1/2.$$

Next, we apply (104) to the special case that $f = g$:

$$\mathbb{E}f^2(X) - \min_{\theta \in [q]}(\mathbb{E}_{\rho'}f^2)(\theta) \leq \max_{\theta, \theta' \in [q]}\left|(\mathbb{E}_{\rho'}fg)(\theta) - (\mathbb{E}_{\rho'}fg)(\theta')\right| \leq \frac{1}{2}\mathbb{E}f^2(X)$$

$$\Rightarrow \mathbb{E}f^2(X) \leq 2\min_{\theta \in [q]}(\mathbb{E}_{\rho'}f^2)(\theta).$$

Clearly, the same statemnet holds for $g$ as well. Substituting these estimates back to (104), we can conclude that when $h(\rho) \geq h^\circ + C_0(\log(R) + 1)$, any two $\mathcal{B}_{\leq \rho'}$-polynomials $f$ and $g$ with $\mathbb{E}f(X) = \mathbb{E}g(X) = 0$ satisfies

$$\max_{\theta, \theta' \in [q]} \left|(\mathbb{E}_{\rho'}fg)(\theta) - (\mathbb{E}_{\rho'}fg)(\theta')\right|$$

$$\leq \frac{20}{\varepsilon} \exp\left(-\frac{\varepsilon}{2}(h(\rho') - C_{H.6}(\log(R) + 1) - h^\circ)\right)\sqrt{\min_{\theta \in [q]}(\mathbb{E}_{\rho'}f^2)(\theta)}\sqrt{\min_{\theta \in [q]}(\mathbb{E}_{\rho'}g^2)(\theta)}$$

$$\leq \exp\left(-\frac{\varepsilon}{2}(h(\rho') - C_0(\log(R) + 1) - h^\circ)\right)\sqrt{\min_{\theta \in [q]}(\mathbb{E}_{\rho'}f^2)(\theta)}\sqrt{\min_{\theta \in [q]}(\mathbb{E}_{\rho'}g^2)(\theta)}.$$

Therefore, the theorem follows.

$$\square$$

# I Variance Estimate for degree 1 polynomial

This section is dedicated to prove Proposition C.3. Let us restate it here:

**Proposition I.1.** *There exists a constant $C = C(M, d) \geq 1$ so that the following holds: Fix $\rho' \in T$, and $0 \leq k \leq \mathrm{h}(\rho')$, then for any degree 1 function $f$ with variables $(x_u : u \in D_k(\rho'))$. There exists functions $f_u(x) = f_u(x_u)$ for $u \in D_k(\rho')$ so that the following holds:*

1. *$f(X) = \sum_{u \in D_k(\rho')} f_u(X_u)$ almost surely. (They may not agree as functions from $[q]^T$ to $\mathbb{R}$.)*

2. *For any $v \in T_{\rho'}$ with $\mathrm{h}(u) \geq k$,*

$$\sum_{u \in D_k(v)} \mathrm{Var}[f_u(X_u)] \leq CR^3 \mathrm{Var}\Big[\sum_{u \in D_k(v)} f_u(X_u)\Big].$$

**Example I.2.** Suppose $u, v \in \mathfrak{c}(\rho')$ for $u, v, \rho' \in T$ and consider

$$M = \frac{1}{2}\begin{bmatrix} 1 & 0 & 1 & 0 \\ 0 & 1 & 0 & 1 \\ 1 & 0 & 1 & 0 \\ 0 & 1 & 0 & 1 \end{bmatrix}.$$

Let us consider the function $f(x) = f_u(x) + f_v(x)$ where

$$f_u(x) = \mathbf{1}_{1,3}(x_u) = \begin{cases} 1 & \text{if } x_u \in \{1, 3\}, \\ 0 & \text{otherwise.} \end{cases}$$

and $f_v(x) = -\mathbf{1}_{1,3}(x_v)$.

Condition on $X_{\rho'} \in \{1, 3\}$, $f(X_u) + f(X_v) = 1 - 1 = 0$ condition on $X_{\rho'} \in \{1, 3\}$ and condition on $X_{\rho'} \in \{2, 4\}$, $f(X_u) + f(X_v) = 0 - 0 = 0$. Put it differntly, $\mathrm{Var}[f(X)] = 0$ since $f(X) = 0$ almost surely. However, observe that $\pi$ is the uniform measure on $[4]$, which implies

$$\mathrm{Var}[f_u(X_u)] = \mathrm{Var}[f_v(X_v)] = \frac{1}{4} > 0.$$

Therefore, it is not true that (23) holds for the standard (Efron-Stein) decomposition of $f(x) = \sum_{v \in D_k(\rho')} f_v(x_v)$.

Let us make a simple observation to give the insight for the construction. If $f(X_u)$ is a function of $X_{\mathfrak{p}(u)}$, then for each $i \in [q]$, the function $f$ must take the same value for all possible outcomes of $X_u$ conditioned on $X_{\mathfrak{p}(u)} = i$. In other words, the values of $f$ are constant on the set

$$S_i = \mathrm{supp}(\mathrm{row}_i(M)) \tag{105}$$

for every $i \in [q]$. Now, let us consider the case where $f(X_u)$ is a function of $X_{\mathfrak{p}^r(u)}$. This can be reformulated as follows: for $k \in [0, r-1]$, $\mathbb{E}\big[f(X_u) \,\big|\, X_{\mathfrak{p}^k(u)}\big]$ is a function of $X_{\mathfrak{p}^{k+1}(u)}$. Equivalently, the values of $M^k f$ are constant on the set $S_i$ for every $i \in [q]$.

Therefore, it is evident that the construction of the basis should primarily revolve around the sets $\{S_i\}_{i \in [q]}$ and their interaction with $M$.

Following from this discussion, the proof of the Proposition I.1 is divided into the following steps:

Step 1 (Section I.1): We try to give a precise description of when $f(X_u)$ is a function of $X_{\mathfrak{p}^k(u)}$ for some $k \in \mathbb{N}$. To this end, we introduce the following notation.

**Definition I.3.** *We define the following partial order relation $\leq$ on the collection of all partitions of $[q]$: Specifically, for two partitions $\mathbf{P}$ and $\mathbf{P}'$, we say that $\mathbf{P} \leq \mathbf{P}'$ if $\mathbf{P}'$ is finer than or equal to $\mathbf{P}$.*

Further, there exists $r \in \mathbb{N}$ such that $\mathbf{P}^{t,0}$ for $t \geq r$ is the trivial partition.

**Lemma I.4.** *There exists a chain of paritions*

$$\mathbf{P}^{0,0} \geq \mathbf{P}^{1,0} \geq \mathbf{P}^{2,0} \cdots \geq \mathbf{P}^{r,0} \geq \ldots$$

*A function $f : [q] \mapsto \mathbb{R}$ satisfies that $f(X_u)$ is a function of $X_{\mathfrak{p}^r(u)}$ for some $r \in \mathbb{N}$ if and only if $f$ is a linear combination of $\mathbf{1}_P$ for $P \in \mathbf{P}^{r,0}$.*

(The double index for the partitions is due to a technical reason, which will be clear in the construction of the partitions.)

Step 2 (Section I.2): Next, we try to extract a basis of functions according to the partitions fro the previous step, along with suitable quantitative estimates:

**Proposition I.5.** *Let $M$ be an ergodic and irreducible transition matrix defined on the state space $[q]$. We can construct*

- *a basis of functions from $[q]$ to $\mathbb{R}$, denoted as*

$$\{\xi_{\mathsf{w}}\}_{\mathsf{w}\in\mathsf{W}},$$

  *where $\mathsf{W}$ is a set of size $q$,*

- *a function*

$$r : \mathsf{W} \to \mathbb{N} \cup \{0\},$$

- *and a constant $C > 1$ (which depends on $M$)*

*so that the following holds:*

1. *Let*

$$r_0 := \max_{\mathsf{w}\in\mathsf{W}} r(\mathsf{w}).$$

   *There exists unique $\mathsf{w}_0 \in \mathsf{W}$ such that $r(\mathsf{w}_0) = r_0$. Moreover, $\xi_{\mathsf{w}_0} \equiv 1$.*

2. *For each $\mathsf{w} \neq \mathsf{w}_0$, $\xi_{\mathsf{w}}(X_u)$ is a function of $X_v$ where $v = \mathfrak{p}^{r(\mathsf{w})}(u)$ and $\mathbb{E}\xi_{\mathsf{w}}(X_u) = 0$.*

3.
$$\mathrm{Var}\Big[\sum_{\mathsf{w}} c_{\mathsf{w}}\xi_{\mathsf{w}}(X_u)\Big] \leq C(\max_{\mathsf{w}\,:\,r(\mathsf{w})\neq r_0} |c_{\mathsf{w}}|)^2.$$

4. *For any $0 \leq r' < r_0$ such that $\{\mathsf{w} \in \mathsf{W} : r(\mathsf{w}) = r'\}$ is not empty,*

$$\mathbb{E}\mathrm{Var}\Big[\mathbb{E}\big[\sum_{\mathsf{w}\,:\,r(\mathsf{w})=r'} c_{\mathsf{w}}\xi_{\mathsf{w}}(X_u)\,|\,X_v\big]\,\Big|\,X_{\mathfrak{p}(v)}\Big] \geq \frac{1}{C}(\max_{\mathsf{w}\,:\,r(\mathsf{w})=r'} |c_{\mathsf{w}}|)^2.$$

5. *For any $0 \leq r' < r_0$ such that $\{\mathsf{w} \in \mathsf{W} : r(\mathsf{w}) < r'\}$ is not empty,*

$$\mathbb{E}\mathrm{Var}\Big[\mathbb{E}\big[\sum_{\mathsf{w}\,:\,r(\mathsf{w})<r'} c_{\mathsf{w}}\xi_{\mathsf{w}}(X_u)\,|\,X_v\big]\,\Big|\,X_{\mathfrak{p}(v)}\Big] \leq C(\max_{\mathsf{w}\,:\,r(\mathsf{w})<r'} |c_{\mathsf{w}}|)^2.$$

**Remark I.6.** For $\mathsf{w} \in \mathsf{W}$ and $l \in [r(\mathsf{w})]$, let

$$\xi_{\mathsf{w}}^{(l)} := M^l \xi_{\mathsf{w}}, \tag{106}$$

where we treated $\xi$ as an vector in $\mathbb{R}^{[q]}$. Equivalently,

$$\xi^{(l)}(\theta) = \mathbb{E}\big[\xi(X_u)\,|\,X_v = \theta\big]$$

where $u, v \in T$ are vertices such that $v = \mathfrak{p}^l(u)$.

Step 3 (Section I.3): Finally, we will use the basis from the previous step to decompose degree-1 polynomials to prove Proposition I.1.

## I.1   Partitions of $[q]$

Let us begin with the following observation.

**Lemma I.7.** *Suppose $\{O_\alpha\}_{\alpha\in I}$ is a collection of non-empty subsets of $[q]$. Then, there exists a unique partition $\mathbf{P}$ of $[q]$ that satisfies the following 2 conditions:*

1. *For each $\alpha \in I$ and $P \in \mathbf{P}$, either $O_\alpha \in P$ or $O_\alpha \cap P = \emptyset$.*

2. *For any other partition* $\mathbf{P}'$ *that also satisfies the above property,* $\mathbf{P}' \leq \mathbf{P}$.

*Proof.* The proof can be carried out by constructing the partition $\mathbf{P}$.

Without lose of generality, we may assume the collection $\{O_\alpha\}_{\alpha \in I}$ contains $\big\{\{\theta\}\big\}_{\theta \in [q]}$, since for a singleton $\{\theta\}$ and a set $P$, it is always true that either $\{\theta\} \subseteq P$ or $\{\theta\} \cap P = \emptyset$. Consequently, we may assume

$$\bigcup_{\alpha \in I} O_\alpha = [q]. \tag{107}$$

First, we define an equivalence relation $\simeq$ on $\{O_\alpha\}_{\alpha \in I}$ as follows: For any $\alpha, \alpha' \in I$, we denote $O_\alpha \simeq O_{\alpha'}$ if there exists a chain $(\alpha_1, \alpha_2, \dots, \alpha_l)$ such that $O_{\alpha_{i-1}} \cap O_{\alpha_i} \neq \emptyset$ for $i \in [l]$. Let $I_1, \dots, I_{k_0} \subseteq I$ be the partition of $I$ such that $\{O_\alpha\}_{\alpha \in I_k}$ for $k \in [k_0]$ form the equivalence classes of the relation. Now, let $\mathbf{P} := \{P_1, \dots, P_{k_0}\}$, where

$$P_k := \cup_{\alpha \in I_k} O_k.$$

**Claim 1:** For every $\alpha \in I$ and $k \in [k_0]$, either $O_\alpha \subseteq P_k$ or $O_\alpha \cap P_k = \emptyset$.

To prove this claim, consider any $\alpha$ and $k$ described above. Suppose $O_\alpha \cap P_k \neq \emptyset$. Let $\theta \in O_\alpha \cap P_k$ and pick an index $\alpha' \in I_k$ such that $\theta \in O_{\alpha'}$. Such an index exists because $P_k = \bigcup_{\alpha'' \in I_k} O_{\alpha''}$. Then, we have $O_\alpha \cap O_{\alpha'} \neq \emptyset$, implying $\alpha \in I_k$. Consequently, $O_\alpha \subseteq P_k$. Therefore, the claim is proven.

**Claim 2:** $\mathbf{P}$ is a partition of $[q]$.

We need to verify three properties:

1. $\bigcup_{k \in [k_0]} P_k = [q]$,

2. $\forall k \in [k_0]$, $P_k \neq \emptyset$, and

3. $P_k \cap P_{k'} = \emptyset$ whenever $k \neq k'$.

First, for each $\theta \in [q]$, by (107), there exists $\alpha \in I$ such that $\theta \in O_\alpha$. Then, $\theta \in O_\alpha \in P_k$ where $k$ is the index such that $\alpha \in I_k$. Hence, we conclude that $\bigcup_{k \in [k_0]} P_k = [q]$.

Second, for each $k \in [k_0]$, let $\alpha \in I_k$. We have $\emptyset \neq O_\alpha \subseteq P_k$. Thus, $P_k$ is not an empty set.

Finally, for any distinct $k, k' \in [k_0]$, suppose $\theta \in P_k \cap P_{k'}$. By (107), let $\alpha \in I$ be the index so that $\theta \in O_\alpha$. Hence, both $O_\alpha \cap P_k$ and $O_\alpha \cap P_{k'}$. In particular, it is necessary that $\alpha \in I_k$ and $\alpha \in I_{k'}$, which forces $k = k'$, leading to a contradiction. Therefore, $P_k \cap P_{k'} = \emptyset$ whenever $k \neq k'$. Hence, the claim follows.

**Claim 3:** $\mathbf{P}' \leq \mathbf{P}$ for any $\mathbf{P}'$ described in the statement.

To prove the claim, it suffices to show that for any $P' \in \mathbf{P}'$ and $P_k \in \mathbf{P}$ with $k \in [k_0]$, if $P' \cap P_k \neq \emptyset$, then $P_k \subseteq P'$.

Let us consider an arbitrary pair of $P' \in \mathbf{P}'$ and $P_k \in \mathbf{P}$ and assume that $P' \cap P_k \neq \emptyset$. There exists an index $\alpha$ such that $O_\alpha \cap P' \cap P_k \neq \emptyset$. Based on the assumptions regarding $\mathbf{P}$ and $\mathbf{P}'$, we have $\alpha \in I_k$ and $O_\alpha \subseteq P'$.

For every other $\alpha' \in I_k$, there exists a chain $(\alpha = \alpha_0, \alpha_1, \dots, \alpha_{l_0} = \alpha')$ such that $O_{\alpha_{l-1}} \cap O_{\alpha_l} \neq \emptyset$ for $l \in [l_0]$. Observe that if $O_{\alpha_{l-1}} \subseteq P'$, then $O_{\alpha_l} \subseteq P'$, due to $O_{\alpha_l} \cap P' \supseteq O_{\alpha_l} \cap O_{\alpha_{l-1}} \neq \emptyset$. With $O_0 \subseteq P'$ as our starting point, we can apply this observation repeatedly to conclude that $O_{\alpha'} \subset P'$. Since the argument works for every $\alpha' \in I_k$, we conclude that $P_k = \bigcup_{\alpha'' \in I_k} O_{\alpha''} \subseteq P'$.

$\square$

**Definition I.8.** *For any given collection of subsets* $\{O_\alpha\}_{\alpha \in I}$ *of* $[q]$, *let* $\mathbf{P}(\{O_\alpha\}_{\alpha \in I})$ *denote the partition* $\mathbf{P}$ *defined in Lemma I.7.*

*For any given partition $\mathbf{Q}$ of $[q]$, let*

$$\mathbf{P}_{\mathrm{SC}}(\mathbf{Q}) := \mathbf{P}\big(\{Q\}_{Q\in\mathbf{Q}} \cup \{S_i\}_{i\in[q]}\big).$$

**Remark I.9.** Clearly, $\mathbf{P}_{\mathrm{SC}}(\mathbf{Q}) \leq \mathbf{Q}$.

**Definition I.10.** *Let*

$$\mathbf{P}^{0,0} = \{\{1\},\{2\},\ldots,\{q\}\}$$

*and*

$$\mathbf{P}^{1,0} = \mathbf{P}_{SC}(\mathbf{P}^{0,0}).$$

Let us remark that $\mathbf{P}^{1,0}$ is the finest partition of $[q]$ so that each part $P \in \mathbf{P}^{1,0}$ either contains $S_i$ or disjoint from $S_i$ for $i \in [q]$.

We use double indices for indexing the partitions because constructing such a chain of partitions requires the creation of multiple partitions along the way, as we will illustrate shortly.

To proceed, let us begin with a simple observation.

**Lemma I.11.** *If $P \in \mathbf{P}^{1,0}$, then*

$$M\mathbf{1}_P = \mathbf{1}_Q$$

*where*

$$Q = \{i \in [q] \ : \ S_i \subseteq P\}.$$

*Suppose $\mathbf{P}^{1,0} = \{P_1, P_2, \ldots, P_{k_0}\}$. Then, the collection $\mathbf{Q} := \{Q_1, Q_2, \ldots, Q_{k_0}\}$ where*

$$M\mathbf{1}_{P_i} = \mathbf{1}_{Q_i}$$

*is also a partition provided that $M$ is irreducible.*

*Proof.* For $i$ with $S_i \cap P = \emptyset$, it is immediate that $(M\mathbf{1}_P)_i = 0$. Conversely, when $S_i \cap P \neq \emptyset$, it is necessary that $S_i \subseteq P$. Consequently, $(M\mathbf{1}_P)_i = \sum_{j\in[q]} M_{ij} = 1$.

To establish that $\mathbf{Q}$ is a partition, we need to demonstrate the following three conditions:

1. $Q_k \cap Q_{k'} = \emptyset$ for all distinct $k, k' \in [k_0]$.

2. $\bigcup_{k\in[k]} Q_k = [q]$.

3. $Q_k \neq \emptyset$ for all $k \in [k_0]$.

For the first condition, suppose there exists $i \in Q_k \cap Q_{k'}$ for some distinct $k$ and $k'$. By definition, $S_i \subseteq P_k$ and $S_i \subseteq P_{k'}$, which is a contradiction. Hence, $Q_k \cap Q_{k'} = \emptyset$.

For the second condition, for every $i \in [q]$, we know that $S_i \subseteq P_k$ for some $k$. Consequently, $i \in Q_k$, ensuring $\bigcup_{\alpha\in[k]} Q_k = [q]$.

For the third condition, if we assume $Q_k = \emptyset$, implying that no $i \in [q]$ satisfies $S_i \subseteq P_k$, then $M$ is not irreducible, since the states in $P_k$ cannot be reached. $\qquad\square$

**Definition I.12.** *Let $\mathbf{P}^{1,1} = \mathbf{Q}$ where $\mathbf{Q}$ is the partition described in Lemma I.11.*

**Lemma I.13.** *If $P$ is a finite union of parts in $\mathbf{P}^{1,0}$, then*

$$M\mathbf{1}_P = \mathbf{1}_Q \tag{108}$$

*where $Q$ is a finite union of parts in $\mathbf{P}^{1,1}$. The above map induces a bijection between subsets of $[q]$ that are finite union of parts of $\mathbf{P}^{1,0}$ and subsets of $[q]$ that are finite union of parts of $\mathbf{P}^{1,1}$, in which preserve the inclusion relation is preserved.*

*Proof.* Let us express $\mathbf{P}^{1,0} = \{P_1, P_2, \ldots, P_{[k_0]}\}$ and $\mathbf{P}^{1,1} = \{Q_1, Q_2, \ldots, Q_{k_0}\}$ where $\mathbf{1}_{Q_k} = M\mathbf{1}_{P_k}$.

For each $I \subseteq [k_0]$, let $P_I = \bigcup_{k\in I} P_k$ and $Q_I = \bigcup_{k\in I} Q_k$. Since $\mathbf{1}_{P_I} = \sum_{k\in I} \mathbf{1}_{P_k}$ and $\mathbf{1}_{Q_I} = \sum_{k\in I} \mathbf{1}_{Q_k}$, clearly we have

$$\mathbf{1}_{Q_I} = M\mathbf{1}_{P_I}.$$

Since naturally both finite union of parts of $P$ and of $Q$ are identified with a subset $I \subset [k_0]$ in the above way, the statement of the lemma follows. $\qquad\square$

An immediate consequence is the following.

**Corollary I.14.** *The transition matrix $M$ induces a bijection between partitions that are $\leq \mathbf{P}^{1,0}$ and partitions that are $\leq \mathbf{P}^{1,1}$. For convenience, we adopt the following definitions:*

1. *For any partition $\mathbf{P}$ such that $\mathbf{P} \leq \mathbf{P}^{1,0}$, define*

$$M\mathbf{P} := \{Q \; : \; \exists P \in \mathbf{P} \text{ such that } \mathbf{1}_Q = M\mathbf{1}_P\} \leq \mathbf{P}^{1,1}.$$

2. *Given any $\mathbf{P} \leq \mathbf{P}^{1,0}$ and for each $P \in \mathbf{P}$, let $MP$ represent a part in $M\mathbf{P}$ where*

$$\mathbf{1}_{MP} = M\mathbf{1}_P.$$

Next, we will build a collection of partitions $\mathbf{P}^{r,s}$ for $r \geq 0$ and $0 \leq s \leq r$ starting with $\mathbf{P}^{0,0} = \big\{\{1\}, \{2\}, \ldots, \{q\}\big\}$ and establishing the relationship illustrated by the diagram below.

$$
\begin{array}{ccccccccc}
\mathbf{P}^{0,0} & \underset{SC}{\geq} & \mathbf{P}^{1,0} & \geq & \mathbf{P}^{2,0} & \geq & \mathbf{P}^{3,0} & \geq & \mathbf{P}^{4,0} \quad \ldots \\
& & \downarrow & & \downarrow & & \downarrow & & \downarrow \\
& & \mathbf{P}^{1,1} & \underset{SC}{\geq} & \mathbf{P}^{2,1} & \geq & \mathbf{P}^{3,1} & \geq & \mathbf{P}^{4,1} \quad \ldots \\
& & & & \downarrow & & \downarrow & & \downarrow \\
& & & & \mathbf{P}^{2,2} & \underset{SC}{\geq} & \mathbf{P}^{3,2} & \geq & \mathbf{P}^{4,2} \quad \ldots \\
& & & & & & \downarrow & & \downarrow \\
& & & & & & \mathbf{P}^{3,3} & \underset{SC}{\geq} & \mathbf{P}^{4,3} \quad \ldots \\
& & & & & & & & \downarrow \\
& & & & & & & & \mathbf{P}^{4,4} \quad \ldots \\
& & & & & & & & \ddots
\end{array}
$$

( In the above diagram, $\mathbf{P} \to \mathbf{Q}$ indicates that $\mathbf{Q} = M\mathbf{P}$; $\mathbf{Q} \underset{SC}{\geq} \mathbf{P}$ indicates $\mathbf{P} = \mathbf{P}_{\text{SC}}(\mathbf{Q})$.)

Indeed, the initial definition of $\mathbf{P}^{0,0}$ and the relation diagram determine the collection of partitions completely. Let us summarise it as a statement:

**Lemma I.15.** *There exists a unique collection of partitions $\{\mathbf{P}^{r,s}\}_{r \geq s \geq 0}$ that satisfies the following properties: For $0 \leq s < r$,*

1. $\mathbf{P}^{0,0} = \big\{\{1\}, \{2\}, \ldots, \{q\}\big\}$.

2. $\mathbf{P}^{r,s} \leq \mathbf{P}^{1,0}$.

3. $\mathbf{P}^{r,s+1} = M\mathbf{P}^{r,s}$.

4. $\mathbf{P}^{r+1,s} \leq \mathbf{P}^{r,s}$.

5. $\mathbf{P}^{r+1,r} = \mathbf{P}_{\text{SC}}(\mathbf{P}^{r,r})$.

*Proof of Lemma I.15.* The proof is proceeded by induction. We assume that $\mathbf{P}^{r,s}$ is constructed and uniquely determined for $0 \leq r < r_0$ and $0 \leq s \leq r$ for some $r_0 \geq 0$ so that it satisfies the properties described in the lemma.

We will define the partitions in the next column $\{\mathbf{P}^{r_0,s}\}_{s \in [0,r_0]}$ by starting with $\mathbf{P}^{r_0,r_0-1} = \mathbf{P}_{\text{SC}}(\mathbf{P}^{r_0-1,r_0-1})$.

Besides constructing the rest of partitions, we also need to show that these partitions satisfy the following **list of conditions** ( let us denote it as **List A**): For $s \in [0, r_0, -1]$,

1. $\mathbf{P}^{r_0,s} \leq \mathbf{P}^{1,0}$.

2. $\mathbf{P}^{r_0,s} \leq \mathbf{P}^{r_0-1,s}$ for $s \in [0, r_0 - 1]$.

3. $\mathbf{P}^{r_0,s+1} = M\mathbf{P}^{r_0,s}$.

By definition of the map $\mathbf{P}_{\mathrm{SC}}$, the first and second condition in the list are satisfied for $s = r_0 - 1$. Relying on $\mathbf{P}^{r_0, r_0 - 1} \leq \mathbf{P}^{1,0}$, we can define $\mathbf{P}^{r_0, r_0} = M\mathbf{P}^{r_0, r_0 - 1}$. Hence, the third condition in the list is also satisfied for $s = r_0 - 1$.

It remains to construct $\mathbf{P}^{r_0, s}$ for $s \in [0, r_0 - 2]$ and they satisfy those 3 conditions in the list. This can be proceeded inductively starting from $s = r_0 - 2$.

**Claim**: For $s \in [0, r_0 - 2]$, if $\mathbf{P}^{r_0, s+1} \leq \mathbf{P}^{r_0 - 1, s+1}$, then there exists a unique partition $\mathbf{P}^{r_0, s}$ which satisfies the conditions in **List A** for $s$.

Suppose the Claim holds. With $\mathbf{P}^{r_0, r_0 - 1} \leq \mathbf{P}^{r_0 - 1, r_0 - 1}$, we could apply the claim repeatedly and the lemma follows. The rest of the proof is to show the claim holds.

Let us assume $\mathbf{P}^{r_0, s+1} \leq \mathbf{P}^{r_0 - 1, s+1}$ for some $s \in [0, r_0 - 2]$. First, from our assumption on $\{\mathbf{P}^{r, s}\}$ for $0 \leq s \leq r_0 - 1$, $\mathbf{P}^{r_0 - 1, s+1} = M\mathbf{P}^{r_0 - 1, s}$. By Corollary I.14, $\mathbf{P}^{r_0 - 1, s+1} \leq \mathbf{P}^{1,1}$. Since $\mathbf{P}^{r_0, s+1} \leq \mathbf{P}^{r_0 - 1, s+1}$, we conclude that $\mathbf{P}^{r_0, s+1} \leq \mathbf{P}^{1,1}$.

Applying Corollary I.14 again, we know there exists an unique partition $\mathbf{P} \leq \mathbf{P}^{1,0}$ so that $\mathbf{P}^{r_0, s+1} = M\mathbf{P}$. We set $\mathbf{P}^{r_0, s} := \mathbf{P}$. In particular, the choice of $\mathbf{P}^{r_0, s}$ is unique in order to satisfy the first and third condition from the list.

It remains to show that $\mathbf{P}^{r_0, s}$ also satisfies the second condition in **List A**. Notice that from Corollary I.14, the induced map of $M$ on partitions preserves $\leq$ relation. Hence, $\mathbf{P}^{r_0, s+1} \leq \mathbf{P}^{r_0 - 1, s+1}$ implies $\mathbf{P}^{r_0 - 1, s} \leq \mathbf{P}^{r_0 - 1, s}$. Therefore, the claim holds.

$\square$

*Proof of Lemma I.4.* We start with the proof on the $\Rightarrow$ implication. Suppose $f$ is a function satisfied the first condition described in the lemma.

Since $f(X_u)$ is a function of $X_{\mathfrak{p}^r(u)}$, this is equivalent to

$$0 = \mathbb{E}\left[\mathrm{Var}\left[f(X_u) \,\middle|\, X_{\mathfrak{p}^r(u)}\right]\right].$$

Relying on the identity $\mathrm{Var}[Y] = \mathbb{E}\mathrm{Var}[Y \mid Z] + \mathrm{Var}\left[\mathbb{E}[Y \mid Z]\right]$ and $(X_{\mathfrak{p}^r(u)}, X_{\mathfrak{p}^{r-1}(u)}, \ldots, X_u)$ is a Markov Chain,

$$\mathbb{E}\left[\mathrm{Var}\left[f(X_u) \,\middle|\, X_{\mathfrak{p}^r(u)}\right]\right] = \sum_{s=1}^{r} \mathbb{E}\left[\mathrm{Var}\left[f(X_u) \,\middle|\, X_{\mathfrak{p}^s(u)}\right]\right].$$

Hence, $\mathbb{E}\left[\mathrm{Var}\left[f(X_u) \,\middle|\, X_{\mathfrak{p}^s(u)}\right]\right] = 0$ for $s \in [r]$, which in turn implies $\mathbb{E}\left[f(X_u) \,\middle|\, X_{\mathfrak{p}^{s-1}(u)}\right]$ conditioned on $X_{\mathfrak{p}^s(u)}$ is a constant function for each $s \in [r]$. Equivalently, $M^{s-1}f$ takes the same value for all elements in each $S_i$ for $i \in [q]$.

**Claim:** For $s \in [r]$, if $f$ can expressed in the form $f = \sum_{P \in \mathbf{P}^{s-1,0}} c_{s-1,P} \mathbf{1}_P$, then it can be expressed in the form $f = \sum_{P \in \mathbf{P}^{s,0}} c_{s,P} \mathbf{1}_P$.

Clearly, if the claim holds, then we can apply it repeatedly to draw the conclusion that $f$ is a linear combination of $\mathbf{1}_P$ for $P \in \mathbf{P}^{r,0}$.

Now, we fix $s \in [r]$ and assume $f = \sum_{P \in \mathbf{P}^{s-1,0}} c_{s-1,P} \mathbf{1}_P$. Then,

$$\mathbb{E}\left[f(X_u) \,\middle|\, X_{\mathfrak{p}^{s-1}(u)} = a\right] = (M^{s-1}f)(a) = \sum_{P \in \mathbf{P}^{s-1,0}} c_{s-1,P} M^{s-1} \mathbf{1}_P = \sum_{P \in \mathbf{P}^{s-1,0}} c_{s-1,P} \mathbf{1}_{P^{s-1}},$$

where for each $P \in \mathbf{P}^{s-1,0}$, $P^{s-1} \in \mathbf{P}^{s-1,s-1}$ is the corresponding part such that $M^{s-1}\mathbf{1}_P = \mathbf{1}_{P^{s-1}}$. In other words, $M^{s-1}f$ is a linear combination of $\mathbf{1}_P$ for $P \in \mathbf{P}^{s-1,s-1}$.

Because $M^{s-1}f$ takes the same value not only for all elements in each $S_i$ for $i \in [q]$, but also for all elements in each $P$ for $P \in \mathbf{P}^{s-1,s-1}$, it implies $M^{s-1}f$ takes the same value for all elements in each $P' \in \mathbf{P}_{\mathrm{SC}}(\mathbf{P}^{s-1,s-1}) = \mathbf{P}^{s,s-1}$.

Together with the fact that the induced map of $M$ on partitions preserves $\leq$ relation, we conclude that $c_{s-1,P_1} = c_{s-1,P_2}$ for $P_1, P_2 \in P^{s-1,0}$ whenever $P_1$ and $P_2$ are both contained in some $P \in \mathbf{P}^{s,0}$.

Equivalently, within each $P \in \mathbf{P}^{s,0}$, $f$ is a constant function. Hence, we can express $f$ as a linear combination of $\mathbf{1}_P$ for $P \in \mathbf{P}^{s,0}$.

For the $\Leftarrow$ implication, suppose $f$ is a linear combination of $\mathbf{1}_P$ with $P \in \mathbf{P}^{r,0}$.

What we need to show is for $s \in [0, r-1]$, $M^s f$ takes the same values for all elements in each $S_i$ for $i \in [q]$. From the chain $\mathbf{P}^{r,0} \to \mathbf{P}^{r,1} \to \cdots \to \mathbf{P}^{r,r}$ and by (108), for $s \in [0, r-1]$, $M^s f$ is a linear combination of $\mathbf{1}_P$ with $P \in \mathbf{P}^{r,s}$.

Since $\mathbf{P}^{s+1,s} = \mathbf{P}_{\mathrm{SC}}(bP^{s,s}) \leq \mathbf{P}^{1,0}$ and $\mathbf{P}^{s+1,s} \geq \cdots \geq \mathbf{P}^{r,s}$, we have $\mathbf{P}^{r,s} \leq \mathbf{P}^{1,0}$, which implies $M^s f$ takes the same values for all elements in each $S_i$ for $i \in [q]$. Therefore, the proof is completed.

Now, it remains to prove the second statement of the lemma.

First, if there exists $r \in \mathbb{N}$ such that $\mathbf{P}^{r,0}$ is trivial. Then $\mathbf{P}^{t,0}$ is also trivial for $t > r$ since $\mathbf{P}^{t,0} \leq \mathbf{P}^{r,0}$. Hence, it is enough to show the existence of $r$ such that $\mathbf{P}^{r,0}$ is trivial. $\mathbf{P}^{r,0}$ is trivial.

From the assumption on $M$, we knew that the stationary distribution $\pi$ of $M$ satisfies $\min_{i \in [q]} \pi(i) > 0$ and $M^r$ converges entry-wise to the matrix whose row is identically $\pi$. Therefore, for sufficiently large $r$, $\min_{i,j \in [q]}(M^r)_{ij} > 0$.

Now, let us fix such $r$ and assume $\mathbf{P}^{r,0}$ is not trivial. Let us express $\mathbf{P}^{r,s} = \{P_1^{r,s}, \ldots, P_{k_r}^{r,s}\}$ with for $s \in [0, r]$ and $k_r \geq 2$ where the index is assigned so that $P_k^{r,s} = MP_k^{r,s-1}$ for $s \in [r]$ and $k \in [k_r]$. First,

$$\mathbf{1}_{P_1^{r,r}} = M^r \mathbf{1}_{P_1^{r,0}}.$$

With $\mathbf{1}_{P_1^{r,0}}$ is non-negative and not zero, every component of $M^r \mathbf{1}_{P_1^{r,0}}$ is non-zero. This forces $P_1^{r,r} = [q]$, which contradicts to the assumption that $\mathbf{P}^{r,r}$ is non-trivial.

$\square$

## I.2 A basis of functions from $[q] \mapsto \mathbb{R}$ according to the partition

From now on, let $r_0$ be the smallest non-negative integer such that $\mathbf{P}^{r,0}$ is trivial. Consider the collection

$$\big\{(P, s) \ : \ s \in [0, r_0], P \in \mathbf{P}^{s,0}\big\}$$

We will establish an identification between elements of the set described above and words whose alphabet consists of non-negative integers. This identification is constructed through induction, following these steps:

- First, we identify $([q], r_0)$ with the word $(1)$.
- Assuming that elements in $\{(P, s+1) \ : \ P \in \mathbf{P}^{s+1,0}\}$ have already been identified with unique words, we proceed as follows: For each $(P, s+1)$, suppose there are $k$ pairs of $(P', s)$ such that $P' \subseteq P$. We identify these $k$ pairs with the words $(\mathsf{w}, \mathsf{i})$ for $\mathsf{i} \in [0, k-1]$, in any order of preference. For each $(P', s)$, due to $\mathbf{P}^{s,0}$ is a finer than or equal to $\mathbf{P}^{s+1,0}$, there exists an unique pair $(P, s+1)$ so that $P' \subseteq P$. This guarantees the above procedure assigns each $(P', s)$ a unique word.

We denote the set of words described above as $\widetilde{\mathsf{W}}$, and we adopt the notation $\mathsf{w} \sim (P, s)$ to indicate that $(P, s)$ is associated with the word $\mathsf{w}$. For a given $\mathsf{w} \in \widetilde{\mathsf{W}}$, we represent the corresponding pair as $(P_\mathsf{w}, r(\mathsf{w}))$, where $r(\mathsf{w}) = r_0 + 1 - \mathrm{len}(\mathsf{w})$.

Now, let us make the following observations

1. If $\mathsf{w} \in \widetilde{\mathsf{W}}$ is a word with $\mathrm{len}(\mathsf{w}) < r_0 + 1$, then $(\mathsf{w}, 0) \in \widetilde{\mathsf{W}}$.

2. Each $(P, s)$ corresponds to a word of length $r_0 + 1 - s$.

3. Suppose $\mathsf{w}, \mathsf{w}' \in \widetilde{\mathsf{W}}$ such that $\mathsf{w}$ is a prefix of $\mathsf{w}'$. Then, $P_{\mathsf{w}'} \subseteq P_\mathsf{w}$.

Let $T_{\widetilde{\mathsf{W}}}$ be the tree defined on $\widetilde{\mathsf{W}}$ using the prefix relation. In this tree, edges are drawn from $w'$ to $w$ if $r(\mathsf{w}') = r(\mathsf{w}) + 1$ and $P_{\mathsf{w}} \subseteq P_{\mathsf{w}'}$. Now, we will select $q$ parts from these elements $(P, s)$ based on their corresponding words.

**Lemma I.16.** *Let $\mathsf{W} \subseteq \widetilde{\mathsf{W}}$ be the subcollection of words which end with a positive integer. Then,* $|\mathsf{W}| = q$.

*Proof.* First of all, there are exactly $q$ words in $\widetilde{\mathsf{W}}$ with length $r + 1$, since $\mathbf{P}^{0,0} = \left\{\{i\}\right\}_{i \in [q]}$ has $q$ parts. For each $i \in [q]$, let $\mathsf{w}'_i$ be the word corresponding to $(\{i\}, 0)$ and let $\mathsf{w}_i$ be the longest word ending with a positive integer so that is either a prefix of equals to $\mathsf{w}'_i$. This is well-defined since every word in $\widetilde{\mathsf{W}}$ is a word starting with 1.

The proof of the lemma follows if we can show the following **claim**: $\mathsf{w}_1, \mathsf{w}_2, \ldots, \mathsf{w}_q$ are distinct and are all words which ends with a positive integer.

To prove the claim, we begin by showing $\mathsf{w}_i \neq \mathsf{w}_j$ whenever $i \neq j$. Suppose $\mathsf{w}_i = \mathsf{w}_j$ for some distinct pair of $i, j \in [q]$. Let $\tilde{\mathsf{w}}$ be the longest prefix of $\mathsf{w}'_1, \mathsf{w}'_2$, necessarily we have $\mathsf{w}_i = \mathsf{w}_j$ is either a prefix of $\tilde{\mathsf{w}}$ or $\tilde{\mathsf{w}}$ itself. Further, the length of $\tilde{\mathsf{w}}$ is less equal than $r$, since otherwise it implies $\mathsf{w}'_i = \mathsf{w}'_j$, which is a contradiction.

Now, let $(\tilde{\mathsf{w}}, \mathsf{e}_i)$ and $(\tilde{\mathsf{w}}, \mathsf{e}_j)$ be the two words which are prefix of $\mathsf{w}'_i$ and $\mathsf{w}'_j$, respectively. From the definition that $\tilde{\mathsf{w}}$ is the longest common prefix, $\mathsf{e}_i$ and $\mathsf{e}_j$ are distinct non-negative integers. Since $\mathsf{w}_i$ is a prefix of $(\mathsf{w}, \mathsf{e}_i)$, it is necessary that $\mathsf{e}_i = 0$, otherwise it violates the definition of $\mathsf{w}_i$. For the same reason, $\mathsf{e}_j = 0$. Therefore, we reach a contradiction.

The remaining part to prove the claim is to show that $\{\mathsf{w}_i\}_{i \in [q]}$ are all the words in $\widetilde{\mathsf{W}}$ ending with a positive integer. Suppose $\mathsf{w}$ is a word in which ends with a positive integer. If $\mathrm{len}(\mathsf{w}) < r + 1$, we can keep fill 0 until its length is $r + 1$ and denote the resulting word by $\mathsf{w}'$. Observe that $\mathsf{w}' \in \widetilde{\mathsf{W}}$. Together with the length of $\mathsf{w}'$ is $r + 1$, necessarily $\mathsf{w}' = \mathsf{w}'_i$ for some $i$. Recall the definition of $\mathsf{w}_i$, we conclude $\mathsf{w} = \mathsf{w}_i$. Therefore, the claim follows.

$\square$

**Lemma I.17.** *For any given $0 \leq r' < r$, suppose $\mathsf{W}_{r'} := \{\mathsf{w} \in \mathsf{W} : r(\mathsf{w}) = r'\}$ is non-empty. Consider a linear combination $\sum_{\mathsf{w} \in \mathsf{W}_{r'}} c_{\mathsf{w}} \mathbf{1}_{P_{\mathsf{w}}}$. If it can be expressed as a linear combination of $\mathbf{1}_P$ for $P \in \mathbf{P}^{r'+1,0}$, then $c_{\mathsf{w}}$ are identically 0.*

*Proof.* Let $\mathsf{w}_1, \ldots, \mathsf{w}_{k_0}$ be the words with $r(\mathsf{w}_k) = r' + 1$ and corresponding to each part of $\mathbf{P}^{r'+1,0}$. Then, the words that corresponds to pairs of the form $(P, r')$ with $P \in \mathbf{P}^{r',0}$ are

$$\left\{(\mathsf{w}_k, \mathsf{t})\right\}_{k \in [k_0], \mathsf{t} \in [0, \mathsf{t}_k]}$$

where $\mathsf{t}_k$ are non-negative integers. Now, we express

$$\sum_{\mathsf{w} \in \mathsf{W}_{r'}} c_{\mathsf{w}} \mathbf{1}_{P_{\mathsf{w}}} = \sum_{k \in [k_0]} \sum_{\mathsf{t} \in [\mathsf{t}_k]} c_{(\mathsf{w}_k, \mathsf{t})} \mathbf{1}_{P_{(\mathsf{w}_k, \mathsf{t})}}.$$

For each $k \in [k_0]$ and any $\theta \in P_{\mathsf{w}_k}$, we have

$$\sum_{k' \in [k_0]} \sum_{\mathsf{t} \in [\mathsf{t}_{k'}]} c_{(\mathsf{w}_{k'}, \mathsf{t})} \mathbf{1}_{P_{(\mathsf{w}_{k'}, \mathsf{t})}}(\theta) = \sum_{\mathsf{t} \in [\mathsf{t}_k]} c_{(\mathsf{w}_k, \mathsf{t})} \mathbf{1}_{P_{(\mathsf{w}_k, \mathsf{t})}}(\theta).$$

Therefore, $\sum_{\mathsf{w} \in \mathsf{W}_{r'}} c_{\mathsf{w}} \mathbf{1}_{P_{\mathsf{w}}}$ can be expressed as $\sum_{k \in [k_0]} c_{\mathsf{w}_k} \mathbf{1}_{P_{\mathsf{w}_k}}$ if and only if $\sum_{\mathsf{t} \in [\mathsf{t}_k]} c_{(\mathsf{w}, \mathsf{t})} \mathbf{1}_{P_{(\mathsf{w}, \mathsf{t})}}$ is a constant on $P_{\mathsf{w}_k}$.

For each $k \in [k_0]$, let $\theta \in P_{(\mathsf{w}_k, 0)}$, then we have

$$\sum_{k' \in [k_0]} \sum_{\mathsf{t} \in [\mathsf{t}_{k'}]} c_{(\mathsf{w}_{k'}, \mathsf{t})} \mathbf{1}_{P_{(\mathsf{w}_{k'}, \mathsf{t})}}(\theta) = \sum_{\mathsf{t} \in [\mathsf{t}_k]} c_{(\mathsf{w}_k, \mathsf{t})} \mathbf{1}_{P_{(\mathsf{w}_k, \mathsf{t})}}(\theta) = 0,$$

which forces $c_{(\mathsf{w}_k, \mathsf{t})} = 0$ for every $\mathsf{t} > 0$ (if it exists). Therefore, the proof is complete. $\square$

**Definition I.18.** *Let $\mathfrak{B} := \{\xi_{\mathsf{w}}\}_{\mathsf{w} \in \mathsf{W}}$ be a collection of $q$ functions from $[q]$ to $\mathbb{R}$, defined as follows:*

1. *If* $w = (1)$, $\xi_w = \mathbf{1}_{P_w} = 1$.

2. *If* $w \neq (1)$,

$$\xi_w(\theta) := \mathbf{1}_{P_w}(\theta) - \mathbb{E}_{Y \sim \pi} \mathbf{1}_{P_w}(Y).$$

**Remark I.19.** The remaining goal in this subsection is to show that $\mathfrak{B}$ is the desired basis described in Proposition I.5. We also remark that the first two properties stated in Proposition I.5 are already satisfied with this construction: $\operatorname{argmax}_{w \in W} r(w) = (1)$ with $\xi_{(1)} = \mathbf{1}_{[q]} = 1$; $\xi_w(X_u)$ is a function of $X_v$ where $v = \mathfrak{p}^{r(w)}(u)$.

**Lemma I.20.** *The collection $\mathfrak{B}$ forms a linear basis for functions from $[q]$ to $\mathbb{R}$.*

*Proof.* Since there are exactly $q$ functions, our goal is to show

$$\mathbb{R}^{[q]} = \operatorname{span}(\{\xi_w\}_{w \in W}),$$

and the R.H.S. is the same as $\operatorname{span}(\{\mathbf{1}_{P_w}\}_{w \in W})$. It suffices to show for each $i \in [q]$, $\mathbf{1}_{\{i\}}$ can be expressed as a linear combination of $\mathbf{1}_{P_w}$ with $w \in W$.

To prove this statement, we will use induction, showing that for $s$ from $r_0$ to 0, each $\mathbf{1}_P$ with $P \in \mathbf{P}^{s,0}$ can be expressed as a linear combination of of $\mathbf{1}_{P_w}$ with $w \in W$. Since $\mathbf{P}^{0,0} = \{\{1\}, \ldots, \{q\}\}$, the proof follows once we establish this inductive statement.

First, when $s = r$, since $\mathbf{1}_{[q]}$ is the only part in $\mathbf{P}^{r_0,0}$ and $[q] = P_{(1)}$, the statement holds for $s = r_0$.

Now, suppose the inductive hypothesis holds for $s+1$ with $s < r_0$. Pick any $P \in \mathbf{P}^{s,0}$, let $w = (w', t)$ be the word associate with $(P, s)$. If $t = 0$, then

$$\mathbf{1}_P = \mathbf{1}_{P_{w'}} - \sum_{P''} \mathbf{1}_{P''}$$

where the sum is taken over all parts $P'' \in \mathbf{P}^{s,0} \setminus \{P\}$ contained in $P_{w'}$. Each $P''$ in the summation (if it exists) must corresponds to a word of the form $(w', t'')$ with $t'' > 0$, or equivalently $(w', t'') \in W$. From the induction hypothesis, $\mathbf{1}_{P_{w'}}$ is a linear combination of $\mathbf{1}_{P_w}$ with $w \in W$. Therefore, we conclude that $\mathbf{1}_P$ is also a linear combination of $\mathbf{1}_{P_w}$ with $w \in W$. If $t > 0$, then $w \in W$, and the same conclusion follows immediately. With no restriction on the choice of $P$, the induction hypothesis holds for $s$ as well.

Therefore, the lemma follows from induction. $\qquad\square$

**Lemma I.21.** *For any given $0 \leq r' < r$, suppose $W_{r'} := \{w \in W : r(w) = r'\}$ is non-empty. Then, there exists a constant $C \geq 1$ (which could depends on $M$) such that the following holds: Let $u, v \in T$ be two vertices such that $v = \mathfrak{p}^{r'}(u)$. We have*

$$\mathbb{E}\operatorname{Var}\Big[\mathbb{E}\Big[\sum_{w \in W_{r'}} c_w \xi_w(X_u) \,\big|\, X_v\Big] \,\Big|\, X_{\mathfrak{p}(v)}\Big] \geq \frac{1}{C}(\max_{w \in W_{r'}} |c_w|)^2. \tag{109}$$

*Proof.* First, both sides of (109) scale by a factor $h^2$ if every term $c_w$ is multiplied by $h \in \mathbb{R}$. Hence, it suffices to establish the inequality in the case

$$\max_{w \in W_{r'}} |c_w| = 1.$$

Given this, consider the set $\big\{(c_w)_{w \in W_{r'}} : \max_{w \in W_{r'}} |c_w| = 1\big\} \subseteq \mathbb{R}^{W_{r'}}$. It is compact set and

$$\mathbb{E}\operatorname{Var}\Big[\mathbb{E}\Big[\sum_{w \in W_{r'}} c_w \xi_w(X_u) \,\big|\, X_v\Big] \,\Big|\, X_{\mathfrak{p}(v)}\Big] \tag{110}$$

is continuous in $(c_w)_{w \in W_{r'}}$ (it is a polynomial of $c_w$). By a compact argument one can estalbish the existence of $C \geq 1$ described in the lemma if for every $(c_w)_{w \in W_{r'}}$ with $\max_{w \in W_{r'}} |c_w| = 1$,

$$\mathbb{E}\operatorname{Var}\Big[\mathbb{E}\Big[\sum_{w \in W_{r'}} c_w \xi_w(X_u) \,\big|\, X_v\Big] \,\Big|\, X_{\mathfrak{p}(v)}\Big] > 0.$$

We can simplify this by observing that
$$\sum_{\mathsf{w}\in\mathsf{W}_{r'}} c_{\mathsf{w}}\xi_{\mathsf{w}} = \sum_{\mathsf{w}\in\mathsf{W}_{r'}} c_{\mathsf{w}}\mathbf{1}_{P_{\mathsf{w}}} + \text{constant},$$
and hence,
$$\mathbb{E}\mathrm{Var}\Big[\mathbb{E}\big[\sum_{\mathsf{w}\in\mathsf{W}_{r'}} c_{\mathsf{w}}\xi_{\mathsf{w}}(X_u)\,\big|\,X_v\big]\,\Big|\,X_{\mathfrak{p}(v)}\Big] = \mathbb{E}\mathrm{Var}\Big[\mathbb{E}\big[\sum_{\mathsf{w}\in\mathsf{W}_{r'}} c_{\mathsf{w}}\mathbf{1}_{P_{\mathsf{w}}}(X_u)\,\big|\,X_v\big]\,\Big|\,X_{\mathfrak{p}(v)}\Big] \quad (111)$$
$$= \mathbb{E}\mathrm{Var}\Big[\sum_{\mathsf{w}\in\mathsf{W}_{r'}} c_{\mathsf{w}}\mathbf{1}_{P_{\mathsf{w}}}(X_u)\,\Big|\,X_{\mathfrak{p}(v)}\Big],$$
where the second equality follows from that $\sum_{\mathsf{w}\in\mathsf{W}_{r'}} c_{\mathsf{w}}\mathbf{1}_{P_{\mathsf{w}}}(X_u)$ is a function of $X_v$ by Lemma I.4.

Moreover, to show $\mathbb{E}\mathrm{Var}\Big[\sum_{\mathsf{w}\in\mathsf{W}_{r'}} c_{\mathsf{w}}\mathbf{1}_{P_{\mathsf{w}}}(X_u)\,\Big|\,X_{\mathfrak{p}(v)}\Big] > 0$, this is the same as showing
$$\sum_{\mathsf{w}\in\mathsf{W}_{r'}} c_{\mathsf{w}}\mathbf{1}_{P_{\mathsf{w}}}(X_u)$$
is not a function of $X_{\mathfrak{p}(v)}$. By Lemma I.4, this is equivalent to show $\sum_{\mathsf{w}\in\mathsf{W}_{r'}} c_{\mathsf{w}}\mathbf{1}_{P_{\mathsf{w}}}$ is not a linear combination of $\mathbf{1}_P$ for $P \in \mathbf{P}^{r'+1}$, which was proven in Lemma I.17. Therefore, the proof is complete.

$\square$

**Lemma I.22.** *For any given $0 \le r' < r$, suppose $\mathsf{W}_{<r'} := \{\mathsf{w} \in \mathsf{W} : r(\mathsf{w}) < r'\}$ is non-empty. Then, there exists $C \ge 1$ so that the following holds: Let $u, v \in T$ be two nodes such that $v = \mathfrak{p}^{r'}(u)$. We have*
$$\mathbb{E}\mathrm{Var}\Big[\mathbb{E}\big[\sum_{\mathsf{w}\in\mathsf{W}_{<r'}} c_{\mathsf{w}}\xi_{\mathsf{w}}(X_u)\,\big|\,X_v\big]\,\Big|\,X_{\mathfrak{p}(v)}\Big] \le C(\max_{\mathsf{w}\in\mathsf{W}_{<r'}} |c_{\mathsf{w}}|)^2. \quad (112)$$

*Proof.* The proof is more straightforward compared to the arguments presented in the proof of Lemma I.21. First, both sides of (112) scale by a factor $h^2$ if we scaled each $c_{\mathsf{w}}$ by $h \in \mathbb{R}$. Therefore, it suffices to establish the inequality when
$$\mathbb{E}\mathrm{Var}\Big[\mathbb{E}\big[\sum_{\mathsf{w}\in\mathsf{W}_{<r'}} c_{\mathsf{w}}\xi_{\mathsf{w}}(X_u)\,\big|\,X_v\big]\,\Big|\,X_{\mathfrak{p}(v)}\Big] = 1.$$

If there is no $(c_{\mathsf{w}})_{\mathsf{w}\in\mathsf{W}_{<r'}}$ satisfying the above condition, then the proof is completed. Now we assume this set is not empty. Notice that $\mathbb{E}\mathrm{Var}\Big[\mathbb{E}\big[\sum_{\mathsf{w}\in\mathsf{W}_{<r'}} c_{\mathsf{w}}\xi_{\mathsf{w}}(X_u)\,\big|\,X_v\big]\,\Big|\,X_{\mathfrak{p}(v)}\Big]$ is a continuous function of $(c_{\mathsf{w}})_{\mathsf{w}\in\mathsf{W}_{<r'}}$ which takes value $0$ when $(c_{\mathsf{w}})_{\mathsf{w}\in\mathsf{W}_{<r'}} = \vec{0}$. Thus, there is an open ball $B \subseteq \mathbb{R}^{\mathsf{W}_{<r'}}$ centered at $\vec{0}$ such that for $(c_{\mathsf{w}})_{\mathsf{w}\in\mathsf{W}_{r'}} \in B$,
$$\mathbb{E}\mathrm{Var}\Big[\mathbb{E}\big[\sum_{\mathsf{w}\in\mathsf{W}_{<r'}} c_{\mathsf{w}}\xi_{\mathsf{w}}(X_u)\,\big|\,X_v\big]\,\Big|\,X_{\mathfrak{p}(v)}\Big] < 1.$$
On the other hand, by choosing $C$ sufficiently large, the set
$$\Big\{(c_{\mathsf{w}})_{\mathsf{w}\in\mathsf{W}_{r'}} : (\max_{\mathsf{w}\in\mathsf{W}_{<r'}} |c_{\mathsf{w}}|)^2 \le 1/C\Big\},$$
which is the cube of side length $2/C^{1/2}$ centered at $\vec{0}$, is contained in $B$. Therefore, the lemma follows. $\square$

*Proof of Proposition I.5.* From Remark I.19 and Lemma I.20, it remains to show $\mathfrak{B}$ satisfies the last 3 properties stated in the Proposition.

As for the third property, notice that the variance of $\sum_{\mathsf{w}\in\mathsf{W}} c_{\mathsf{w}}\xi_w(X_u)$ is not zero as long as $c_{\mathsf{w}}$ are not identically $0$ for $\mathsf{w} \ne \mathsf{w}_0$. Following the same arguments in the proof of Lemma I.21, the property follows if $C \ge 1$ is sufficiently large.

The last two follows by applying Lemma I.21 and Lemma I.22 to every $0 \le r' < r$ and choosing the constant $C$ can be chosen to be the maximum of those constants $C$ from the two lemmas.

$\square$

## I.3   Proof of Proposition I.1

In this subsection, we consider soley degree 1 polynomial of the leave values.

**Definition I.23.** *For any given $\rho' \in T$ and a degree-1 polynomial $f$ of $\{x_u\}_{u \in L_{\rho'}}$, the function can be expressed uniquely in the form*

$$f(x) = \sum_{\mathsf{w} \in \mathsf{W},\, u \in L_{\rho'}} c_{\mathsf{w},u} \xi_{\mathsf{w}}(x_u) \tag{113}$$

*where $\{\xi_{\mathsf{w}}\}_{\mathsf{w} \in \mathsf{W}}$ is the basis introduced in Proposition I.5.*

*For $u \in T_{\rho'}$, let $f_u(x) := \sum_{\mathsf{w} \in \mathsf{W},\, v \in L_u} c_{\mathsf{w},v} \xi_{\mathsf{w}}(x_v)$. Observe that from this definition, for each $0 \le l \le r$,*

$$f(x) = \sum_{u \in T_{\rho'} \,:\, \mathrm{h}(u)=l} f_u(x).$$

*Further, for $u \in T_{\rho'} \backslash L_{\rho'}$, let*

$$c_{\mathsf{w},u} := \sum_{v \in L_u} c_{\mathsf{w},v}.$$

**Remark I.24.** From the definition above, for each $\rho' \in T$ and degree-1 polynomial $f$ of variables $\{x_u\}_{u \in L_{\rho'}}$, we have

$$\forall u \in T_{\rho'},\ \forall x \in \mathbb{R}^q,\ (\mathbb{E}_u f_u)(x) = \sum_{\mathsf{w}} c_{\mathsf{w},u} \xi_{\mathsf{w}}^{(l)}(x_u),$$

where $\xi_{\mathsf{w}}^{(l)}(\theta)$ is introduced in Remark I.6.

**Proposition I.25.** *There exists a constant $C = C(M,d) \ge 1$ so that the following holds: Suppose*

$$f(x) = \sum_{u \in L_{\rho'},\, \mathsf{w} \in \mathsf{W}} c_{\mathsf{w},u} \xi(x_u)$$

*where $\rho' \in T$ is a node satisfying $\mathrm{h}(\rho') \le r_0$ and*

$$c_{\mathsf{w},u} = c_{\mathsf{w},v}$$

*for $u, v \in L_{\rho'}$ satisfying $\mathrm{h}(\rho(u,v)) \le r(\mathsf{w})$, where $\rho(u,v)$ is the lowest common ancestor of $u$ and $v$. Then,*

$$\sum_{u \in L_{\rho'}} \mathrm{Var}[f_u(X)] \le CR^3 \mathbb{E}\mathrm{Var}\big[f(X) \,|\, X_{\rho'}\big]$$

If Proposition I.25 is proven, then Proposition I.1 follows as a corollary:

*Proof of Proposition I.1.* Reduction to $\mathrm{h}(\rho') \le r_0$: Without loss of generality, it is sufficient to consider degree 1 functions of $L$, rather than degree 1 functions of variables in $D_k(u)$ for some $u$ in the tree and $0 \le k \le \mathrm{h}(u)$.

Recall that
$$D_{r_0}(\rho) = \{w \in T \,:\, \mathrm{h}(w) = r_0\}.$$
We know that we can express $f(x) = \sum_{w \in D_{r_0}(\rho)} f_w(x)$ so that each of them is a degree-1 polynomial with variables $\{x_u\}_{u \in L_w}$.

Together with the variance decomposition for degree-1 polynomials (See Lemma C.1)

$$\mathrm{Var}[f(X)] \ge \sum_{w \in D_{r_0}(\rho)} \mathbb{E}\mathrm{Var}[f_w(X_w) \,|\, X_w],$$

it suffices to prove the same statement for degree-1 polynomials of $x_u$ with $u \in L_{\rho'}$ for $\rho'$ satisfying $\mathrm{h}(\rho') \le r_0$.

Now, we fix such $\rho'$ and consider
$$f(x) = \sum_{\mathsf{w},u \in L_{\rho'}} c_{\mathsf{w},u}\xi_{\mathsf{w}}(x_u).$$

Averaging the Coefficients: For each $\mathsf{w} \in \mathsf{W}$ and for each $u \in D_{r(\mathsf{w})}(\rho')$, we know that for any $v_1, v_2 \in L_u$,
$$\xi_{\mathsf{w}}(X_{v_1}) = \xi_{\mathsf{w}}(X_{v_2})$$
almost surely. As a consequcne, we have
$$\sum_{v \in L_u} c_{\mathsf{w},v}\xi_{\mathsf{w}}(X_v) = \sum_{v \in L_u} \frac{\sum_{v \in L_u} c_{\mathsf{w},v}}{|L_u|}\xi_{\mathsf{w}}(X_v)$$

almost surely. Now, we repeat this averaging process for each $\mathsf{w} \in \mathsf{W}$ and for each $u \in D_{r(\mathsf{w})}(\rho')$. We denote the resulting function by $\tilde{f}$. While $\tilde{f}$ and $f$ may not be the same function, $\tilde{f}(X) = f(X)$ almost surely. On the other hand, $\tilde{f}$ is a function which satisfies the condition in Proposition I.25. Following from the proposition, we have
$$\sum_{u \in L} \mathbb{E}\mathrm{Var}[\tilde{f}_u(X)] \leq CR^3\mathbb{E}\mathrm{Var}[\tilde{f}(X) \mid X_{\rho'}] = CR^3\mathrm{Var}[f(X) \mid X_{\rho'}].$$

The proof is complete. $\qquad\square$

Let us begin with an intermediate step toward the proof of the Proposition I.25.

**Lemma I.26.** *Suppose $f$ is a function described in Definition I.23. For any given $1 \leq l < r$ such that $\mathsf{W}_l := \{\mathsf{w} \in \mathsf{W} : r(\mathsf{w}) = l\}$ is non-empty. Let $u \in T_{\rho'}$ with $\mathrm{h}(u) = r(\mathsf{w})$, suppose*
$$t = \max_{\mathsf{w} \in \mathsf{W}_l} |c_{\mathsf{w},u}| > 0.$$

*Then one of the following statement holds:*

- *Either $\mathbb{E}\mathrm{Var}\big[(\mathbb{E}_u f_u)(X_u) \mid X_{\mathfrak{p}(u)}\big] \geq \frac{\pi_{\min}}{2C_0}t^2$, or*

- *$\max_{\mathsf{w} \in \mathsf{W}_{<l}} |c_{\mathsf{w},u}| \geq \frac{\sqrt{\pi_{\min}}}{2C_0}t$.*

*Here, $C_0 \geq 1$ is the constant $C$ described in Proposition I.5 and $\pi_{\min} := \min_{\theta \in [q]} \pi(\theta)$.*

*Further, in the case when $l = 0$, then we simply have $\mathbb{E}\mathrm{Var}\big[(\mathbb{E}_u f_u)(X_u) \mid X_{\mathfrak{p}(u)}\big] \geq \frac{1}{C_0}t^2$.*

*Proof.* We decompose $(\mathbb{E}_u f_u)(x)$ into three components:
$$(\mathbb{E}_u f_u)(x) = \sum_{\mathsf{w} : r(\mathsf{w}) < l} c_{w,u}\xi_w^{(l)}(x_u) + \sum_{\mathsf{w} : r(\mathsf{w}) = l} c_{w,u}\xi_w^{(l)}(x_u) + \sum_{\mathsf{w} : r(\mathsf{w}) > l} c_{w,u}\xi_w^{(l)}(x_u),$$

where $\xi_w^{(l)}$ is introduced in Remark I.6.

For each $\mathsf{w}$ with $r(\mathsf{w}) > l$, $\xi_w^{(l)}(X_u)$ is a function of $X_v$ with $v = \mathfrak{p}^{r(\mathsf{w})-l}(u)$. Hence, the last component $\sum_{\mathsf{w} : r(\mathsf{w}) > l} c_{w,u}\xi_w^{(l)}(x_u)$ is a constant function whenever we condition on $X_{\mathfrak{p}(u)}$. Consequently,

$$\mathbb{E}\mathrm{Var}\big[(\mathbb{E}_u f_u)(X_u) \mid X_{\mathfrak{p}(u)}\big] = \mathbb{E}\mathrm{Var}\Big[\sum_{\mathsf{w} : r(\mathsf{w}) < l} c_{w,u}\xi_w^{(l)}(X_u) + \sum_{\mathsf{w} : r(\mathsf{w}) = l} c_{w,u}\xi_w^{(l)}(X_u) \,\Big|\, X_{\mathfrak{p}(u)}\Big]. \tag{114}$$

From Proposition I.5, we know that

$$\mathbb{E}\mathrm{Var}\Big[\sum_{\mathsf{w} : r(\mathsf{w}) = l} c_{w,u}\xi_w^{(l)}(X_u) \,\Big|\, X_{\mathfrak{p}(u)}\Big] \geq \frac{t^2}{C_0}, \tag{115}$$

where the constant $C_0$ is the constant $C$ stated in the Proposition. Intuitively, from (115) it should be clear that if the R.H.S. of (114) is small, then $\mathbb{E}\mathrm{Var}\left[\sum_{\mathsf{w}:r(\mathsf{w})<l} c_{w,u}\xi_w^{(l)}(X_u) \,\middle|\, X_{\mathfrak{p}(u)}\right]$ cannot be small. Let us derive this with a coarse estimate.

By (115), we know there exists $\theta \in [q]$ such that

$$\mathrm{Var}\left[\sum_{\mathsf{w}:r(\mathsf{w})=l} c_{w,u}\xi_w^{(l)}(X_u) \,\middle|\, X_{\mathfrak{p}(u)} = \theta\right] \geq \frac{t^2}{C_0}.$$

Now, suppose $\mathrm{Var}\left[\sum_{\mathsf{w}:r(\mathsf{w})<l} c_{w,u}\xi_w^{(l)}(X_u) \,\middle|\, X_{\mathfrak{p}(u)} = \theta\right] < \frac{t^2}{4C_0}$. We could apply triangle inequality to get

$$\sqrt{\mathrm{Var}\left[\sum_{\mathsf{w}:r(\mathsf{w})<l} c_{w,u}\xi_w^{(l)}(X_u) + \sum_{\mathsf{w}:r(\mathsf{w})=l} c_{w,u}\xi_w^{(l)}(X_u) \,\middle|\, X_{\mathfrak{p}(u)} = \theta\right]}$$

$$\geq \sqrt{\mathrm{Var}\left[\sum_{\mathsf{w}:r(\mathsf{w})=l} c_{w,u}\xi_w^{(l)}(X_u) \,\middle|\, X_{\mathfrak{p}(u)} = \theta\right]} - \sqrt{\mathrm{Var}\left[\sum_{\mathsf{w}:r(\mathsf{w})<l} c_{w,u}\xi_w^{(l)}(X_u) \,\middle|\, X_{\mathfrak{p}(u)} = \theta\right]}$$

$$\geq \frac{t}{2C_0^{1/2}},$$

and together with (114),

$$\mathbb{E}\mathrm{Var}\left[(\mathbb{E}_u f_u)(X_u) \,|\, X_{\mathfrak{p}(u)}\right] \geq \pi(\theta)\mathrm{Var}\left[\sum_{\mathsf{w}:r(\mathsf{w})<l} c_{w,u}\xi_w^{(l)}(X_u) + \sum_{\mathsf{w}:r(\mathsf{w})=l} c_{w,u}\xi_w^{(l)}(X_u) \,\middle|\, X_{\mathfrak{p}(u)}\right]$$

$$\geq \frac{\pi_{\min}}{2C_0}t^2.$$

Consider the opposite case where $\mathrm{Var}\left[\sum_{\mathsf{w}:r(\mathsf{w})<l} c_{w,u}\xi_w^{(l)}(X_u) \,\middle|\, X_{\mathfrak{p}(u)} = \theta\right] \geq \frac{t^2}{4C_0}$. First,

$$\mathbb{E}\mathrm{Var}\left[\sum_{\mathsf{w}:r(\mathsf{w})<l} c_{w,u}\xi_w^{(l)}(X_u) \,\middle|\, X_{\mathfrak{p}(u)} = \theta\right] \geq \frac{\pi_{\min}}{4C_0}t^2.$$

By applying the 4th property stated in Proposition I.5, we conclude that

$$\max_{\mathsf{w}:r(\mathsf{w})<l} |c_{\mathsf{w},u}| \geq \frac{\sqrt{\pi_{\min}}}{2C_0}t.$$

In the case when $l = 0$. The argument is simpler, which follows directly from (114) and the Proposition I.5.

$\square$

*Proof of Proposition I.25.* Let $t_0 = \max_{\mathsf{w},u\in L_{\rho'}} |c_{\mathsf{w},u}|$ and let $\mathsf{w}' \in \mathsf{W}$ and $u' \in L_{\rho'}$ be the pair such that $t_0 = |c_{\mathsf{w}',u'}|$. Further, let $l_0 = r(\mathsf{w}')$ and $u_0 = \mathfrak{p}^l(u')$.

If $l_0 > 0$, then we have

$$|c_{\mathsf{w}',u_0}| = \sum_{v\in L_u} |c_{\mathsf{w}',v}| \geq |c_{\mathsf{w}',u'}| = t_0,$$

where the first equality follows from the assumptions of the coefficients. We will try to construct a sequence of triples $(l_k, t_k, u_k)$ indexed by $k$ such that $(l_k)_{k\geq 0}$ is strictly decreasing such that $\mathsf{W}_{l_k} \neq \emptyset$, $\mathrm{h}(u_k) = l_k$, and $t_k = \max_{\mathsf{w}\in\mathsf{W}_{l_k}} |c_{\mathsf{w},u_k}|$.

Suppose we have a triple $(l_k, t_k, u_k)$ such that $l_k \geq 0$, $\mathrm{h}(u_k) = l_k$, $\mathsf{W}_{l_k} \neq \emptyset$, and $t_k = \max_{\mathsf{w}\in\mathsf{W}_{l_k}} |c_{\mathsf{w},u_k}|$ for some index $k \geq 0$.

We apply Lemma I.26 to get

1. Either $\mathbb{E}\mathrm{Var}\left[(\mathbb{E}_{u_k} f_{u_k})(X_{u_k}) \,\middle|\, X_{\mathfrak{p}(u_k)}\right] \geq \frac{\pi_{\min}}{2C_0}t_k^2$, or

2. $\max_{\mathsf{w}\in\mathsf{W}_{<\ell_k}} |c_{\mathsf{w},u_k}| \geq \frac{\sqrt{\pi_{\min}}}{2C_0} t_k$. (This case cannot happen if $\ell_k = 0$.)

If the first case is true, then we terminate the process of finding next triple $(\ell_{k+1}, t_{k+1}, u_{k+1})$.

If the second case is true, let $\mathsf{w}'' \in \mathsf{W}$ be the vertex such that $|c_{\mathsf{w}'',u_k}| = \max_{\mathsf{w}\in\mathsf{W}_{<\ell_k}} |c_{\mathsf{w},u_k}|$ and set $\ell_{k+1} = r(w'')$. Since

$$c_{\mathsf{w},u_k} = \sum_{u\in D_{\ell_{k+1}}(u_k)} c_{\mathsf{w},u},$$

we have

$$t_{k+1} := \max_{u\in T_{u_k}\,:\,\mathrm{h}(u)=\ell_{k+1}} |c_{\mathsf{w},u}| \geq \frac{1}{Rd^{\ell_k-\ell_{k+1}}} |c_{\mathsf{w},u_k}| = \frac{1}{Rd^{\ell_k-\ell_{k+1}}} t_k. \tag{116}$$

Further, let $u_{k+1} = \mathrm{argmax}_{u\in T_{u_k}\,:\,\mathrm{h}(u)=\ell_{k+1}} |c_{\mathsf{w},u}|$.

In this way, we produce a new triple satisfying the same assumption as $(l_k, t_k, u_k)$ described above.

Since $l_0 > l_1 > l_2 \ldots$ is a monotone decreasing chain of non-negative number, it means this argument must terminated in $r_0$ steps. Now, suppose it terminates at the $k$-th step, resulting a triple $(l_k, t_k, u_k)$, and

$$\mathbb{E}\mathrm{Var}\big[(\mathbb{E}_{u_k} f_{u_k})(X_{u_k}) \,\big|\, X_{\mathfrak{p}(u_k)}\big] \geq \frac{\pi_{\min}}{2C_0} t_k^2 \overset{(116)}{\geq} \frac{\pi_{\min}}{2C_0} \Big(\frac{\sqrt{\pi_{\min}}}{2C_0} Rd\Big)^{-2r_0} t_0^2.$$

On the other hand, from Proposition I.5,

$$\sum_{u\in L_{\rho'}} \mathrm{Var}[f_u(X_u)] \leq CRd^{r_0} t_0^2.$$

Therefore, we conclude that

$$\sum_{u\in L_{\rho'}} \mathrm{Var}[f_u(X_u)] \leq C(M,d) R^{2r_0+1} \mathbb{E}\mathrm{Var}\big[f(X)\,|\,X_{\rho'}\big].$$

$\square$

## J  Properties of Markov Chains and Galton-Watson Tree

### J.0.1  Proof of Lemma A.5

Recall that real Jordon Canonical form of $M$ is a $q \times q$ diagonal block matrix $\mathbf{J} = \mathrm{diag}(\mathbf{J}_0, \mathbf{J}_1, \ldots, \mathbf{J}_{s_1})$ for some $s_1 \leq q$.

Since $M$ is ergodic, the eigenspace corresponds to eigenvalue 1 is 1-dimensional. Thus, We may assume $\mathbf{J}_0 = [1]$ is the unique Jordan block corresponds to eigenvalue 1.

For each $s \in [1, s_1]$, $\mathbf{J}_s$ is either a $m_s \times m_s$ matrix of the form $\mathbf{J}_s = \begin{bmatrix} \lambda_s & 1 & & \\ & \lambda_s & \ddots & \\ & & \ddots & 1 \\ & & & \lambda_s \end{bmatrix}$

for some $\lambda_s \in \mathbb{R}$ satisfying $|\lambda_s| \leq \lambda$; or a $J_s$ is a $2m_s \times 2m_s$ matrix of the form $\mathbf{J}_s = \begin{bmatrix} \lambda_s R_s & I_2 & & \\ & \lambda_s R_s & \ddots & \\ & & \ddots & I_2 \\ & & & \lambda_s R_s \end{bmatrix}$, where $|\lambda_s| \leq \lambda$, and $R_s = \begin{bmatrix} \cos(\theta_s) & \sin(\theta_s) \\ -\cos(\theta_s) & \sin(\theta_s) \end{bmatrix}$ is a rotation ma-

trix in $\mathbb{R}^2$ with parameter $\theta_s \in (0, 2\pi)$. In the later case, it corresponds to the conjugate pair of eigenvalues $\lambda_s(\cos(\theta_s) \pm \mathbf{i}\sin(\theta_s))$

According to Jordon Decomposition, there exists an invertible matrix $P$ such that $M = PJP^{-1}$.

For $i \in [1, q-1]$, let $\phi_i$ be the $i + 1$th column of $P$. Because $P$ is invertible, $\{\phi_i\}_{i \in [q]}$ form a linear basis of functions from $[q]$ to $\mathbb{R}$.

Since $\pi$ is a left-eigenvector of $M$ with eigenvalue 1, we have

$$\mathbb{E}_{Y \sim \pi} \phi_i(Y) = \pi^\top \phi_i = 0,$$

because $\phi_i$ is a sum of up to two generalized eigenvectors with eigenvalues not equal to 1.

(A generalized eigenvector $v$ with eigenvalue $\lambda'$ of $M$ is a vector which satisfies $(M - \lambda')^k v = \vec{0}$ for some positive integer $k$. Whenever $\lambda' \neq 1$,

$$\pi^\top v = (\frac{1}{(1 - \lambda')^k} \pi^\top (M - \lambda')^k) v = \frac{1}{(1 - \lambda')^k} \pi^\top \cdot \vec{0} = 0.$$

If index $i$ corresponds to $\mathbf{J}_s$ which associated with a real eigenvalue, then $\phi_i$ is a generalized eigenvector with eigenvalue $\lambda_s$; And if $\mathbf{J}_s$ associates with a complex conjugate pair or eigenvalues, then $\phi_i$ is a sum of two generalized eigenvectors with eigenvalues $\lambda_s(\cos(\theta_s) + \mathbf{i}\sin(\theta_s))$ and $\lambda_s(\cos(\theta_s) - \mathbf{i}\sin(\theta_s))$, respectively. ) As a consequence, every function $f : [q] \mapsto \mathbb{R}$ can be uniquely decomposed in the form

$$f = \mathbb{E}f + \sum_{i \in [q-1]} \delta_i \phi_i. \tag{117}$$

With this unique decomposition, let us define a semi-norm

$$\|f\|_M = \max_{i \in [q-1]} |\delta_i|.$$

**Lemma J.1.** *There exists $C > 0$ so that for every $f : [q] \to \mathbb{R}$,*

$$C^{-1}\|f\|_M^2 \leq \mathrm{Var}_{Y \sim \pi}(f(Y)) \leq C\|f\|_M^2. \tag{118}$$

*Proof.* Without lose of generality, let $f = \sum_{i \in [2,q]} \delta_i \phi_i$, since both $\|f\|_M$ and $\mathrm{Var}_{Y \sim \pi}(f(Y))$ are invariant under a constant shift.

Let $D_\pi = \mathrm{diag}(\pi_1, \ldots, \pi_q)$. Also, let $\vec{\delta} = (0, \delta_2, \ldots, \delta_q)$. Then,

$$\|f\|_M = \|\vec{\delta}\|_\infty \qquad \text{and} \qquad \mathrm{Var}_{Y \sim \pi}(f(Y)) = \vec{\delta}^\top P^\top D_\pi P \vec{\delta}.$$

Let $s_{\max}$ and $s_{\min}$ be the maximum and minimum singular value of $P^\top D_\pi P$, respectively. Together with $q^{-1/2}\|\vec{\delta}\|_2 \leq \|\vec{\delta}\|_\infty \leq \|\vec{\delta}\|_2$, we have

$$s_{\min}^2 q^{-1}\|f\|_M^2 \leq s_{\min}^2 q^{-1}\|\vec{\delta}\|_2^2 \leq \mathrm{Var}_{Y \sim \pi}(f(Y)) \leq s_{\max}^2 \|\vec{\delta}\|_2^2 \leq s_{\max}^2 \|f\|_M^2. \tag{119}$$

If $s_{\min} > 0$, then we can complete the proof by taking $C = \max\{s_{\max}^2, q/s_{\min}^2\}$. It remains to show that $s_{\min} > 0$, or equvialently $P^\top D_\pi P$ is invertible. Because $M$ is ergodic, each entry of $\pi$ is positive, and thus $D_\pi$ is invertible. Hence, $P^\top D_\pi P$ is invertible because it is a product of three invertible matrices.

$\square$

**Lemma J.2.** *There exists $C > 0$ so that for every $f : [q] \to \mathbb{R}$,*

$$C^{-1}\|f\|_M \leq \|f - \mathbb{E}_{Y \sim \pi} f(Y)\|_\infty \leq C\|f\|_M. \tag{120}$$

*Proof.* This simply follows from both $\|f\|_M$ and $\|f - \mathbb{E}f\|_\infty$ are both norms on the finite dimensional space $\{f : [q] \to \mathbb{R} : \mathbb{E}_{Y \sim \pi} f(Y) = 0\}$. $\square$

**Lemma J.3.** *There exists $C \geq 1$ depending on $M$ such that For any function $f : [q] \mapsto \mathbb{R}$ and $k \in \mathbb{N}$,*

$$\|M^k f\|_M \leq Ck^q \lambda^k \|f\|_M. \tag{121}$$

**Remark J.4.** Notice that $M^k f$ can be interpreted as

$$\mathbb{E}\big[f(X_u) \,\big|\, X_{\mathfrak{p}^k(u)} = i\big] = (M^k f)(i),$$

for every $u \in T$ where $\mathfrak{p}^k(u)$ is well-defined.

*Proof.*

$$\|M^k f\|_M = \|P\mathbf{J}^k(\sum_{i\in[q]} \delta_i e_i)\|_M = \|\mathbf{J}^k(\sum_{i\in[2,q]} \delta_i e_i)\|_\infty. \leq q \max_{i\in[2,q]} \|\delta_i\| \max_{i,j\in[2,q]} |\mathbf{J}^k_{ij}|. \tag{122}$$

Notice that $\mathbf{J}^k$ is the diagonal block matrix whose blocks $\mathbf{J}^k_s$ for $s \in [s_1]$. The block $\mathbf{J}^k_s$ can be computed directly: In the case when $\mathbf{J}_s$ corresponds to a complex conjugate pair of eigenvalues,

$$\mathbf{J}^k_s = \begin{bmatrix} \lambda_s^k R_s^k & \binom{k}{1}\lambda_s^{k-1}R_s^{k-1} & \cdots & \binom{k}{m_s-1}\lambda_s^{k-m_s+1}R_s^{k-m_s+1} \\ & \lambda_s^k R_s^k & \ddots & \vdots \\ & & \ddots & \binom{k}{1}\lambda_s^{k-1}R_s^{k-1} \\ & & & \lambda_s^k R_s^k, \end{bmatrix} \tag{123}$$

where we treat $\binom{k}{r} = 0$ if $r > k$. It can be verified directly by induction, relying on the identity $\binom{k}{r-1} + \binom{k}{r} = \binom{k+1}{r}$. Further, removing the $R_s$ terms in the above equation we obtain the formula for $\mathbf{J}^k_s$ when $\mathbf{J}_s$ corresponds to a real eigenvalue.

Therefore, with $\binom{k}{q} \leq k^q$, $|\lambda_s^r| \leq \lambda^r$, and $\max_{i,j} R_{s\,ij}^r < 1$ for $r \geq 1$, we obtain the bound

$$\max_{s\in[2,q]} \max_{i,j\in[q]} |(\mathbf{J_s}^k)_{ij}| \leq C' k^q \lambda^k, \tag{124}$$

where $C'$ is a constant which depends on $q$ and $\lambda$.

Now we substitute the above bound into (122) to get

$$\|M^k f\|_M \leq q C' k^q \lambda^k \|f\|_M.$$

The proof is completed by taking $C = qC'$.

$\square$

*Proof of Lemma A.5.* The proof of Lemma A.5 follows from the $\|\cdot\|_M$ decay from Lemma J.3 and that both $\mathrm{Var}[f]$ and $\|f - \mathbb{E}f\|_\infty$ are comparable to $\|f\|_M$ within a constant multiplicative factor (Lemma J.1 and Lemma J.2). $\square$

