# OpenReview forum: "Low Degree Hardness for Broadcasting on Trees"
_NeurIPS.cc/2024/Conference — NeurIPS 2024 poster_

### Official Review · Reviewer_8xrJ · 2024-07-03

**Soundness:** 4
**Presentation:** 3
**Contribution:** 4
**Rating:** 8
**Confidence:** 2

**Summary:**

The authors study hardness of broadcasting on trees for low degree polynomials. The main result shows that log(N) degree polynomials of the leaf values have vanishing correlation with the root, resolving the main open question of Kohler and Mossel (NeurIPS 2022).

The tree broadcasting problem consists of a $d$-ary depth $\ell$ tree T (or some relaxation thereof), an ergodic Markov chain $M$ over state space $[q]$, and a starting root distribution $\nu$ over $[q]$. The broadcasting process starts by drawing $j \sim \nu$, then `broadcasts’ a value to every child in the tree based on the transition matrix $M$. This is repeated $\ell$ times until the process terminates when it hits all leaves. Roughly speaking, the broadcasting problem asks whether it is possible to infer the starting value at the root given the values at the leaves of the tree given by the above process.

The most basic algorithm for tree broadcasting looks at linear statistics of the leaves, e.g. count statistics of how many times each state in [q] appears. There is a well-known threshold for linear algorithms called the Kesten-Stigum bound which states that such inference is possible if and only if $d\lambda^2 < 1$, where $\lambda$ is the 2nd largest eigenvalue of $M$. It is a natural (and known open) question whether this bound continues to hold against *low degree polynomials* of the leaves. This work answers this question in the positive for polynomials up to degree $\log(|T|)$: all such functions have vanishing correlation with the root.

**Strengths:**

Broadcasting is a well studied and useful model. The question of how well low-degree statistics of the leaves correlate with the root is extremely natural mathematically, and is supported by a number of recent works showing the general power of low-degree polynomials in predicting statistical-computational gaps. The only known bounds for this general problem were for the very specialized $\lambda=0$ case.

The techniques introduced by the authors, including the notion of fractal capacity (a more involved proxy for low degree functions that aids their proof), and the resulting inductive method for low-degree (low fractal capacity) functions may be of independent interest. As the authors note, it is one of the first methods for dealing with strongly non-product structures in this context, which is a very frequent barrier throughout related areas (e.g. analysis of Markov chains, boolean function analysis, etc).

The authors give a nice overview of their method in the symmetric linear case that is easy to follow (though a bit dense with the notions of fractal capacity introduced before this overview). Moving to the low-degree case requires highly non-trivial modifications of this method, but the intuition given there is very helpful.

**Weaknesses:**

I do not see any substantial weaknesses in this work.

One could of course ask for lower bounds against super-logarithmic degree polynomials as in known in the $\lambda=0$ case, but the results in this paper still mark a major step forward on this problem.

I would request the authors run a grammar/syntax check. There are a huge number of typographical/syntax errors that slow down the reading of the work to some extent.

**Questions:**

N/A

---

### Official Review · Reviewer_Pvza · 2024-07-12

**Soundness:** 4
**Presentation:** 4
**Contribution:** 3
**Rating:** 7
**Confidence:** 3

**Summary:**

The paper shows that for a Markov process in a tree, propagating a starting state at the root to the leaves it is "hard" to infer the starting state from the leaf states.
Concretely they show that for any function $f$ of the leaf states with bounded Efron-Stein degree the variance of $f$ conditioned on the root state is bounded by a function of the total variance of $f$. That is, the output of such a function varies relatively little with respect to the choice of the root value which means that little can be inferred about the root value from the leaf values.
In particular since bounded degree polynomials have bounded Efron-Stein degree this implies that any low degree polynomial of the leaf values can not be correlated with the root, i.e. as tree depth goes to infinity the correlation goes to zero.This holds even for cases where it would be possible to recover some information about the root, so the result really does imply some bound on the power of low-degree functions.

**Strengths:**

The paper shows a very nice result, and I feel like the overview in the main body gives a good idea of how the proof of the result proceeds.

**Weaknesses:**

There are no real weaknesses to the paper, however due to space constraints it is not really possible to ascertain the claimed results from the main body of the paper. I could not check the appendix carefully.

**Questions:**

- p2,l60 there is a broken reference
- On page 6 the application of the CS-inequality has a typo in the final term, it should be $i\in [m]$ below the sum
- p7,l203: of in a Markov Chain -> of a / in a
- p8,l229 there is a missing ")" before the first square

**Limitations:**

No Limitations, this is theoretical research first and foremost

---

### Official Review · Reviewer_j6Pu · 2024-07-14

**Soundness:** 3
**Presentation:** 1
**Contribution:** 3
**Rating:** 6
**Confidence:** 2

**Summary:**

This paper obtained low degree hardness results for broadcasting on trees (BOT).
The BOT problem is as follows.
Consider a rooted tree and an associated Markov process defined by a transition matrix $M$.
The process is initialized by a distribution on the root.
Then for each vertex with state $s$, its children are i.i.d. according to the $s$-th row of $M$.
The goal is to infer the state of the root given all leaves.

This paper shows that below the Kesten--Stigum threshold, any polynomial of the leaves of degree at most $\log N$ fails to recover the root state.
This is particularly interesting since for some channels the KS threshold is information theoretically sharp and therefore they exhibit a stat-comp gap.

**Strengths:**

NA

**Weaknesses:**

NA

**Questions:**

I'm not familiar with this literature but I did try to check some of the cited papers to get a sense.
The result of this paper is certainly above the bar.
However, there's (big) room for improvement of the presentation.
In fact, given the relevance of BOT and low-deg, upon proper revision of the exposition, the authors may want to send this paper to a journal instead.

Major comments:

1. A related work section at the level of technique (i.e., low-deg) is needed.
To balance the space, the authors may want to reduce some parts of Sec 1.1 which are quite standard.

1. Do the notions of $\mathcal{B}(\mathcal{A})$ and fractal capacity exist in the literature, or did the authors introduce them?
It's better to make it clear since the mere definitions of these are a nontrivial contribution IMO.

Minor comments:

1. Line 9, what do you mean by "high level of complexity"? It was just mentioned that BP runs in linear time.

1. Line 10, at this point, it's not even clear what "chain" means. Please try to make the abstract self-contained.

1. Line 20, based on.

1. Line 22 doesn't parse grammatically.

1. Line 22-25, please provide references.

1. I don't understand line 37. Why compare low-deg **algorithms** to SoS **lower bounds**?
Also, what does "easy to use" mean?

1. Line 40, [36] doesn't seem to involve AMP. Do you mean https://arxiv.org/abs/2212.06996 ?

1. Line 42, "broadcast on trees" or "broadcasting on trees"? Please unify the terminology.

1. Line 47, What do you mean by "a linear estimator in the number of leaves"?
The estimator is a linear function of the leaves?
Or it runs in linear time as a function of the number of leaves?

1. Line 54, our --> out

1. Line 56, does "sufficiently many states" mean $q\ge C$ for $C>5$? If so, it's better to make this clear.

1. Line 60, broken reference.

1. The discussion in line 54-61 is written in a chaotic way.
Please reorganize the relevant existing results.

1. Line 68, leaves of for

1. I don't understand the point of line 67-70.
Is the message that for **small** $d$, large degree polynomials fail, but for **large** $d$, efficient reconstruction is possible?
In any case, these lines can be made more clear.

1. Line 93, the notation $T$ hasn't even been introduced, so no need to abuse it.

1. Line 96, begin with define

1. Line 126, the notation $X_A = (X_v)_{v\in A}$ hasn't been defined so far (correct me otherwise).

1. Could the authors comment on the equation between line 139-140?
It says that conditioned on the root, the variance of (a low degree function of) the leaves will drop drastically (exponentially in depth).
This seems to imply that $f(X_L)$ is highly correlated with $X_\rho$ and the correlation is **increasing** with the depth.
Apparently my interpretation is very wrong and contradicts the main message of the paper.
Could the authors remark on why this is the correct statement to prove and how this implies Corollary 1.8?

1. Corollary 1.8: Please define the correlation $\mathrm{Cor}$.

1. Line 146, "the main result is optimal in the fractal sense". This is interesting.
Could the authors expand on this (maybe after the fractal capacity and stuff are properly defined later)?

1. Line 154, the font of $b_1, \dots, b_k$ changed.

1. Definition 1.11, to make sure I understand it correctly, $\mathcal{B}(\mathcal{A})$ is **not** the closure (under decomposition) of $\mathcal{A}$, right?

1. Line 171, for $i$ for $i$

1. Line 178, redundant line between end of proof and $\square$.

1. Definition 1.14, an $\mathcal{A}$-polynomial

1. Line 193, $2$-ary --> binary

1. Line 194, eigenvalue --> eigenvalues

1. Line 199, including in cases

1. End of page 6, comparing to --> compared to

1. The last equation of page 6 is unnecessary.

1. Equation in line 202, is last $\lesssim$ simply $=$?

1. I(1), what is $S$?

1. I(2), what is $S'$? In fact $S'$ is not even used in I(2).

1. Line 229, What is $X$? I don't think $X$ is the whole tree?
Also, there's a missing right parenthesis.

1. Line 230, satisfies --> satisfy.

1. Line 234, "builds on this strategy", which strategy? This sentence feels out of place.

1. Equation above line 242, what is $\mathcal{J}$?

1. Line 242, whose variables is --> are

1. Below line 244, $x_{x_{\le w_1}}$.

1. Line 252, will also holds --> hold

1. Equation above line 262, the argument of the function is included on the LHS but not on the RHS of the equation.

1. Somewhere near the end of page 9, the font of $h_{\mathcal{A}_k}$ changed.

1. Not that it matters, but I don't think the authors used the latest version of the NeurIPS template which has a more tedious checklist.

---

> ### Author Response · Authors · 2024-08-06
>
> We sincerely appreciate the reviewer to carefully go over the manuscript. The feedback is valuable and we will address the issues raised by the reviewer in the revised version of the paper. Below we will address the some major issues raised by the reviewer:
>
> 1.    Low-degree polynomial
> Thanks for suggesting to add references related low-degree polynomials. We should definitely included some, including in terms of the low degree algorithms for stochastic-block models. We note that almost all the work in this area uses independence of random variables in a strong way and our results are novel as they prove low-degree lower bounds in a setup where they is no obvious way to present the underlying variables as a product measure.
>
> 2. Fractal capacity
>
> To our best knowledge, the definition of fractal capacity is new. However, there are many notions of capacity of fractal so ours may be related to some. We will add standard references to fractal capacity.
>
> For the limited space, below we will address those mathematical questions raised by the reviewer:
>
> - Q1:
> The statement about BP being linear pertains to the size of $|L|$. Thus, while it is "linear", it is actually a polynomial of Efron-Stein degree exponential in the depth $\ell$.
>
> - Q6-7:
> We wanted to briefly discuss the classes of algorithms captured by low-degree polynomials. This is why we mention that they are considered more efficient than SoS algorithms of similar degrees and also that they capture many local / spectral and AMP algorithms. The statement has caveats so perhaps we can being the paragraph in the paper by saying ``it is often argued that".
> - Q19:
> Consider the random variable $$Y = \mathbb{E}[f(X_L)\,|\, X_\rho]\,.$$
> Then, $Y$ can be interpreted as a function of $X$:
> When $X = \theta$,
>     $Y$ is the expected value of $f(X_L)$ condition on $X_\rho = \theta$. The statement of the theorem states that ${\rm Var}(Y)$ is exponentially small in $\ell$ comparing to ${\rm Var}(f(X_L))$.
> Suppose  $\text{Var}(Y) = 0$ . This means the conditional expectation is the same regardless of the value of  $X_\rho$ , which implies that  $f(X_L)$  and  $X_\rho$  have zero correlation. This can be verified using the definition of correlation:
> \begin{align*}
> {\rm Corr}(f(X_L), g(X_\rho)) &= \frac{{\rm Cov}(f(X_L), g(X_\rho))}{{\rm Var}(f(X_L))^{1/2} {\rm Var}(g(X_\rho))^{1/2}} .
> \end{align*}
> Now, for the covariance, we can estimate it via conditioning on $X_\rho$ and then apply Cauchy-Schwartz inequality to get
> $$
> {\rm Cov}(f(X_L), g(X_\rho))
> = \mathbb{E}\Big[(f(X_L) - \mathbb{E}f) (g(X_\rho) - \mathbb{E}g)\Big]  = \mathbb{E} \Big[\mathbb{E}\big[(f(X_L) - \mathbb{E}f) (g(X_\rho) - \mathbb{E}g) \,\big|\, X_\rho\big] \Big]
> \le  \mathbb{E}\Big[\mathbb{E}\big[(f(X_L) - \mathbb{E}f)  \,\big|\, X_\rho\big] (g(X_\rho) - \mathbb{E}g) \Big] $$
> $$\le  \sqrt{\mathbb{E}\Big(\mathbb{E}\big[(f(X_L) - \mathbb{E}f)\Big)^2} \sqrt{ \mathbb{E}(g - \mathbb{E}g)^2}
> =   \sqrt{{\rm Var}[ \mathbb{E}[f(X_L)\,|\, X_\rho]]} \sqrt{ {\rm Var}g}. $$
>     Substituting it back and apply the inequality stated in the theorem, we get
>   $$
>             {\rm Corr}(f(X_L), g(X_\rho))
>         \le
>             (\max\{d\lambda^2, d\lambda\})^{\ell/8},
>     $$
>     or applied to the assumption ${\rm Var}(Y)=0$ to get the Correlation is also $0$.\\
>
> - Q21: By "optimal' here it means that, the fractal capacity of a set $S$ can be at most $\ell+1$. Yet, every polynomial of the leaves with fractal-capacity $\le c\ell$ has exponential small correlation with the root $X_\rho$.
>     In other words, using the trivial upper bound, we captured the correct order of the fractal capacity when reconstruction is not possible. We will added this to the manuscript.
>
> - Q23:
>     Yes, $\mathcal{B}(\mathcal{A})$ in general is not the closure of $\mathcal{A}$.
>
>     Notice that while we define $\mathcal{B}(\mathcal{A})$ for any subcollection of leaves, we only analyze $\mathcal{B}(\mathcal{A})$ in the case when $\mathcal{A}$ is itself closed under decomposition. For such $\mathcal{A}$, $\mathcal{B}(\mathcal{A})$ will get larger as long as $\mathcal{A}$ does not contain every non-empty subset of $L$.
>     Consider the simplest case when $\mathcal{A} = \mathcal{A}_1$, which is the set of singletons. Then it is not hard to check that every two element sets is contained in ${\mathcal B}({\mathcal A})$.
>
> - Q33 and 35:   Here we use $f_\alpha(x_S)$ to indicate that $f_\alpha$ is a function of $x_S$ for some $S \subseteq L$, and the same for $f_\beta(x_{S'})$. And sometimes we simply write $f_\alpha(x)$ without specifying the the set $x_S$ (for Q35.) For I(2), there is a typo to be corrected: it should be "$S' \subseteq L$ satisfying $S' \cap \{v' \in L : v' \leq w'\} = \emptyset$." For the equation between 223 and 224 to hold, one simply needs $S' \cap \{v' \in L : v' \leq w'\} = \emptyset$, so that one could apply conditional independence.

---

> ### Comment · Reviewer_j6Pu · 2024-08-11
>
> I thank the authors for the detailed response.
>
> In the response to Q19 where $\mathrm{Cov}(f(X_L), g(X_\rho))$ is upper bounded, there seem to be some inaccuracies, if I'm not mistaken.
>
> - The first $\le$ is an equality.
> - In the second $\le$, there is a missing conditioning on $X_\rho$ in the first term.
>
> Others look good to me.
> I have raised my score to 6.

---

> > ### Author Response · Authors · 2024-08-11
> >
> > We would like to thank the reviewer for the consideration!
> >
> > For Q19, the reviewer's comment is correct. The first inequality is indeed an inequality, and the second inequality is missing the conditioning.

---

### Official Review · Reviewer_d9ko · 2024-07-14

**Soundness:** 3
**Presentation:** 3
**Contribution:** 3
**Rating:** 7
**Confidence:** 2

**Summary:**

This submission considers the broadcasting on trees problem, where given a rooted tree, information is propagated from the root to the leaves using a Markov process. The algorithmic task is to infer the information at the root given only the information at the leaves. Previous works had identified that the KS-threshold (a threshold based on the spectral gap of the Markov process and the degree-structure of the tree) is the critical threshold when this is possible information-theoretically in some specific cases. However, in general it is not the right information-theoretic threshold, i.e., in some cases inference is possible even below it. The submission gives evidence that it is indeed the right threshold however when considering computationally bounded algorithms. In particular, it establishes hardness in the low-degree framework below the KS-threshold.

**Strengths:**

Previous work had asked the question whether the KS-threshold is the correct threshold for efficient algorithms. This work answers this question in the affirmative. The ideas used in the proof are novel and clever.

**Weaknesses:**

The first part of the introduction is a bit hard to follow for non-experts, maybe some additional context would help.

**Questions:**

No questions.

**Limitations:**

Yes

---

### Decision · Program_Chairs · 2024-09-25

**Decision:**

Accept (poster)

**Comment:**

This paper answers a natural open problem regarding efficient algorithms for broadcasting on trees. The reviewers agree that the technical contribution of the paper is substantial and clearly above the NeurIPS bar. The only doubt I have is that this paper seems better suited for a theory conference (like STOC or FOCS) than for an ML conference. On the other hand, some of the cited related work appeared in ML venues, and none of the reviewers seemed bothered by the choice of venue, so I recommend acceptance. The reviewers pointed out many places where the presentation quality is suboptimal, and I expect the authors to work on this aspect of the paper for the camera-ready version.